# Cardinality-Aware Set Prediction and Top-$k$ Classification

**Corinna Cortes**
Google Research
New York, NY 10011
corinna@google.com

**Anqi Mao**
Courant Institute
New York, NY 10012
aqmao@cims.nyu.edu

**Christopher Mohri**
Stanford University
Stanford, CA 94305
xmohri@stanford.edu

**Mehryar Mohri**
Google Research & CIMS
New York, NY 10011
mohri@google.com

**Yutao Zhong**
Courant Institute
New York, NY 10012
yutao@cims.nyu.edu

## Abstract

We present a detailed study of cardinality-aware top-$k$ classification, a novel approach that aims to learn an accurate top-$k$ set predictor while maintaining a low cardinality. We introduce a new target loss function tailored to this setting that accounts for both the classification error and the cardinality of the set predicted. To optimize this loss function, we propose two families of surrogate losses: cost-sensitive comp-sum losses and cost-sensitive constrained losses. Minimizing these loss functions leads to new cardinality-aware algorithms that we describe in detail in the case of both top-$k$ and threshold-based classifiers. We establish $\mathcal{H}$-consistency bounds for our cardinality-aware surrogate loss functions, thereby providing a strong theoretical foundation for our algorithms. We report the results of extensive experiments on CIFAR-10, CIFAR-100, ImageNet, and SVHN datasets demonstrating the effectiveness and benefits of our cardinality-aware algorithms.

## 1   Introduction

Top-$k$ classification consists of predicting the $k$ most likely classes for a given input, as opposed to solely predicting the single most likely class. Several compelling reasons support the adoption of this framework. First, it enhances accuracy by allowing the model to consider the top $k$ predictions, accommodating uncertainty and providing a more comprehensive prediction. This is particularly valuable in scenarios where multiple correct answers exist, such as image tagging, where a top-$k$ classifier can identify multiple relevant objects in an image. Second, top-$k$ classification is applicable in ranking and recommendation tasks such as suggesting the top $k$ most relevant products in e-commerce based on user queries. The confidence scores associated with the top $k$ predictions also serve as a means to estimate the model's uncertainty, which is crucial in applications requiring insight into the model's confidence level.

The predictions of a top-$k$ classifier are also useful in several natural settings. For example, ensemble learning can benefit from top-$k$ predictions as they can be combined from multiple models, contributing to improved overall performance by introducing a more robust and diverse set of predictions. In addition, top-$k$ predictions can serve as input for downstream tasks like natural language generation or dialogue systems, enhancing the performance of these tasks by providing a broader range of potential candidates. Finally, the interpretability of the model's decision-making process is enhanced by examining the top $k$ predicted classes, allowing users to gain insights into the rationale behind the model's predictions.

38th Conference on Neural Information Processing Systems (NeurIPS 2024).

The appropriate $k$ for a task at hand may be determined by the application itself like a recommendor system always expecting a fixed set size to be returned. For other applications, it may be natural to let the cardinality of the returned set vary with the model's confidence or other properties of the task. Designing effective algorithms with learning guarantees for this setting is our main goal.

In this paper, we introduce the problem of cardinality-aware set prediction, which is to learn an accurate set predictor while maintaining a low cardinality. The core idea is that an effective algorithm should dynamically adjust the cardinality of its prediction sets based on input instances. For top-$k$ classifiers, this means selecting a larger $k$ for difficult inputs to ensure high accuracy, while opting for a smaller $k$ for simpler inputs to maintain low cardinality. Similarly, for threshold-based classifiers, a lower threshold can be used for difficult inputs to minimize the risk of misclassification, whereas a higher threshold can be applied to simpler inputs to reduce cardinality.

To tackle this problem, we introduce a novel target loss function which captures both the classification error and the cardinality of a prediction set. Minimizing this target loss function directly is an instance-dependent cost-sensitive learning problem, which is intractable for most hypothesis sets. Instead, we derive two families of general surrogate loss functions that benefit from smooth properties and favorable optimization solutions.

To provide theoretical guarantees for our cardinality-aware top-$k$ approach, we first study consistency properties of surrogate loss functions for the general top-$k$ problem with a fixed $k$. Unlike standard classification, the consistency of surrogate loss functions for the top-$k$ problem has been relatively unexplored. A crucial property in this context is the asymptotic notion of *Bayes-consistency*, which has been extensively studied in standard binary and multi-class classification [Zhang, 2004a, Bartlett et al., 2006, Zhang, 2004b, Bartlett and Wegkamp, 2008]. While Bayes-consistency has been explored for various top-$k$ surrogate losses [Lapin et al., 2015, 2016, 2018, Yang and Koyejo, 2020, Thilagar et al., 2022], some face limitations. Non-convex "hinge-like" surrogates [Yang and Koyejo, 2020], surrogates inspired by ranking [Usunier et al., 2009], and polyhedral surrogates [Thilagar et al., 2022] cannot lead to effective algorithms as they cannot be efficiently computed and optimized. Negative results also indicate that several convex "hinge-like" surrogates [Lapin et al., 2015, 2016, 2018] fail to achieve Bayes-consistency [Yang and Koyejo, 2020]. On the positive side, it has been shown that the logistic loss (or cross-entropy loss used with the softmax activation) is a Bayes-consistent loss for top-$k$ classification [Lapin et al., 2015, Yang and Koyejo, 2020].

We show that, remarkably, several widely used families of surrogate losses used in standard multi-class classification admit $\mathcal{H}$-*consistency bounds* [Awasthi, Mao, Mohri, and Zhong, 2022a,b, Mao, Mohri, and Zhong, 2023f,b] with respect to the top-$k$ loss. These are strong non-asymptotic consistency guarantees that are specific to the actual hypothesis set $\mathcal{H}$ adopted, and therefore also imply asymptotic Bayes-consistency. We establish this property for the broad family of *comp-sum losses* [Mao, Mohri, and Zhong, 2023f], comprised of the composition of a non-decreasing and non-negative function with the sum exponential losses. This includes the logistic loss, the sum-exponential loss, the mean absolute error loss, and the generalized cross-entropy loss. Additionally, we extend these results to *constrained losses*, a family originally introduced for multi-class SVM [Lee et al., 2004], which includes the constrained exponential, hinge, squared hinge, and $\rho$-margin losses. The guarantees of $\mathcal{H}$-consistency provide a strong foundation for principled algorithms in top-$k$ classification by directly minimizing these surrogate loss functions.

We then leverage these results to derive strong guarantees for the two families of cardinality-aware surrogate losses: cost-sensitive comp-sum and cost-sensitive constrained losses. Both families are obtained by augmenting their top-$k$ counterparts [Lapin et al., 2015, 2016, Berrada et al., 2018, Reddi et al., 2019, Yang and Koyejo, 2020, Thilagar et al., 2022] with instance-dependent cost terms. We establish strong $\mathcal{H}$-consistency bounds, implying Bayes-consistency, for both families relative to the cardinality-aware target loss. Our $\mathcal{H}$-consistency bounds for the top-$k$ problem are further beneficial here in that the cardinality-aware problem can consist of fixing and selecting from a family top-$k$ classifiers–we now know how to effectively learn each top-$k$ classifier.

The rest of the paper is organized as follows. In Section 2, we formally introduce the cardinality-aware set prediction problem along with our new families of surrogate loss functions. Section 3 instantiates our algorithms in the case of both top-$k$ classifiers and threshold-based classifiers, and Section 4 presents strong theoretical guarantees. In Section 5, as well as in Appendix J and Appendix K, we present experimental results on the CIFAR-10, CIFAR-100, ImageNet, and SVHN datasets, demonstrating the effectiveness of our algorithms.

## 2    Cardinality-aware set prediction

In this section, we introduce cardinality-aware set prediction, where the goal is to devise algorithms that dynamically adjust the prediction set's size based on the input instance to both achieve high accuracy and maintain a low average cardinality. Specifically, for top-$k$ classifiers, our objective is to determine a suitable cardinality $k$ for each input $x$, with higher values of $k$ for instances that are more difficult to classify.

To address this problem, we first define a cardinality-aware loss function that accounts for both the classification error and the cardinality of the set predicted (Section 2.1). However, minimizing this loss function directly is computationally intractable for non-trivial hypothesis sets. Thus, to optimize it, we introduce two families of surrogate losses: cost-sensitive comp-sum losses (Section 2.2) and cost-sensitive constrained losses (Section 2.3). We will later show that these loss functions benefits from favorable guarantees in terms of $\mathcal{H}$-consistency (Section 4.3).

### 2.1    Cardinality-aware problem formulation and loss function

The learning setup for cardinality-aware set prediction is as follows.

**Problem setup.**    We denote by $\mathcal{X}$ the input space and $\mathcal{Y} = [n] := \{1, \ldots, n\}$ the label space. Let $\{g_k : k \in \mathcal{K}\}$ denote a collection of given set predictors, induced by a parameterized set predictor $g_k : \mathcal{X} \mapsto 2^{\mathcal{Y}}$, where each $\mathcal{K} \subset \mathbb{R}$ is a set of indices. This could be a subset of the family of top-$k$ classifiers induced by some classifier $h$, or a family of threshold-based classifiers based on some scoring function $s : \mathcal{X} \times \mathcal{Y} \mapsto \mathbb{R}$. In that case, $g_k(x)$ then comprises the set of $y$s with a score $s(x, y)$ exceeding the threshold $\tau_k$ defining $g_k$. This formulation covers as a special case standard conformal prediction set predictors [Shafer and Vovk, 2008], as well as set predictors defined as confidence sets described in [Denis and Hebiri, 2017]. We will denote by $|g_k(x)|$ the cardinality of the set $g_k(x)$ predicted by $g_k$ for the input $x$. To simplify the discussion, we will assume that $|g_k(x)|$ is an increasing function of $k$, for any $x$. For a family of top-$k$ classifiers or threshold-based classifiers, this simply means that they are sorted in increasing order of $k$ or decreasing order of the threshold values.

To account for the cost associated with cardinality, we introduce a non-negative and increasing function $\mathsf{cost} : \mathbb{R}_+ \to \mathbb{R}_+$, where $\mathsf{cost}(|g_k(x)|)$ represents the *cost* associated to the cardinality $|g_k(x)|$. Common choices for cost include $\mathsf{cost}(|g_k(x)|) = |g_k(x)|$, or a logarithmic function $\mathsf{cost}(|g_k(x)|) = \log(|g_k(x)|)$ as in our experiments (see Section 5), to moderate the magnitude of the cost relative to the binary classification loss. Our analysis is general and requires no assumption about cost.

Our goal is to learn to assign to each input instance $x$ the most appropriate index $k \in \mathcal{K}$ to both achieve high accuracy and maintain a low average cardinality.

**Cardinality-aware loss function.**    As in the ordinary multi-class classification problem, we consider a family $\mathcal{R}$ of scoring functions $r : \mathcal{X} \times \mathcal{K} \to \mathbb{R}$. For any $x$, $r(x, k)$ denotes the score assigned to the *label* (or index) $k \in \mathcal{K}$, given $x \in \mathcal{X}$. The label predicted is $\mathsf{r}(x) = \operatorname{argmax}_{k \in \mathcal{K}} r(x, k)$, with ties broken in favor of the largest index. To account for both classification accuracy and cardinality cost, we define the *cardinality-aware loss function* for a scoring function $r$ and input-output label pair $(x, y) \in \mathcal{X} \times \mathcal{Y}$ as a linearized loss of these two criteria:

$$\ell(r, x, y) = 1_{y \notin g_{\mathsf{r}(x)}(x)} + \lambda \, \mathsf{cost}(\left|g_{\mathsf{r}(x)}(x)\right|), \tag{1}$$

where the first term is the standard loss for a top-$k$ prediction taking the value one when the correct label $y$ is not included in the top-$k$ set and zero otherwise, and $\lambda > 0$ is a hyperparameter that governs the balance between prioritizing accuracy versus limiting cardinality. The learning problem then consists of using a labeled training sample $(x_1, y_1), \ldots (x_m, y_m)$ drawn i.i.d. from some (unknown) distribution $\mathcal{D}$ to select $r \in \mathcal{R}$ with a small expected cardinality-aware loss $\mathbb{E}_{(x,y) \sim \mathcal{D}}[\ell(r, x, y)]$.

The loss function (1) can be equivalently expressed in terms of an instance-dependent cost function $c : \mathcal{X} \times \mathcal{K} \times \mathcal{Y} \to \mathbb{R}_+$:

$$\ell(r, x, y) = c(x, \mathsf{r}(x), y), \tag{2}$$

where $c(x, k, y) = 1_{y \notin g_k(x)} + \lambda \, \mathsf{cost}(|g_k(x)|)$. Minimizing (2) is an instance-dependent cost-sensitive learning problem. However, directly minimizing this target loss is intractable. To optimize this loss function, we introduce two families of surrogate losses in the next sections: cost-sensitive comp-sum losses and cost-sensitive constrained losses. Note that throughout this paper, we will denote all target

(or true) losses on which performance is measured with an $\ell$, while surrogate losses introduced for ease of optimization are denoted by $\widetilde{\ell}$.

## 2.2 Cost-sensitive comp-sum surrogate losses

Our surrogate cost-sensitive comp-sum, *c-comp*, losses are defined as follows: for all $(r, x, y) \in \mathcal{R} \times \mathcal{X} \times \mathcal{Y}$, $\widetilde{\ell}_{\mathrm{c-comp}}(r, x, y) = \sum_{k \in \mathcal{K}} (1 - c(x, k, y)) \widetilde{\ell}_{\mathrm{comp}}(r, x, k)$, where the comp-sum loss $\widetilde{\ell}_{\mathrm{comp}}$ is defined as in [Mao, Mohri, and Zhong, 2023f]. That is, for any $r$ in a hypothesis set $\mathcal{R}$ and $(x, y) \in \mathcal{X} \times \mathcal{Y}$, $\widetilde{\ell}_{\mathrm{comp}}(r, x, y) = \Phi\left(\sum_{y' \neq y} e^{r(x, y') - r(x, y)}\right)$, where $\Phi \colon \mathbb{R}_+ \to \mathbb{R}_+$ is non-decreasing. See Section 4.2 for more details. For example, when the logistic loss is used, we obtain the cost-sensitive logistic loss:

$$\widetilde{\ell}_{\mathrm{c-log}}(r, x, y) = \sum_{k \in \mathcal{K}} (1 - c(x, k, y)) \widetilde{\ell}_{\mathrm{log}}(r, x, k) = \sum_{k \in \mathcal{K}} (c(x, k, y) - 1) \left[ -\log\left( \sum_{k' \in \mathcal{K}} e^{r(x, k') - r(x, k)} \right) \right].$$

The negative log-term becomes larger as the score $r(x, k)$ increases. Thus, the loss function imposes a greater penalty on higher scores $r(x, k)$ through a penalty term $(c(x, k, y) - 1)$ that depends on the cost assigned to the expert's prediction $\mathsf{g}_k(x)$.

## 2.3 Cost-sensitive constrained surrogate losses

Constrained losses are defined as a summation of a function $\Phi$ applied to the scores, subject to a constraint, as in [Lee et al., 2004]. For any $r \in \mathcal{R}$ and $(x, y) \in \mathcal{X} \times \mathcal{Y}$, they are expressed as

$$\widetilde{\ell}_{\mathrm{cstnd}}(h, x, y) = \sum_{y' \neq y} \Phi(-r(x, y')), \text{ with the constraint } \sum_{y \in \mathcal{Y}} r(x, y) = 0,$$

where $\Phi \colon \mathbb{R} \to \mathbb{R}_+$ is non-increasing. See Section 4.2 for a detailed discussion. Inspired by these constrained losses, we introduce a new family of surrogate losses, *cost-sensitive constrained* (*c-cstnd* losses) which are defined, for all $(r, x, y) \in \mathcal{R} \times \mathcal{X} \times \mathcal{Y}$, by $\widetilde{\ell}_{\mathrm{c-cstnd}}(r, x, y) = \sum_{k \in \mathcal{K}} c(x, k, y) \Phi(-r(x, k))$, with the constraint $\sum_{k \in \mathcal{K}} r(x, k) = 0$, where $\Phi \colon \mathbb{R} \to \mathbb{R}_+$ is non-increasing. For example, for $\Phi(t) = e^{-t}$, we obtain the cost-sensitive constrained exponential loss:

$$\widetilde{\ell}_{\mathrm{c-exp}}^{\mathrm{cstnd}}(r, x, y) = \sum_{k \in \mathcal{K}} c(x, k, y) e^{r(x, k)}, \text{ with the constraint } \sum_{k \in \mathcal{K}} r(x, k) = 0.$$

# 3 Cardinality-aware algorithms

Minimizing the cost-sensitive surrogate loss functions described in the previous section directly leads to novel cardinality-aware algorithms. In this section, we briefly detail the instantiation of our algorithms in the specific cases of top-$k$ classifiers (our main focus) and threshold-based classifiers.

**Top-$k$ classifiers.** Here, the collection of set predictors is a subset of the top-$k$ classifiers, defined by $\mathsf{g}_k(x) = \{\mathsf{h}_1(x), \ldots, \mathsf{h}_k(x)\}$, where $\mathsf{h}_1(x), \ldots, \mathsf{h}_k(x)$ are the induced top-$k$ labels for a classifier $h$. The cardinality in this case coincides with the index: $|\mathsf{g}_k(x)| = k$, for any $x \in \mathcal{X}$. The cost is defined as $c(x, k, y) = 1_{y \notin \{\mathsf{h}_1(x), \ldots, \mathsf{h}_k(x)\}} + \lambda \mathsf{cost}(k)$, where $\mathsf{cost}(k)$ can be chosen to be $k$ or $\log(k)$. Thus, our cardinality-aware algorithms for top-$k$ classification can be described as follows. At training time, we assume access to a sample set $\{(x_i, y_i)\}_{i=1}^{m}$ and the costs each top-$k$ set incurs, $\{c(x_i, k, y_i)\}_{i=1}^{m}$, where $k \in \mathcal{K}$, a pre-fixed subset. The goal is to minimize the target cardinality-aware loss function $\sum_{i=1}^{m} \ell(r, x_i, y_i) = \sum_{i=1}^{m} c(x_i, \mathsf{r}(x_i), y_i)$ over a hypothesis set $\mathcal{R}$. Our algorithm consists of minimizing a surrogate loss such as the cost-sensitive logistic loss, defined as $\hat{r} = \operatorname{argmin}_{r \in \mathcal{R}} \sum_{i=1}^{m} \sum_{k \in \mathcal{K}} (1 - c(x_i, k, y_i)) \log\left( \sum_{k' \in \mathcal{K}} e^{r(x, k') - r(x, k)} \right)$. At inference time, we use the top-$\hat{\mathsf{r}}(x)$ set $\{\mathsf{h}_1(x), \ldots, \mathsf{h}_{\hat{\mathsf{r}}(x)}(x)\}$ for prediction, with the accuracy $1_{y \in \{\mathsf{h}_1(x), \ldots, \mathsf{h}_{\hat{\mathsf{r}}(x)}(x)\}}$ and cardinality $\hat{\mathsf{r}}(x)$ for that instance.

In Section 5, we compare the accuracy-versus-cardinality curves of our cardinality-aware algorithms obtained by varying $\lambda$ with those of top-$k$ classifiers, demonstrating the effectiveness of our algorithms. What $\lambda$ to select for a given application will depend on the desired accuracy. Note that the performance of the algorithm in [Denis and Hebiri, 2017] in this setting is theoretically the same as that of top-$k$

classifiers. The algorithm is designed to maximize accuracy within a constrained cardinality of $k$, and it always reaches maximal accuracy at the boundary $K$ after the cardinality is constrained to $k \leq K$.

**Threshold-based classifiers.** Here, the set predictor is defined via a set of thresholds $\tau_k$: $\mathsf{g}_k(x) = \{y \in \mathcal{Y}: s(x,y) > \tau_k\}$. When the set is empty, we just return $\operatorname{argmax}_{y \in \mathcal{Y}} s(x,y)$ by default. The description of the costs and other components of the algorithms is similar to that of top-$k$ classifiers. A special case of threshold-based classifier is conformal prediction [Shafer and Vovk, 2008], which is a general framework that provides provably valid confidence intervals for a black-box scoring function. Split conformal prediction guarantees that $\mathbb{P}(Y_{m+1} \in C_{s,\alpha}(X_{m+1})) \geq 1 - \alpha$ for some scoring function $s: \mathcal{X} \times \mathcal{Y} \to \mathbb{R}$, where $C_{s,\alpha}(X_{m+1}) = \{y: s(X_{m+1}, y) \geq \hat{q}_\alpha\}$ and $\hat{q}_\alpha$ is the $\lceil \alpha(m+1) \rceil / m$ empirical quantile of $s(X_i, Y_i)$ over a held-out set $\{(X_i, Y_i)\}_{i=1}^m$ drawn i.i.d. from some distribution $\mathcal{D}$ (or just exchangeably). Note, however, that the framework does not supply an effective guarantee on the size of the sets $C_{s,\alpha}(X_{m+1})$.

In Appendix K, we present in detail a series of early experiments for our algorithm used with threshold-based classifiers and include more discussion. Our experiments suggest that, when the training sample is sufficiently large, our algorithm can outperform conformal prediction.

## 4 Theoretical guarantees

Here, we present theory for our cardinality-aware algorithms. Our analysis builds on theory of top-$k$ algorithms, and we start by providing stronger results than previously known for top-$k$ surrogates.

### 4.1 Preliminaries

We denote by $\mathcal{D}$ a distribution over $\mathcal{X} \times \mathcal{Y}$ and write $p(x,y) = \mathcal{D}(Y = y \mid X = x)$ for the conditional probability of $Y = y$ given $X = x$, and use $p(x) = (p(x,1), \ldots, p(x,n))$ to denote the corresponding conditional probability vector. We denote by $\ell: \mathcal{H}_{\text{all}} \times \mathcal{X} \times \mathcal{Y} \to \mathbb{R}$ a loss function defined for the family of all measurable functions $\mathcal{H}_{\text{all}}$. Given a hypothesis set $\mathcal{H} \subseteq \mathcal{H}_{\text{all}}$, the conditional error of a hypothesis $h$ and the best-in-class conditional error are defined as follows: $\mathcal{C}_\ell(h,x) = \mathbb{E}_{y|x}[\ell(h,x,y)] = \sum_{y \in \mathcal{Y}} p(x,y)\ell(h,x,y)$ and $\mathcal{C}_\ell^*(\mathcal{H},x) = \inf_{h \in \mathcal{H}} \mathcal{C}_\ell(h,x)$. Accordingly, the generalization error of a hypothesis $h$ and the best-in-class generalization error are defined by: $\mathcal{E}_\ell(h) = \mathbb{E}_{(x,y) \sim \mathcal{D}}[\ell(h,x,y)] = \mathbb{E}_x[\mathcal{C}_\ell(h,x)]$ and $\mathcal{E}_\ell^*(\mathcal{H}) = \inf_{h \in \mathcal{H}} \mathcal{E}_\ell(h) = \inf_{h \in \mathcal{H}} \mathbb{E}_x[\mathcal{C}_\ell(h,x)]$. Given a score vector $(h(x,1), \ldots, h(x,n))$ generated by hypothesis $h$, we sort its components in decreasing order and write $\mathsf{h}_k(x)$ to denote the $k$-th label, that is $h(x, \mathsf{h}_1(x)) \geq h(x, \mathsf{h}_2(x)) \geq \ldots \geq h(x, \mathsf{h}_n(x))$. Similarly, for a given conditional probability vector $p(x) = (p(x,1), \ldots, p(x,n))$, we write $\mathsf{p}_k(x)$ to denote the $k$-th element in decreasing order, that is $p(x, \mathsf{p}_1(x)) \geq p(x, \mathsf{p}_2(x)) \geq \ldots \geq p(x, \mathsf{p}_n(x))$. In the event of a tie for the $k$-th highest score or conditional probability, the label $\mathsf{h}_k(x)$ or $\mathsf{p}_k(x)$ is selected based on the highest index when considering the natural order of labels.

The target generalization error for top-$k$ classification is given by the top-$k$ loss, which is denoted by $\ell_k$ and defined, for any hypothesis $h$ and $(x,y) \in \mathcal{X} \times \mathcal{Y}$ by

$$\ell_k(h,x,y) = 1_{y \notin \{\mathsf{h}_1(x), \ldots, \mathsf{h}_k(x)\}}.$$

The loss takes value one when the correct label $y$ is not included in the top-$k$ predictions made by the hypothesis $h$, zero otherwise. In the special case where $k = 1$, this is precisely the familiar zero-one classification loss. Like the zero-one loss, optimizing the top-$k$ loss is NP-hard for common hypothesis sets. Therefore, alternative surrogate losses are typically used to design learning algorithms. A crucial property of these surrogate losses is *Bayes-consistency*. This requires that, asymptotically, nearly minimizing a surrogate loss over the family of all measurable functions leads to the near minimization of the top-$k$ loss over the same family [Steinwart, 2007].

**Definition 4.1.** A surrogate loss $\widetilde{\ell}$ is said to be *Bayes-consistent with respect to the top-k loss $\ell_k$* if, for all given sequences of hypotheses $\{h_n\}_{n \in \mathbb{N}} \subset \mathcal{H}_{\text{all}}$ and any distribution, $\lim_{n \to +\infty} \mathcal{E}_{\widetilde{\ell}}(h_n) - \mathcal{E}_{\widetilde{\ell}}^*(\mathcal{H}_{\text{all}}) = 0$ implies $\lim_{n \to +\infty} \mathcal{E}_{\ell_k}(h_n) - \mathcal{E}_{\ell_k}^*(\mathcal{H}_{\text{all}}) = 0$.

Bayes-consistency is an asymptotic guarantee and applies only to the family of all measurable functions. Recently, Awasthi, Mao, Mohri, and Zhong [2022a,b] (see also [Awasthi et al., 2021a,b, 2023a,b, Mao et al., 2023c,d,e,a, 2024c,b,a,e,h,i,d,f,g, Mohri et al., 2024]) proposed a stronger consistency guarantee, referred to as $\mathcal{H}$-*consistency bounds*. These are upper bounds on the target

estimation error in terms of the surrogate estimation error that are non-asymptotic and hypothesis set-specific.

**Definition 4.2.** Given a hypothesis set $\mathcal{H}$, a surrogate loss $\widetilde{\ell}$ is said to admit an $\mathcal{H}$-consistency bound with respect to the top-$k$ loss $\ell_k$ if, for some non-decreasing function $f$, the following inequality holds for all $h \in \mathcal{H}$ and for any distribution: $f\big(\mathcal{E}_{\ell_k}(h) - \mathcal{E}_{\ell_k}^*(\mathcal{H})\big) \leq \mathcal{E}_{\widetilde{\ell}}(h) - \mathcal{E}_{\widetilde{\ell}}^*(\mathcal{H})$.

We refer to $\mathcal{E}_{\ell_k}(h) - \mathcal{E}_{\ell_k}^*(\mathcal{H})$ as the target estimation error and $\mathcal{E}_{\widetilde{\ell}}(h) - \mathcal{E}_{\widetilde{\ell}}^*(\mathcal{H})$ as the surrogate estimation error. These bounds imply Bayes-consistency when $\mathcal{H} = \mathcal{H}_{\text{all}}$, by taking the limit.

A key quantity appearing in $\mathcal{H}$-consistency bounds is the *minimizability gap*, which measures the difference between the best-in-class generalization error and the expectation of the best-in-class conditional error, defined for a given hypothesis set $\mathcal{H}$ and a loss function $\ell$ by: $\mathcal{M}_\ell(\mathcal{H}) = \mathcal{E}_\ell^*(\mathcal{H}) - \mathbb{E}_x[\mathcal{C}_\ell^*(\mathcal{H}, x)]$. As shown by Mao, Mohri, and Zhong [2023f], the minimizability gap is non-negative and is upper bounded by the approximation error $\mathcal{A}_\ell(\mathcal{H}) = \mathcal{E}_\ell^*(\mathcal{H}) - \mathcal{E}_\ell^*(\mathcal{H}_{\text{all}})$: $0 \leq \mathcal{M}_\ell(\mathcal{H}) \leq \mathcal{A}_\ell(\mathcal{H})$. When $\mathcal{H} = \mathcal{H}_{\text{all}}$ or more generally $\mathcal{A}_\ell(\mathcal{H}) = 0$, the minimizability gap vanishes. However, in general, it is non-zero and provides a finer measure than the approximation error. Thus, $\mathcal{H}$-consistency bounds provide a stronger guarantee than the excess error bounds.

## 4.2 Theoretical guarantees for top-$k$ surrogate losses

We study the surrogate loss families of *comp-sum* losses and *constrained* losses in multi-class classification, which have been shown in the past to benefit from $\mathcal{H}$-consistency bounds with respect to the zero-one classification loss, that is $\ell_k$ with $k = 1$ [Awasthi et al., 2022b, Mao et al., 2023f] (see also [Zheng et al., 2023, Mao et al., 2023b]). We extend these results to top-$k$ classification and prove $\mathcal{H}$-consistency bounds for these loss functions with respect to $\ell_k$ for any $1 \leq k \leq n$.

Another commonly used family of surrogate losses in multi-class classification is the *max* losses, which are defined through a convex function, such as the hinge loss function applied to the margin [Crammer and Singer, 2001, Awasthi et al., 2022b]. However, as shown in [Awasthi et al., 2022b], no non-trivial $\mathcal{H}$-consistency guarantee holds for max losses with respect to $\ell_k$, even when $k = 1$.

We first characterize the best-in-class conditional error and the conditional regret of top-$k$ loss, which will be used in the analysis of $\mathcal{H}$-consistency bounds. We denote by $S^{[k]} = \{X \subset S \mid |X| = k\}$ the set of all $k$-subsets of a set $S$. We will study any hypothesis set that is regular.

**Definition 4.3.** Let $A(n, k)$ be the set of ordered $k$-tuples with distinct elements in $[n]$. We say that a hypothesis set $\mathcal{H}$ is *regular for top-$k$ classification*, if the top-$k$ predictions generated by the hypothesis set cover all possible outcomes: $\forall x \in \mathcal{X}, \{(\mathsf{h}_1(x), \ldots, \mathsf{h}_k(x)) : h \in \mathcal{H}\} = A(n, k)$.

Common hypothesis sets such as that of linear models or neural networks, or the family of all measurable functions, are all regular for top-$k$ classification.

**Lemma 4.4.** *Assume that $\mathcal{H}$ is regular. Then, for any $h \in \mathcal{H}$ and $x \in \mathcal{X}$, the best-in-class conditional error and the conditional regret of the top-$k$ loss can be expressed as follows:*

$$\mathcal{C}_{\ell_k}^*(\mathcal{H}, x) = 1 - \sum_{i=1}^{k} p(x, \mathsf{p}_i(x)) \quad \Delta\mathcal{C}_{\ell_k, \mathcal{H}}(h, x) = \sum_{i=1}^{k} [p(x, \mathsf{p}_i(x)) - p(x, \mathsf{h}_i(x))].$$

The proof is included in Appendix A. For $k = 1$, the result coincides with the known identities for standard multi-class classification with regular hypothesis sets [Awasthi et al., 2022b, Lemma 3].

As with [Awasthi et al., 2022b, Mao et al., 2023f], in the following sections, we will consider hypothesis sets that are symmetric and complete. This includes the class of linear models and neural networks typically used in practice, as well as the family of all measurable functions. We say that a hypothesis set $\mathcal{H}$ is *symmetric* if it is independent of the ordering of labels. That is, for all $y \in \mathcal{Y}$, the scoring function $x \mapsto h(x, y)$ belongs to some real-valued family of functions $\mathcal{F}$. We say that a hypothesis set is *complete* if, for all $(x, y) \in \mathcal{X} \times \mathcal{Y}$, the set of scores $h(x, y)$ can span over the real numbers, that is, $\{h(x, y) : h \in \mathcal{H}\} = \mathbb{R}$. Note that any symmetric and complete hypothesis set is regular for top-$k$ classification.

Next, we analyze the broad family of comp-sum losses, which includes the commonly used logistic loss (or cross-entropy loss used with the softmax activation) as a special case.

Comp-sum losses are defined as the composition of a function $\Phi$ with the sum exponential losses, as in [Mao et al., 2023f]. For any $h \in \mathcal{H}$ and $(x, y) \in \mathcal{X} \times \mathcal{Y}$, they are expressed as

$$\widetilde{\ell}_{\mathrm{comp}}(h, x, y) = \Phi\left(\sum_{y' \neq y} e^{h(x,y')-h(x,y)}\right),$$

where $\Phi \colon \mathbb{R}_+ \to \mathbb{R}_+$ is non-decreasing. When $\Phi$ is chosen as the function $t \mapsto \log(1 + t)$, $t \mapsto t$, $t \mapsto 1 - \frac{1}{1+t}$ and $t \mapsto \frac{1}{q}\left(1 - \left(\frac{1}{1+t}\right)^q\right)$, $q \in (0, 1)$, $\widetilde{\ell}_{\mathrm{comp}}(h, x, y)$ coincides with the most commonly used (multinomial) logistic loss, defined as $\widetilde{\ell}_{\log}(h, x, y) = \log\left(\sum_{y' \in \mathcal{Y}} e^{h(x,y')-h(x,y)}\right)$ [Verhulst, 1838, 1845, Berkson, 1944, 1951], the sum-exponential loss $\widetilde{\ell}_{\exp}(h, x, y) = \sum_{y' \neq y} e^{h(x,y')-h(x,y)}$ [Weston and Watkins, 1998, Awasthi et al., 2022b] which is widely used in multi-class boosting [Saberian and Vasconcelos, 2011, Mukherjee and Schapire, 2013, Kuznetsov et al., 2014], the mean absolute error loss $\widetilde{\ell}_{\mathrm{mae}}(h, x, y) = 1 - \left[\sum_{y' \in \mathcal{Y}} e^{h(x,y')-h(x,y)}\right]^{-1}$ known to be robust to label noise for training neural networks [Ghosh et al., 2017], and the generalized cross-entropy loss $\widetilde{\ell}_{\mathrm{gce}}(h, x, y) = \frac{1}{q}\left[1 - \left[\sum_{y' \in \mathcal{Y}} e^{h(x,y')-h(x,y)}\right]^{-q}\right]$, $q \in (0, 1)$, a generalization of the logistic loss and mean absolute error loss for learning deep neural networks with noisy labels [Zhang and Sabuncu, 2018], respectively. We specifically study these loss functions and show that they benefit from $\mathcal{H}$-consistency bounds with respect to the top-$k$ loss.

**Theorem 4.5.** *Assume that $\mathcal{H}$ is symmetric and complete. Then, for any $1 \leq k \leq n$, the following $\mathcal{H}$-consistency bound holds for the comp-sum loss:*

$$\mathcal{E}_{\ell_k}(h) - \mathcal{E}^*_{\ell_k}(\mathcal{H}) + \mathcal{M}_{\ell_k}(\mathcal{H}) \leq k\psi^{-1}\left(\mathcal{E}_{\widetilde{\ell}_{\mathrm{comp}}}(h) - \mathcal{E}^*_{\widetilde{\ell}_{\mathrm{comp}}}(\mathcal{H}) + \mathcal{M}_{\widetilde{\ell}_{\mathrm{comp}}}(\mathcal{H})\right),$$

*In the special case where $\mathcal{A}_{\widetilde{\ell}_{\mathrm{comp}}}(\mathcal{H}) = 0$, for any $1 \leq k \leq n$, the following upper bound holds:*

$$\mathcal{E}_{\ell_k}(h) - \mathcal{E}^*_{\ell_k}(\mathcal{H}) \leq k\psi^{-1}\left(\mathcal{E}_{\widetilde{\ell}_{\mathrm{comp}}}(h) - \mathcal{E}^*_{\widetilde{\ell}_{\mathrm{comp}}}(\mathcal{H})\right),$$

*where $\psi(t) = \frac{1-t}{2}\log(1-t) + \frac{1+t}{2}\log(1+t)$, $t \in [0, 1]$ when $\widetilde{\ell}_{\mathrm{comp}}$ is $\widetilde{\ell}_{\log}$; $\psi(t) = 1 - \sqrt{1 - t^2}$, $t \in [0, 1]$ when $\widetilde{\ell}_{\mathrm{comp}}$ is $\widetilde{\ell}_{\exp}$; $\psi(t) = t/n$ when $\widetilde{\ell}_{\mathrm{comp}}$ is $\widetilde{\ell}_{\mathrm{mae}}$; and $\psi(t) = \frac{1}{qn^q}\left[\left[\frac{(1+t)^{\frac{1}{1-q}} + (1-t)^{\frac{1}{1-q}}}{2}\right]^{1-q} - 1\right]$, for all $q \in (0, 1)$, $t \in [0, 1]$ when $\widetilde{\ell}_{\mathrm{comp}}$ is $\widetilde{\ell}_{\mathrm{gce}}$.*

The proof is included in Appendix B. The second part follows from the fact that when $\mathcal{A}_{\widetilde{\ell}_{\mathrm{comp}}}(\mathcal{H}) = 0$, the minimizability gap $\mathcal{M}_{\widetilde{\ell}_{\mathrm{comp}}}(\mathcal{H})$ vanishes. By taking the limit on both sides, Theorem 4.5 implies the $\mathcal{H}$-consistency and Bayes-consistency of comp-sum losses with respect to the top-$k$ loss. It further shows that, when the estimation error of $\widetilde{\ell}_{\mathrm{comp}}$ is reduced to $\epsilon > 0$, then the estimation error of $\ell_k$ is upper bounded by $k\psi^{-1}(\epsilon)$, which, for a sufficiently small $\epsilon$, is approximately $k\sqrt{2\epsilon}$ for $\widetilde{\ell}_{\log}$ and $\widetilde{\ell}_{\exp}$; $kn\epsilon$ for $\widetilde{\ell}_{\mathrm{mae}}$; and $k\sqrt{2n^q\epsilon}$ for $\widetilde{\ell}_{\mathrm{gce}}$. Note that different from the other loss functions, the bound for the mean absolute error loss is only linear. The downside of this more favorable linear rate is the dependency on the number of classes and the fact that the mean absolute error loss is harder to optimize [Zhang and Sabuncu, 2018]. The bound for the generalized cross-entropy loss depends on both the number of classes $n$ and the parameter $q$.

In the proof, we used the fact that the conditional regret of the top-$k$ loss is the sum of $k$ differences between two probabilities. We then upper bounded each difference with the conditional regret of the comp-sum loss, using a hypothesis based on the two probabilities. The final bound is derived by summing these differences. In Appendix G, we detail the technical challenges and the novelty.

The key quantities in our $\mathcal{H}$-consistency bounds are the minimizability gaps, which can be upper bounded by the approximation error, or more refined terms, depending on the magnitude of the parameter space, as discussed by Mao et al. [2023f]. As pointed out by these authors, these quantities, along with the functional form, can help compare different comp-sum loss functions. In Appendix C, we further discuss the important role of minimizability gaps under the realizability assumption, and the connection with some negative results of Yang and Koyejo [2020].

Constrained losses are defined as a summation of a function $\Phi$ applied to the scores, subject to a constraint, as shown in [Lee et al., 2004]. For any $h \in \mathcal{H}$ and $(x, y) \in \mathcal{X} \times \mathcal{Y}$, they are expressed as

$$\widetilde{\ell}_{\mathrm{cstnd}}(h, x, y) = \sum_{y' \neq y} \Phi(-h(x, y')), \text{ with the constraint } \sum_{y \in \mathcal{Y}} h(x, y) = 0,$$

where $\Phi: \mathbb{R} \to \mathbb{R}_+$ is non-increasing. In Appendix E, we study this family of loss functions and show that several benefit from $\mathcal{H}$-consistency bounds with respect to the top-$k$ loss. In Appendix H, we provide generalization bounds for the top-$k$ loss in terms of finite samples (Theorems H.1 and H.2).

### 4.3 Theoretical guarantees for cardinality-aware surrogate losses

The strong theoretical results of the previous sections establish the effectiveness of comp-sum and constrained losses as surrogate losses for the target top-$k$ loss for common hypothesis sets used in practice. Building on this foundation, we expand our analysis to their cost-sensitive variants in the study of cardinality-aware set prediction in Section 2. We derive $\mathcal{H}$-consistency bounds for these loss functions, thereby also establishing their Bayes-consistency. To do so, we characterize the conditional regret of the target cardinality-aware loss function in Lemma I.1, which can be found in Appendix I. For this analysis, we will assume, without loss of generality, that the cost $c(x, k, y)$ takes values in $[0, 1]$ for any $(x, k, y) \in \mathcal{X} \times \mathcal{K} \times \mathcal{Y}$, which can be achieved by normalizing the cost function.

We will use $\widetilde{\ell}_{\mathrm{c-log}}$, $\widetilde{\ell}_{\mathrm{c-exp}}$, $\widetilde{\ell}_{\mathrm{c-gce}}$ and $\widetilde{\ell}_{\mathrm{c-mae}}$ to denote the corresponding cost-sensitive counterparts for $\widetilde{\ell}_{\mathrm{log}}$, $\widetilde{\ell}_{\mathrm{exp}}$, $\widetilde{\ell}_{\mathrm{gce}}$ and $\widetilde{\ell}_{\mathrm{mae}}$, respectively. Next, we show that these cost-sensitive surrogate loss functions benefit from $\mathcal{H}$-consistency bounds with respect to the target loss $\ell$ given in (1).

**Theorem 4.6.** *Assume that $\mathcal{R}$ is symmetric and complete. Then, the following bound holds for the cost-sensitive comp-sum loss: for all $r \in \mathcal{R}$ and for any distribution,*

$$\mathcal{E}_\ell(r) - \mathcal{E}_\ell^*(\mathcal{R}) + \mathcal{M}_\ell(\mathcal{R}) \leq \gamma\Big(\mathcal{E}_{\widetilde{\ell}_{\mathrm{c-comp}}}(r) - \mathcal{E}_{\widetilde{\ell}_{\mathrm{c-comp}}}^*(\mathcal{R}) + \mathcal{M}_{\widetilde{\ell}_{\mathrm{c-comp}}}(\mathcal{R})\Big);$$

*When $\mathcal{R} = \mathcal{R}_{\mathrm{all}}$, the following holds: $\mathcal{E}_\ell(r) - \mathcal{E}_\ell^*(\mathcal{R}_{\mathrm{all}}) \leq \gamma\Big(\mathcal{E}_{\widetilde{\ell}_{\mathrm{c-comp}}}(r) - \mathcal{E}_{\widetilde{\ell}_{\mathrm{c-comp}}}^*(\mathcal{R}_{\mathrm{all}})\Big)$, where $\gamma(t) = 2\sqrt{t}$ when $\widetilde{\ell}_{\mathrm{c-comp}}$ is either $\widetilde{\ell}_{\mathrm{c-log}}$ or $\widetilde{\ell}_{\mathrm{c-exp}}$; $\gamma(t) = 2\sqrt{|\mathcal{K}|^q t}$ when $\widetilde{\ell}_{\mathrm{c-comp}}$ is $\widetilde{\ell}_{\mathrm{c-gce}}$; and $\gamma(t) = |\mathcal{K}| t$ when $\widetilde{\ell}_{\mathrm{c-comp}}$ is $\widetilde{\ell}_{\mathrm{c-mae}}$.*

The proof is included in Appendix I.1. The second part follows from the fact that when $\mathcal{R} = \mathcal{R}_{\mathrm{all}}$, all the minimizability gaps vanish. In particular, Theorem 4.6 implies the Bayes-consistency of cost-sensitive comp-sum losses. The bounds for cost-sensitive generalized cross-entropy and mean absolute error loss depend on the number of set predictors, making them less favorable when $|\mathcal{K}|$ is large. As pointed out earlier, while the cost-sensitive mean absolute error loss admits a linear rate, it is difficult to optimize even in the standard classification, as reported by Zhang and Sabuncu [2018].

In the proof, we represented the comp-sum loss as a function of the softmax and introduced a softmax-dependent function $\mathcal{S}_\mu$ to upper bound the conditional regret of the target cardinality-aware loss function by that of the cost-sensitive comp-sum loss. This technique is novel and differs from the approach used in the standard scenario (Section 4.2).

We will use $\widetilde{\ell}_{\mathrm{c-exp}}^{\mathrm{cstnd}}$, $\widetilde{\ell}_{\mathrm{c-sq-hinge}}$, $\widetilde{\ell}_{\mathrm{c-hinge}}$ and $\widetilde{\ell}_{\mathrm{c-\rho}}$ to denote the corresponding cost-sensitive counterparts for $\widetilde{\ell}_{\mathrm{exp}}^{\mathrm{cstnd}}$, $\widetilde{\ell}_{\mathrm{sq-hinge}}$, $\widetilde{\ell}_{\mathrm{hinge}}$ and $\widetilde{\ell}_\rho$, respectively. Next, we show that these cost-sensitive surrogate losses benefit from $\mathcal{H}$-consistency bounds with respect to the target loss $\ell$ given in (1).

**Theorem 4.7.** *Assume that $\mathcal{R}$ is symmetric and complete. Then, the following bound holds for the cost-sensitive constrained loss: for all $r \in \mathcal{R}$ and for any distribution,*

$$\mathcal{E}_\ell(r) - \mathcal{E}_\ell^*(\mathcal{R}) + \mathcal{M}_\ell(\mathcal{R}) \leq \gamma\Big(\mathcal{E}_{\widetilde{\ell}_{\mathrm{c-cstnd}}}(r) - \mathcal{E}_{\widetilde{\ell}_{\mathrm{c-cstnd}}}^*(\mathcal{R}) + \mathcal{M}_{\widetilde{\ell}_{\mathrm{c-cstnd}}}(\mathcal{R})\Big);$$

*When $\mathcal{R} = \mathcal{R}_{\mathrm{all}}$, the following holds: $\mathcal{E}_\ell(r) - \mathcal{E}_\ell^*(\mathcal{R}_{\mathrm{all}}) \leq \gamma\Big(\mathcal{E}_{\widetilde{\ell}_{\mathrm{c-cstnd}}}(r) - \mathcal{E}_{\widetilde{\ell}_{\mathrm{c-cstnd}}}^*(\mathcal{R}_{\mathrm{all}})\Big)$, where $\gamma(t) = 2\sqrt{t}$ when $\widetilde{\ell}_{\mathrm{c-cstnd}}$ is $\widetilde{\ell}_{\mathrm{c-exp}}^{\mathrm{cstnd}}$ or $\widetilde{\ell}_{\mathrm{c-sq-hinge}}$; $\gamma(t) = t$ when $\widetilde{\ell}_{\mathrm{c-cstnd}}$ is $\widetilde{\ell}_{\mathrm{c-hinge}}$ or $\widetilde{\ell}_{\mathrm{c-\rho}}$.*

The proof is included in Appendix I.2. The second part follows from the fact that when $\mathcal{R} = \mathcal{R}_{\mathrm{all}}$, all the minimizability gaps vanish. In particular, Theorem 4.7 implies the Bayes-consistency of cost-sensitive constrained losses. Note that while the constrained hinge loss and $\rho$-margin loss have a more favorable linear rate in the bound, their optimization may be more challenging compared to other smooth loss functions.

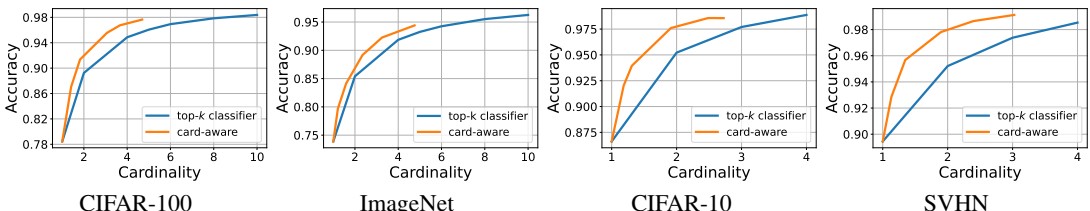

Figure 1: Accuracy versus average cardinality plots obtained by varying $\lambda$ for our cardinality-aware algorithm and top-$k$ classifiers across four datasets, with predictor set $\mathcal{K} = \{1, 2, 4, 8\}$ and cardinality cost $\mathsf{cost}(k) = \log k$. Our cardinality-aware algorithm consistently achieves higher accuracy for any fixed average cardinality across all datasets.

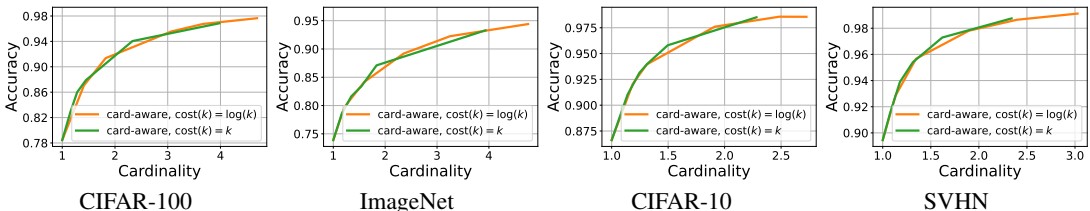

Figure 2: Comparison of cardinality costs $\mathsf{cost}(k) = \log k$ and $\mathsf{cost}(k) = k$, with predictor set $\mathcal{K} = \{1, 2, 4, 8\}$. The accuracy versus average cardinality plots for our cardinality-aware algorithm are similar, suggesting that the choice of cardinality cost has minimal impact on performance.

## 5 Experiments

Here, we report empirical results for our cardinality-aware algorithm and show that it consistently outperforms top-$k$ classifiers on benchmark datasets CIFAR-10, CIFAR-100 [Krizhevsky, 2009], SVHN [Netzer et al., 2011] and ImageNet [Deng et al., 2009].

We used the outputs of the second-to-last layer of ResNet [He et al., 2016] as features for the CIFAR-10, CIFAR-100 and SVHN datasets. For the ImageNet dataset, we used the CLIP [Radford et al., 2021] model to extract features. We adopted a linear model, trained using multinomial logistic loss, for the classifier $h$ on the extracted features from the datasets. We used a two-hidden-layer feedforward neural network with ReLU activation functions [Nair and Hinton, 2010] for the cardinality selector $r$. Both the classifier $h$ and the cardinality selector $r$ were trained using the Adam optimizer [Kingma and Ba, 2014], with a learning rate of $1 \times 10^{-3}$, a batch size of 128, and a weight decay of $1 \times 10^{-5}$.

Figure 1 compares the accuracy versus cardinality curves of the cardinality-aware algorithm with that of top-$k$ classifiers induced by $h$ for the various datasets. The accuracy of a top-$k$ classifier is measured by $\mathbb{E}_{(x,y)\sim S}\big[1 - \ell_k(h, x, y)\big]$, that is the fraction of the sample in which the top-$k$ predictions include the true label. It naturally grows as the cardinality $k$ increases, as shown in Figure 1. The accuracy of the carnality-aware algorithms is measured by $\mathbb{E}_{(x,y)\sim S}\big[1_{y \in \{\mathsf{h}_1(x),\ldots,\mathsf{h}_{r(x)}(x)\}}\big]$, that is the fraction of the sample in which the predictions selected by the model $r$ include the true label, and the corresponding cardinality is measured by $\mathbb{E}_{(x,y)\sim S}\big[\mathsf{r}(x)\big]$, that is the average size of the selected predictions. The cardinality selector $r$ was trained by minimizing the cost-sensitive logistic loss $\widetilde{\ell}_{\mathrm{c-log}}$ with the cost $c(x, k, y)$ defined as $\ell_k(h, x, y) + \lambda \log(k)$ and normalized to $[0, 1]$ through division by its maximum value over $\mathcal{X} \times \mathcal{K} \times \mathcal{Y}$. We allow for top-$k$ experts with $k \in \mathcal{K} = \{1, 2, 4, 8\}$ and vary $\lambda$. Starting from high values of $\lambda$, as $\lambda$ decreases in Figure 1, our cardinality-aware algorithm yields solutions with higher average cardinality and increased accuracy. This is because $\lambda$ controls the trade-off between cardinality and accuracy. The plots end to the right at $\lambda = 0.01$.

Figure 1 shows that the cardinality-aware algorithm is superior across the CIFAR-100, ImageNet, CIFAR-10 and SVHN datasets. For a given average cardinality, the cardinality-aware algorithm always achieves higher accuracy than a top-$k$ classifier. In other words, to achieve the same level of accuracy, the predictions made by the cardinality-aware algorithm can be significantly smaller in size compared to those made by the corresponding top-$k$ classifier. In particular, on the CIFAR-100, CIFAR-10 and SVHN datasets, the cardinality-aware algorithm achieves the same accuracy (98%) as the top-$k$ classifier while using roughly only half of the cardinality on average. As with the ImageNet dataset, it achieves the same accuracy (95%) as the top-$k$ classifier with only two-thirds of the cardinality. This illustrates the effectiveness of our cardinality-aware algorithm.

Figure 2 presents the comparison of $\mathsf{cost}(|\mathsf{g}_k(x)|) = k$ and $\mathsf{cost}(|\mathsf{g}_k(x)|) = \log k$ in the same setting (for each dataset, the orange curve in Figure 2 coincides with the orange curve in Figure 1). The

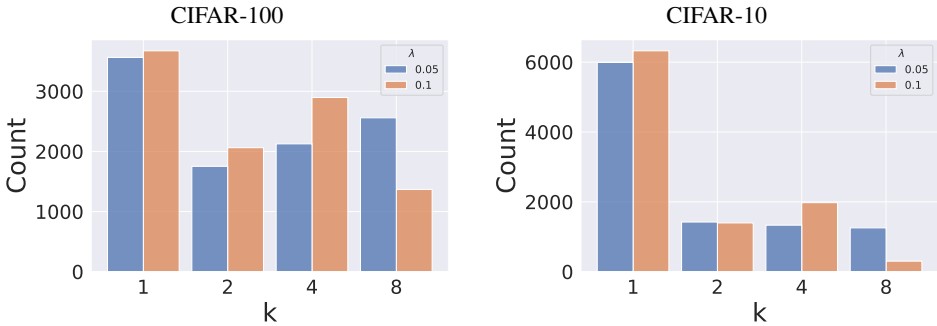

Figure 3: Cardinality distribution for top-$k$ experts with $\mathcal{K} = \{1, 2, 4, 8\}$ on CIFAR-10 and CIFAR-100 datasets, analyzed under two $\lambda$ values. For each dataset, increasing $\lambda$ reduces the number of samples with the highest cardinality ($k = 8$) and increases those with lower cardinalities, as higher $\lambda$ amplifies the influence of cardinality in the cost function. Across datasets, distributions vary for the same $\lambda$ due to differing task complexities.

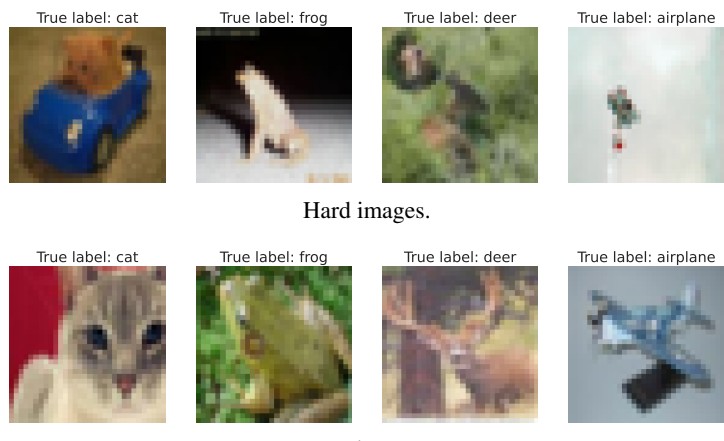

Figure 4: Illustration of hard and easy images on the CIFAR-10 dataset as judged by human annotators, for top-$k$ experts $\mathcal{K} = \{1, 2, 4, 8\}$. *Hard images* are those correctly predicted by our algorithm with a cardinality of $8$ but misclassified with a cardinality of $4$. *Easy images* are correctly predicted with a cardinality of $1$.

comparison suggests that the choice between the linear and logarithmic cardinality costs has negligible impact on our algorithm's performance, highlighting its robustness in this regard. We also empirically demonstrate that our algorithm dynamically adjusts the cardinality of prediction sets based on input complexity, selecting larger sets for more challenging inputs to ensure high accuracy and smaller sets for simpler inputs to keep the cardinality low, as illustrated in Figure 3 and Figure 4. We present additional experimental results with different choices of set $\mathcal{K}$ in Figure 5 and Figure 6 in Appendix J. Our cardinality-aware algorithm consistently outperforms top-$k$ classifiers across all configurations.

## 6 Conclusion

We introduced a new cardinality-aware set prediction framework for which we proposed two families of surrogate losses with strong $\mathcal{H}$-consistency guarantees: cost-sensitive comp-sum and constrained losses. This leads to principled and practical cardinality-aware algorithms for top-$k$ classification, which we showed empirically to be very effective. Additionally, we established a theoretical foundation for top-$k$ classification with fixed cardinality $k$ by proving that several common surrogate loss functions, including comp-sum losses and constrained losses in standard classification, admit $\mathcal{H}$-consistency bounds with respect to the top-$k$ loss. This provides a theoretical justification for the use of these loss functions in top-$k$ classification and opens new avenues for further research in this area.

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

# Contents of Appendix

# A Proof of Lemma 4.4

**Lemma 4.4.** *Assume that $\mathcal{H}$ is regular. Then, for any $h \in \mathcal{H}$ and $x \in \mathcal{X}$, the best-in-class conditional error and the conditional regret of the top-$k$ loss can be expressed as follows:*

$$\mathcal{C}^*_{\ell_k}(\mathcal{H}, x) = 1 - \sum_{i=1}^{k} p(x, \mathsf{p}_i(x)) \quad \Delta\mathcal{C}_{\ell_k, \mathcal{H}}(h, x) = \sum_{i=1}^{k} [p(x, \mathsf{p}_i(x)) - p(x, \mathsf{h}_i(x))].$$

*Proof.* By definition, for any $h \in \mathcal{H}$ and $x \in \mathcal{X}$, the conditional error of top-$k$ loss can be written as

$$\mathcal{C}_{\ell_k}(h, x) = \sum_{y \in \mathcal{Y}} p(x, y) 1_{y \notin \{\mathsf{h}_1(x), \ldots, \mathsf{h}_k(x)\}} = 1 - \sum_{i=1}^{k} p(x, \mathsf{h}_i(x)).$$

By definition of the labels $\mathsf{p}_i(x)$, which are the most likely top-$k$ labels, $\mathcal{C}_{\ell_k}(h, x)$ is minimized for $\mathsf{h}_i(x) = k_{\min}(x)$, $i \in [k]$. Since $\mathcal{H}$ is regular, this choice is realizable for some $h \in \mathcal{H}$. Thus, we have

$$\mathcal{C}^*_{\ell_k}(\mathcal{H}, x) = \inf_{h \in \mathcal{H}} \mathcal{C}_{\ell_k}(h, x) = 1 - \sum_{i=1}^{k} p(x, \mathsf{p}_i(x)).$$

Furthermore, the calibration gap can be expressed as

$$\Delta\mathcal{C}_{\ell_k, \mathcal{H}}(h, x) = \mathcal{C}_{\ell_k}(h, x) - \mathcal{C}^*_{\ell_k}(\mathcal{H}, x) = \sum_{i=1}^{k} (p(x, \mathsf{p}_i(x)) - p(x, \mathsf{h}_i(x))),$$

which completes the proof. $\qquad\square$

# B Proofs of $\mathcal{H}$-consistency bounds for comp-sum losses

**Theorem 4.5.** *Assume that $\mathcal{H}$ is symmetric and complete. Then, for any $1 \le k \le n$, the following $\mathcal{H}$-consistency bound holds for the comp-sum loss:*

$$\mathcal{E}_{\ell_k}(h) - \mathcal{E}^*_{\ell_k}(\mathcal{H}) + \mathcal{M}_{\ell_k}(\mathcal{H}) \le k\psi^{-1}\Big(\mathcal{E}_{\widetilde{\ell}_{\mathrm{comp}}}(h) - \mathcal{E}^*_{\widetilde{\ell}_{\mathrm{comp}}}(\mathcal{H}) + \mathcal{M}_{\widetilde{\ell}_{\mathrm{comp}}}(\mathcal{H})\Big),$$

*In the special case where $\mathcal{A}_{\widetilde{\ell}_{\mathrm{comp}}}(\mathcal{H}) = 0$, for any $1 \le k \le n$, the following upper bound holds:*

$$\mathcal{E}_{\ell_k}(h) - \mathcal{E}^*_{\ell_k}(\mathcal{H}) \le k\psi^{-1}\Big(\mathcal{E}_{\widetilde{\ell}_{\mathrm{comp}}}(h) - \mathcal{E}^*_{\widetilde{\ell}_{\mathrm{comp}}}(\mathcal{H})\Big),$$

*where $\psi(t) = \frac{1-t}{2} \log(1-t) + \frac{1+t}{2} \log(1+t)$, $t \in [0, 1]$ when $\widetilde{\ell}_{\mathrm{comp}}$ is $\widetilde{\ell}_{\log}$; $\psi(t) = 1 - \sqrt{1 - t^2}$, $t \in [0, 1]$ when $\widetilde{\ell}_{\mathrm{comp}}$ is $\widetilde{\ell}_{\exp}$; $\psi(t) = t/n$ when $\widetilde{\ell}_{\mathrm{comp}}$ is $\widetilde{\ell}_{\mathrm{mae}}$; and $\psi(t) = \frac{1}{qn^q}\left[\left[\frac{(1+t)^{\frac{1}{1-q}} + (1-t)^{\frac{1}{1-q}}}{2}\right]^{1-q} - 1\right]$, for all $q \in (0, 1)$, $t \in [0, 1]$ when $\widetilde{\ell}_{\mathrm{comp}}$ is $\widetilde{\ell}_{\mathrm{gce}}$.*

*Proof.* **Case I:** $\widetilde{\ell}_{\mathrm{comp}} = \widetilde{\ell}_{\log}$. For logistic loss $\widetilde{\ell}_{\log}$, the conditional regret can be written as

$$\Delta\mathcal{C}_{\widetilde{\ell}_{\log}, \mathcal{H}}(h, x) = \sum_{y=1}^{n} p(x, y)\widetilde{\ell}_{\log}(h, x, y) - \inf_{h \in \mathcal{H}} \sum_{y=1}^{n} p(x, y)\widetilde{\ell}_{\log}(h, x, y)$$

$$\ge \sum_{y=1}^{n} p(x, y)\widetilde{\ell}_{\log}(h, x, y) - \inf_{\mu \in \mathbb{R}} \sum_{y=1}^{n} p(x, y)\widetilde{\ell}_{\log}(h_{\mu,i}, x, y),$$

where for any $i \in [k]$,

$$h_{\mu,i}(x, y) = \begin{cases} h(x, y), & y \notin \{\mathsf{p}_i(x), \mathsf{h}_i(x)\} \\ \log\big(e^{h(x, \mathsf{p}_i(x))} + \mu\big) & y = \mathsf{h}_i(x) \\ \log\big(e^{h(x, \mathsf{h}_i(x))} - \mu\big) & y = \mathsf{p}_i(x). \end{cases}$$

Note that such a choice of $h_{\mu,i}$ leads to the following equality holds:

$$\sum_{y \notin \{\mathsf{h}_i(x), \mathsf{p}_i(x)\}} p(x, y)\widetilde{\ell}_{\log}(h, x, y) = \sum_{y \notin \{\mathsf{h}_i(x), \mathsf{p}_i(x)\}} p(x, y)\widetilde{\ell}_{\log}(h_{\mu,i}, x, y).$$

Therefore, for any $i \in [k]$, the conditional regret of logistic loss can be lower bounded as

$$\Delta \mathcal{C}_{\widetilde{\ell}_{\log}, \mathcal{H}}(h, x) \geq -p(x, \mathsf{h}_i(x)) \log\left( \frac{e^{h(x, \mathsf{h}_i(x))}}{\sum_{y \in \mathcal{Y}} e^{h(x, y)}} \right) - p(x, \mathsf{p}_i(x)) \log\left( \frac{e^{h(x, \mathsf{p}_i(x))}}{\sum_{y \in \mathcal{Y}} e^{h(x, y)}} \right)$$

$$+ \sup_{\mu \in \mathbb{R}} \left( p(x, \mathsf{h}_i(x)) \log\left( \frac{e^{h(x, \mathsf{p}_i(x))} + \mu}{\sum_{y \in \mathcal{Y}} e^{h(x, y)}} \right) + p(x, \mathsf{p}_i(x)) \log\left( \frac{e^{h(x, \mathsf{h}_i(x))} - \mu}{\sum_{y \in \mathcal{Y}} e^{h(x, y)}} \right) \right)$$

$$= \sup_{\mu \in \mathbb{R}} \left( p(x, \mathsf{h}_i(x)) \log\left( \frac{e^{h(x, \mathsf{p}_i(x))} + \mu}{e^{h(x, \mathsf{h}_i(x))}} \right) + p(x, \mathsf{p}_i(x)) \log\left( \frac{e^{h(x, \mathsf{h}_i(x))} - \mu}{e^{h(x, \mathsf{p}_i(x))}} \right) \right).$$

By the concavity of the function, differentiate with respect to $\mu$, we obtain that the supremum is achieved by $\mu^* = \frac{p(x, \mathsf{h}_i(x)) e^{h(x, \mathsf{h}_i(x))} - p(x, \mathsf{p}_i(x)) e^{h(x, \mathsf{p}_i(x))}}{p(x, \mathsf{h}_i(x)) + p(x, \mathsf{p}_i(x))}$. Plug in $\mu^*$, we obtain

$$\Delta \mathcal{C}_{\widetilde{\ell}_{\log}, \mathcal{H}}(h, x)$$

$$\geq p(x, \mathsf{h}_i(x)) \log\left( \frac{p(x, \mathsf{h}_i(x))}{p(x, \mathsf{h}_i(x)) + p(x, \mathsf{p}_i(x))} \frac{e^{h(x, \mathsf{h}_i(x))} + e^{h(x, \mathsf{p}_i(x))}}{e^{h(x, \mathsf{h}_i(x))}} \right)$$

$$+ p(x, \mathsf{p}_i(x)) \log\left( \frac{p(x, \mathsf{p}_i(x))}{p(x, \mathsf{h}_i(x)) + p(x, \mathsf{p}_i(x))} \frac{e^{h(x, \mathsf{h}_i(x))} + e^{h(x, \mathsf{p}_i(x))}}{e^{h(x, \mathsf{p}_i(x))}} \right)$$

$$\geq p(x, \mathsf{h}_i(x)) \log\left( \frac{2 p(x, \mathsf{h}_i(x))}{p(x, \mathsf{h}_i(x)) + p(x, \mathsf{p}_i(x))} \right) + p(x, \mathsf{p}_i(x)) \log\left( \frac{2 p(x, \mathsf{p}_i(x))}{p(x, \mathsf{h}_i(x)) + p(x, \mathsf{p}_i(x))} \right).$$
$$\text{(minimum is achieved when } h(x, \mathsf{h}_i(x)) = h(x, \mathsf{p}_i(x)))$$

let $S_i = p(x, \mathsf{p}_i(x)) + p(x, \mathsf{h}_i(x))$ and $\Delta_i = p(x, \mathsf{p}_i(x)) - p(x, \mathsf{h}_i(x))$, we have

$$\Delta \mathcal{C}_{\widetilde{\ell}_{\log}, \mathcal{H}}(h, x) \geq \frac{S_i - \Delta_i}{2} \log\left( \frac{S_i - \Delta_i}{S_i} \right) + \frac{S_i + \Delta_i}{2} \log\left( \frac{S_i + \Delta_i}{S_i} \right)$$

$$\geq \frac{1 - \Delta_i}{2} \log(1 - \Delta_i) + \frac{1 + \Delta_i}{2} \log(1 + \Delta_i) \quad \text{(minimum is achieved when } S_i = 1)$$

$$= \psi(p(x, \mathsf{p}_i(x)) - p(x, \mathsf{h}_i(x))),$$

where $\psi(t) = \frac{1-t}{2} \log(1 - t) + \frac{1+t}{2} \log(1 + t)$, $t \in [0, 1]$. Therefore, the conditional regret of the top-$k$ loss can be upper bounded as follows:

$$\Delta \mathcal{C}_{\ell_k, \mathcal{H}}(h, x) = \sum_{i=1}^{k} (p(x, \mathsf{p}_i(x)) - p(x, \mathsf{h}_i(x))) \leq k \psi^{-1}\left( \Delta \mathcal{C}_{\widetilde{\ell}_{\log}, \mathcal{H}}(h, x) \right).$$

By the concavity of $\psi^{-1}$, taking expectations on both sides of the preceding equation, we obtain

$$\mathcal{E}_{\ell_k}(h) - \mathcal{E}_{\ell_k}^*(\mathcal{H}) + \mathcal{M}_{\ell_k}(\mathcal{H}) \leq k \psi^{-1}\left( \mathcal{E}_{\widetilde{\ell}_{\log}}(h) - \mathcal{E}_{\widetilde{\ell}_{\log}}^*(\mathcal{H}) + \mathcal{M}_{\widetilde{\ell}_{\log}}(\mathcal{H}) \right).$$

The second part follows from the fact that when $\mathcal{A}_{\widetilde{\ell}_{\log}}(\mathcal{H}) = 0$, the minimizability gap $\mathcal{M}_{\widetilde{\ell}_{\log}}(\mathcal{H})$ vanishes.

**Case II:** $\widetilde{\ell}_{\mathrm{comp}} = \widetilde{\ell}_{\exp}$. For sum exponential loss $\widetilde{\ell}_{\exp}$, the conditional regret can be written as

$$\Delta \mathcal{C}_{\widetilde{\ell}_{\exp}, \mathcal{H}}(h, x) = \sum_{y=1}^{n} p(x, y) \widetilde{\ell}_{\exp}(h, x, y) - \inf_{h \in \mathcal{H}} \sum_{y=1}^{n} p(x, y) \widetilde{\ell}_{\exp}(h, x, y)$$

$$\geq \sum_{y=1}^{n} p(x, y) \widetilde{\ell}_{\exp}(h, x, y) - \inf_{\mu \in \mathbb{R}} \sum_{y=1}^{n} p(x, y) \widetilde{\ell}_{\exp}(h_{\mu, i}, x, y),$$

where for any $i \in [k]$,

$$h_{\mu, i}(x, y) = \begin{cases} h(x, y), & y \notin \{\mathsf{p}_i(x), \mathsf{h}_i(x)\} \\ \log\left( e^{h(x, \mathsf{p}_i(x))} + \mu \right) & y = \mathsf{h}_i(x) \\ \log\left( e^{h(x, \mathsf{h}_i(x))} - \mu \right) & y = \mathsf{p}_i(x). \end{cases}$$

Note that such a choice of $h_{\mu, i}$ leads to the following equality holds:

$$\sum_{y \notin \{\mathsf{h}_i(x), \mathsf{p}_i(x)\}} p(x, y) \widetilde{\ell}_{\exp}(h, x, y) = \sum_{y \notin \{\mathsf{h}_i(x), \mathsf{p}_i(x)\}} p(x, y) \widetilde{\ell}_{\exp}(h_{\mu, i}, x, y).$$

Therefore, for any $i \in [k]$, the conditional regret of sum exponential loss can be lower bounded as

$$\Delta \mathcal{C}_{\widetilde{\ell}_{\exp},\mathcal{H}}(h,x) \geq \sum_{y' \in \mathcal{Y}} \exp(h(x,y')) \left[ \frac{p(x,\mathsf{h}_i(x))}{\exp(h(x,\mathsf{h}_i(x)))} + \frac{p(x,\mathsf{p}_i(x))}{\exp(h(x,\mathsf{p}_i(x)))} \right]$$

$$+ \sup_{\mu \in \mathbb{R}} \left( - \sum_{y' \in \mathcal{Y}} \exp(h(x,y')) \left[ \frac{p(x,\mathsf{h}_i(x))}{\exp(h(x,\mathsf{p}_i(x))) + \mu} + \frac{p(x,\mathsf{p}_i(x))}{\exp(h(x,\mathsf{h}_i(x))) - \mu} \right] \right).$$

By the concavity of the function, differentiate with respect to $\mu$, we obtain that the supremum is achieved by $\mu^* = \frac{\exp[h(x,\mathsf{h}_i(x))]\sqrt{p(x,\mathsf{h}_i(x))} - \exp[h(x,\mathsf{p}_i(x))]\sqrt{p(x,\mathsf{p}_i(x))}}{\sqrt{p(x,\mathsf{h}_i(x))} + \sqrt{p(x,\mathsf{p}_i(x))}}$. Plug in $\mu^*$, we obtain

$$\Delta \mathcal{C}_{\widetilde{\ell}_{\exp},\mathcal{H}}(h,x)$$
$$\geq \sum_{y' \in \mathcal{Y}} \exp(h(x,y'))$$

$$\left[ \frac{p(x,\mathsf{h}_i(x))}{\exp(h(x,\mathsf{h}_i(x)))} + \frac{p(x,\mathsf{p}_i(x))}{\exp(h(x,\mathsf{p}_i(x)))} - \frac{\left(\sqrt{p(x,\mathsf{h}_i(x))} + \sqrt{p(x,\mathsf{p}_i(x))}\right)^2}{\exp(h(x,\mathsf{p}_i(x))) + \exp(h(x,\mathsf{h}_i(x)))} \right]$$

$$\geq \left[ 1 + \frac{\exp(h(x,\mathsf{p}_i(x)))}{\exp(h(x,\mathsf{h}_i(x)))} \right] p(x,\mathsf{h}_i(x))$$

$$+ \left[ 1 + \frac{\exp(h(x,\mathsf{h}_i(x)))}{\exp(h(x,\mathsf{p}_i(x)))} \right] p(x,\mathsf{p}_i(x)) - \left( \sqrt{p(x,\mathsf{h}_i(x))} + \sqrt{p(x,\mathsf{p}_i(x))} \right)^2$$

$$\left( \sum_{y' \in \mathcal{Y}} \exp(h(x,y')) \geq \exp(h(x,\mathsf{p}_i(x))) + \exp(h(x,\mathsf{h}_i(x))) \right)$$

$$\geq 2p(x,\mathsf{h}_i(x)) + 2p(x,\mathsf{p}_i(x)) - \left( \sqrt{p(x,\mathsf{h}_i(x))} + \sqrt{p(x,\mathsf{p}_i(x))} \right)^2.$$

$$\text{(minimum is attained when } \tfrac{\exp(h(x,\mathsf{p}_i(x)))}{\exp(h(x,\mathsf{h}_i(x)))} = 1)$$

let $S_i = p(x,\mathsf{p}_i(x)) + p(x,\mathsf{h}_i(x))$ and $\Delta_i = p(x,\mathsf{p}_i(x)) - p(x,\mathsf{h}_i(x))$, we have

$$\Delta \mathcal{C}_{\widetilde{\ell}_{\exp},\mathcal{H}}(h,x) \geq 2S_i - \left( \sqrt{\frac{S_i + \Delta_i}{2}} + \sqrt{\frac{S_i - \Delta_i}{2}} \right)^2$$

$$\geq 2 \left[ 1 - \left[ \frac{(1+\Delta_i)^{\frac{1}{2}} + (1-\Delta_i)^{\frac{1}{2}}}{2} \right]^2 \right] \qquad \text{(minimum is achieved when } S_i = 1)$$

$$= 1 - \sqrt{1 - (\Delta_i)^2}$$
$$= \psi(p(x,\mathsf{p}_i(x)) - p(x,\mathsf{h}_i(x))),$$

where $\psi(t) = 1 - \sqrt{1 - t^2}$, $t \in [0,1]$. Therefore, the conditional regret of the top-$k$ loss can be upper bounded as follows:

$$\Delta \mathcal{C}_{\ell_k,\mathcal{H}}(h,x) = \sum_{i=1}^{k} (p(x,\mathsf{p}_i(x)) - p(x,\mathsf{h}_i(x))) \leq k\psi^{-1}\left( \Delta \mathcal{C}_{\widetilde{\ell}_{\exp},\mathcal{H}}(h,x) \right).$$

By the concavity of $\psi^{-1}$, taking expectations on both sides of the preceding equation, we obtain

$$\mathcal{E}_{\ell_k}(h) - \mathcal{E}_{\ell_k}^*(\mathcal{H}) + \mathcal{M}_{\ell_k}(\mathcal{H}) \leq k\psi^{-1}\left( \mathcal{E}_{\widetilde{\ell}_{\exp}}(h) - \mathcal{E}_{\widetilde{\ell}_{\exp}}^*(\mathcal{H}) + \mathcal{M}_{\widetilde{\ell}_{\exp}}(\mathcal{H}) \right).$$

The second part follows from the fact that when $\mathcal{A}_{\widetilde{\ell}_{\exp}}(\mathcal{H}) = 0$, the minimizability gap $\mathcal{M}_{\widetilde{\ell}_{\exp}}(\mathcal{H})$ vanishes.

**Case III:** $\widetilde{\ell}_{\mathrm{comp}} = \widetilde{\ell}_{\mathrm{mae}}$. For mean absolute error loss $\widetilde{\ell}_{\mathrm{mae}}$, the conditional regret can be written as

$$\Delta \mathcal{C}_{\widetilde{\ell}_{\mathrm{mae}},\mathcal{H}}(h,x) = \sum_{y=1}^{n} p(x,y)\widetilde{\ell}_{\mathrm{mae}}(h,x,y) - \inf_{h \in \mathcal{H}} \sum_{y=1}^{n} p(x,y)\widetilde{\ell}_{\mathrm{mae}}(h,x,y)$$

$$\geq \sum_{y=1}^{n} p(x,y)\widetilde{\ell}_{\mathrm{mae}}(h,x,y) - \inf_{\mu \in \mathbb{R}} \sum_{y=1}^{n} p(x,y)\widetilde{\ell}_{\mathrm{mae}}(h_{\mu,i},x,y),$$

where for any $i \in [k]$,

$$h_{\mu,i}(x,y) = \begin{cases} h(x,y), & y \notin \{\mathsf{p}_i(x), \mathsf{h}_i(x)\} \\ \log\big(e^{h(x,\mathsf{p}_i(x))} + \mu\big) & y = \mathsf{h}_i(x) \\ \log\big(e^{h(x,\mathsf{h}_i(x))} - \mu\big) & y = \mathsf{p}_i(x). \end{cases}$$

Note that such a choice of $h_{\mu,i}$ leads to the following equality holds:

$$\sum_{y \notin \{\mathsf{h}_i(x), \mathsf{p}_i(x)\}} p(x,y)\widetilde{\ell}_{\mathrm{mae}}(h,x,y) = \sum_{y \notin \{\mathsf{h}_i(x), \mathsf{p}_i(x)\}} p(x,y)\widetilde{\ell}_{\mathrm{mae}}(h_{\mu,i},x,y).$$

Therefore, for any $i \in [k]$, the conditional regret of mean absolute error loss can be lower bounded as

$$\Delta \mathcal{C}_{\widetilde{\ell}_{\mathrm{mae}},\mathcal{H}}(h,x)$$

$$\geq p(x,\mathsf{h}_i(x))\left(1 - \frac{\exp(h(x,\mathsf{h}_i(x)))}{\sum_{y' \in \mathcal{Y}} \exp(h(x,y'))}\right) + p(x,\mathsf{p}_i(x))\left(1 - \frac{\exp(h(x,\mathsf{p}_i(x)))}{\sum_{y' \in \mathcal{Y}} \exp(h(x,y'))}\right)$$

$$+ \sup_{\mu \in \mathbb{R}}\left(-p(x,\mathsf{p}_i(x))\left(1 - \frac{\exp(h(x,\mathsf{h}_i(x))) - \mu}{\sum_{y' \in \mathcal{Y}} \exp(h(x,y'))}\right) - p(x,\mathsf{h}_i(x))\left(1 - \frac{\exp(h(x,\mathsf{p}_i(x))) + \mu}{\sum_{y' \in \mathcal{Y}} \exp(h(x,y'))}\right)\right).$$

By the concavity of the function, differentiate with respect to $\mu$, we obtain that the supremum is achieved by $\mu^* = -\exp[h(x,\mathsf{p}_i(x))]$. Plug in $\mu^*$, we obtain

$$\Delta \mathcal{C}_{\widetilde{\ell}_{\mathrm{mae}},\mathcal{H}}(h,x)$$

$$\geq p(x,\mathsf{p}_i(x))\frac{\exp(h(x,\mathsf{h}_i(x)))}{\sum_{y' \in \mathcal{Y}} \exp(h(x,y'))} - p(x,\mathsf{h}_i(x))\frac{\exp(h(x,\mathsf{h}_i(x)))}{\sum_{y' \in \mathcal{Y}} \exp(h(x,y'))}$$

$$\geq \frac{1}{n}(p(x,\mathsf{p}_i(x)) - p(x,\mathsf{h}_i(x))) \qquad\qquad \left(\frac{\exp(h(x,\mathsf{h}_i(x)))}{\sum_{y' \in \mathcal{Y}} \exp(h(x,y'))} \geq \frac{1}{n}\right)$$

Therefore, the conditional regret of the top-$k$ loss can be upper bounded as follows:

$$\Delta \mathcal{C}_{\ell_k,\mathcal{H}}(h,x) = \sum_{i=1}^{k}(p(x,\mathsf{p}_i(x)) - p(x,\mathsf{h}_i(x))) \leq kn\big(\Delta \mathcal{C}_{\widetilde{\ell}_{\mathrm{mae}},\mathcal{H}}(h,x)\big).$$

Take expectations on both sides of the preceding equation, we obtain

$$\mathcal{E}_{\ell_k}(h) - \mathcal{E}_{\ell_k}^*(\mathcal{H}) + \mathcal{M}_{\ell_k}(\mathcal{H}) \leq kn\big(\mathcal{E}_{\widetilde{\ell}_{\mathrm{mae}}}(h) - \mathcal{E}_{\widetilde{\ell}_{\mathrm{mae}}}^*(\mathcal{H}) + \mathcal{M}_{\widetilde{\ell}_{\mathrm{mae}}}(\mathcal{H})\big).$$

The second part follows from the fact that when $\mathcal{A}_{\widetilde{\ell}_{\mathrm{mae}}}(\mathcal{H}) = 0$, the minimizability gap $\mathcal{M}_{\widetilde{\ell}_{\mathrm{mae}}}(\mathcal{H})$ vanishes.

**Case IV:** $\widetilde{\ell}_{\mathrm{comp}} = \widetilde{\ell}_{\mathrm{gce}}$. For generalized cross-entropy loss $\widetilde{\ell}_{\mathrm{gce}}$, the conditional regret can be written as

$$\Delta \mathcal{C}_{\widetilde{\ell}_{\mathrm{gce}},\mathcal{H}}(h,x)$$

$$= \sum_{y=1}^{n} p(x,y)\widetilde{\ell}_{\mathrm{gce}}(h,x,y) - \inf_{h \in \mathcal{H}} \sum_{y=1}^{n} p(x,y)\widetilde{\ell}_{\mathrm{gce}}(h,x,y)$$

$$\geq \sum_{y=1}^{n} p(x,y)\widetilde{\ell}_{\mathrm{gce}}(h,x,y) - \inf_{\mu \in \mathbb{R}} \sum_{y=1}^{n} p(x,y)\widetilde{\ell}_{\mathrm{gce}}(h_{\mu,i},x,y),$$

where for any $i \in [k]$,

$$h_{\mu,i}(x,y) = \begin{cases} h(x,y), & y \notin \{\mathsf{p}_i(x), \mathsf{h}_i(x)\} \\ \log\big(e^{h(x,\mathsf{p}_i(x))} + \mu\big) & y = \mathsf{h}_i(x) \\ \log\big(e^{h(x,\mathsf{h}_i(x))} - \mu\big) & y = \mathsf{p}_i(x). \end{cases}$$

Note that such a choice of $h_{\mu,i}$ leads to the following equality holds:

$$\sum_{y \notin \{\mathsf{h}_i(x), \mathsf{p}_i(x)\}} p(x,y)\widetilde{\ell}_{\mathrm{gce}}(h,x,y) = \sum_{y \notin \{\mathsf{h}_i(x), \mathsf{p}_i(x)\}} p(x,y)\widetilde{\ell}_{\mathrm{gce}}(h_{\mu,i},x,y).$$

Therefore, for any $i \in [k]$, the conditional regret of generalized cross-entropy loss can be lower bounded as

$$q\Delta\mathcal{C}_{\widetilde{\ell}_{\text{gce}},\mathcal{H}}(h,x)$$

$$\geq p(x,\mathsf{h}_i(x))\left(1-\left[\frac{\exp(h(x,\mathsf{h}_i(x)))}{\sum_{y'\in\mathcal{Y}}\exp(h(x,y'))}\right]^q\right)+p(x,\mathsf{p}_i(x))\left(1-\left[\frac{\exp(h(x,\mathsf{p}_i(x)))}{\sum_{y'\in\mathcal{Y}}\exp(h(x,y'))}\right]^q\right)$$

$$+\sup_{\mu\in\mathbb{R}}\left(-p(x,\mathsf{h}_i(x))\left(1-\left[\frac{\exp(h(x,\mathsf{p}_i(x)))+\mu}{\sum_{y'\in\mathcal{Y}}\exp(h(x,y'))}\right]^q\right)-p(x,\mathsf{p}_i(x))\left(1-\left[\frac{\exp(h(x,\mathsf{h}_i(x)))-\mu}{\sum_{y'\in\mathcal{Y}}\exp(h(x,y'))}\right]^q\right)\right).$$

By the concavity of the function, differentiate with respect to $\mu$, we obtain that the supremum is achieved by $\mu^* = \frac{\exp[h(x,\mathsf{h}_i(x))]p(x,\mathsf{p}_i(x))^{\frac{1}{q-1}}-\exp[h(x,\mathsf{p}_i(x))]p(x,\mathsf{h}_i(x))^{\frac{1}{q-1}}}{p(x,\mathsf{h}_i(x))^{\frac{1}{q-1}}+p(x,\mathsf{p}_i(x))^{\frac{1}{q-1}}}$. Plug in $\mu^*$, we obtain

$$q\Delta\mathcal{C}_{\widetilde{\ell}_{\text{gce}},\mathcal{H}}(h,x)$$

$$\geq p(x,\mathsf{h}_i(x))\left[\frac{[\exp(h(x,\mathsf{h}_i(x)))+\exp(h(x,\mathsf{p}_i(x)))]p(x,\mathsf{p}_i(x))^{\frac{1}{q-1}}}{\sum_{y'\in\mathcal{Y}}\exp(h(x,y'))\left[p(x,\mathsf{h}_i(x))^{\frac{1}{q-1}}+p(x,\mathsf{p}_i(x))^{\frac{1}{q-1}}\right]}\right]^q$$

$$-p(x,\mathsf{h}_i(x))\left[\frac{\exp(h(x,\mathsf{h}_i(x)))}{\sum_{y'\in\mathcal{Y}}\exp(h(x,y'))}\right]^q$$

$$+p(x,\mathsf{p}_i(x))\left[\frac{[\exp(h(x,\mathsf{h}_i(x)))+\exp(h(x,\mathsf{p}_i(x)))]p(x,\mathsf{h}_i(x))^{\frac{1}{q-1}}}{\sum_{y'\in\mathcal{Y}}\exp(h(x,y'))\left[p(x,\mathsf{h}_i(x))^{\frac{1}{q-1}}+p(x,\mathsf{p}_i(x))^{\frac{1}{q-1}}\right]}\right]^q$$

$$-p(x,\mathsf{p}_i(x))\left[\frac{\exp(h(x,\mathsf{p}_i(x)))}{\sum_{y'\in\mathcal{Y}}\exp(h(x,y'))}\right]^q$$

$$\geq \frac{1}{n^q}\left(p(x,\mathsf{h}_i(x))\left[\frac{2p(x,\mathsf{p}_i(x))^{\frac{1}{q-1}}}{p(x,\mathsf{h}_i(x))^{\frac{1}{q-1}}+p(x,\mathsf{p}_i(x))^{\frac{1}{q-1}}}\right]^q-p(x,\mathsf{h}_i(x))\right)$$

$$+\frac{1}{n^q}\left(p(x,\mathsf{p}_i(x))\left[\frac{2p(x,\mathsf{h}_i(x))^{\frac{1}{q-1}}}{p(x,\mathsf{h}_i(x))^{\frac{1}{q-1}}+p(x,\mathsf{p}_i(x))^{\frac{1}{q-1}}}\right]^q-p(x,\mathsf{p}_i(x))\right)$$

$$\left(\left(\frac{\exp(h(x,\mathsf{p}_i(x)))}{\sum_{y'\in\mathcal{Y}}\exp(h(x,y'))}\right)^q\geq\frac{1}{n^q}\text{ and minimum is attained when }\frac{\exp(h(x,\mathsf{p}_i(x)))}{\exp(h(x,\mathsf{h}_i(x)))}=1\right)$$

let $S_i = p(x,\mathsf{p}_i(x)) + p(x,\mathsf{h}_i(x))$ and $\Delta_i = p(x,\mathsf{p}_i(x)) - p(x,\mathsf{h}_i(x))$, we have

$$\Delta\mathcal{C}_{\widetilde{\ell}_{\text{gce}},\mathcal{H}}(h,x)\geq\frac{1}{qn^q}\left(\left[\frac{(S_i+\Delta_i)^{\frac{1}{1-q}}+(S_i-\Delta_i)^{\frac{1}{1-q}}}{2}\right]^{1-q}-S_i\right)$$

$$\geq\frac{1}{qn^q}\left(\left[\frac{(1+\Delta_i)^{\frac{1}{1-q}}+(1-\Delta_i)^{\frac{1}{1-q}}}{2}\right]^{1-q}-1\right)$$

$$\text{(minimum is achieved when } S_i = 1)$$

$$=\psi(p(x,\mathsf{p}_i(x))-p(x,\mathsf{h}_i(x))),$$

where $\psi(t)=\frac{1}{qn^q}\left[\left[\frac{(1+t)^{\frac{1}{1-q}}+(1-t)^{\frac{1}{1-q}}}{2}\right]^{1-q}-1\right]$, $t\in[0,1]$. Therefore, the conditional regret of the top-$k$ loss can be upper bounded as follows:

$$\Delta\mathcal{C}_{\ell_k,\mathcal{H}}(h,x)=\sum_{i=1}^{k}(p(x,\mathsf{p}_i(x))-p(x,\mathsf{h}_i(x)))\leq k\psi^{-1}\left(\Delta\mathcal{C}_{\widetilde{\ell}_{\text{gce}},\mathcal{H}}(h,x)\right).$$

By the concavity of $\psi^{-1}$, taking expectations on both sides of the preceding equation, we obtain

$$\mathcal{E}_{\ell_k}(h)-\mathcal{E}_{\ell_k}^*(\mathcal{H})+\mathcal{M}_{\ell_k}(\mathcal{H})\leq k\psi^{-1}\left(\mathcal{E}_{\widetilde{\ell}_{\text{gce}}}(h)-\mathcal{E}_{\widetilde{\ell}_{\text{gce}}}^*(\mathcal{H})+\mathcal{M}_{\widetilde{\ell}_{\text{gce}}}(\mathcal{H})\right).$$

The second part follows from the fact that when $\mathcal{A}_{\widetilde{\ell}_{\mathrm{gce}}}(\mathcal{H}) = 0$, the minimizability gap $\mathcal{M}_{\widetilde{\ell}_{\mathrm{gce}}}(\mathcal{H})$ vanishes. $\qquad\square$

## C  Minimizability gaps and realizability

The key quantities in our $\mathcal{H}$-consistency bounds are the minimizability gaps, which can be upper bounded by the approximation error, or more refined terms, depending on the magnitude of the parameter space, as discussed by Mao et al. [2023f]. As pointed out by these authors, these quantities, along with the functional form, can help compare different comp-sum loss functions.

Here, we further discuss the important role of minimizability gaps under the realizability assumption, and the connection with some negative results of Yang and Koyejo [2020].

**Definition C.1** (**top-$k$-$\mathcal{H}$-realizability**)**.** A distribution $\mathcal{D}$ over $\mathcal{X} \times \mathcal{Y}$ is *top-$k$-$\mathcal{H}$-realizable*, if there exists a hypothesis $h \in \mathcal{H}$ such that $\mathbb{P}_{(x,y)\sim\mathcal{D}}(h(x,y) > h(x, \mathsf{h}_{k+1}(x))) = 1$.

This extends the $\mathcal{H}$-realizability definition from standard (top-1) classification [Long and Servedio, 2013] to top-$k$ classification for any $k \geq 1$.

**Definition C.2.** We say that a hypothesis set $\mathcal{H}$ *is closed under scaling*, if it is a cone, that is for all $h \in \mathcal{H}$ and $\beta \in \mathbb{R}_+$, $\beta h \in \mathcal{H}$.

**Definition C.3.** We say that a surrogate loss $\widetilde{\ell}$ is *realizable $\mathcal{H}$-consistent with respect to $\ell_k$*, if for all $k \in [1, n]$, and for any sequence of hypotheses $\{h_n\}_{n\in\mathbb{N}} \subset \mathcal{H}$ and top-$k$-$\mathcal{H}$-realizable distribution, $\lim_{n\to+\infty} \mathcal{E}_{\widetilde{\ell}}(h_n) - \mathcal{E}^*_{\widetilde{\ell}}(\mathcal{H}) = 0$ implies $\lim_{n\to+\infty} \mathcal{E}_{\ell_k}(h_n) - \mathcal{E}^*_{\ell_k}(\mathcal{H}) = 0$.

When $\mathcal{H}$ is closed under scaling, for $k = 1$ and all comp-sum loss functions $\ell = \widetilde{\ell}_{\log}, \widetilde{\ell}_{\exp}, \widetilde{\ell}_{\mathrm{gce}}$ and $\widetilde{\ell}_{\mathrm{mae}}$, it can be shown that $\mathcal{E}^*_{\widetilde{\ell}}(\mathcal{H}) = \mathcal{M}_{\widetilde{\ell}}(\mathcal{H}) = 0$ for any $\mathcal{H}$-realizable distribution. For example, for $\ell = \widetilde{\ell}_{\log}$, by using the Lebesgue dominated convergence theorem, we have

$$\mathcal{M}_{\widetilde{\ell}_{\log}}(\mathcal{H}) \leq \mathcal{E}^*_{\widetilde{\ell}_{\log}}(\mathcal{H}) \leq \lim_{\beta\to+\infty} \mathcal{E}_{\widetilde{\ell}_{\log}}(\beta h^*) = \lim_{\beta\to+\infty} \log\left[1 + \sum_{y'\neq y} e^{\beta(h^*(x,y') - h^*(x,y))}\right] = 0,$$

where $h^*$ satisfies $\mathbb{P}_{(x,y)\sim\mathcal{D}}(h^*(x,y) > h^*(x, \mathsf{h}_2(x))) = 1$ Therefore, Theorem 4.5 implies that all these loss functions are realizable $\mathcal{H}$-consistent with respect to $\ell_{0-1}$ ($\ell_k$ for $k = 1$) when $\mathcal{H}$ is closed under scaling.

**Theorem C.4.** *Assume that $\mathcal{H}$ is closed under scaling. Then, $\widetilde{\ell}_{\log}, \widetilde{\ell}_{\exp}, \widetilde{\ell}_{\mathrm{gce}}$ and $\widetilde{\ell}_{\mathrm{mae}}$ are realizable $\mathcal{H}$-consistent with respect to $\ell_{0-1}$.*

The formal proof is presented in Appendix D. However, for $k > 1$, since in the realizability assumption, $h(x,y)$ is only larger than $h(x, \mathsf{h}_{k+1}(x))$ and can be smaller than $h(x, \mathsf{h}_1(x))$, there may exist an $\mathcal{H}$-realizable distribution $\mathcal{D}$ such that $\mathcal{M}_{\widetilde{\ell}_{\log}}(\mathcal{H}) > 0$. This explains the inconsistency of the logistic loss on top-$k$ separable data with linear predictors, when $k = 2$ and $n > 2$, as shown in [Yang and Koyejo, 2020]. More generally, the exact same example in [Yang and Koyejo, 2020, Proposition 5.1] can be used to show that all the comp-sum losses, $\widetilde{\ell}_{\log}, \widetilde{\ell}_{\exp}, \widetilde{\ell}_{\mathrm{gce}}$ and $\widetilde{\ell}_{\mathrm{mae}}$ are not realizable $\mathcal{H}$-consistent with respect to $\ell_k$. Nevertheless, as previously shown, when the hypothesis set $\mathcal{H}$ adopted is sufficiently rich such that $\mathcal{M}_{\widetilde{\ell}}(\mathcal{H}) = 0$ or even $\mathcal{A}_{\widetilde{\ell}}(\mathcal{H}) = 0$, they are guaranteed to be $\mathcal{H}$-consistent. This is typically the case in practice when using deep neural networks.

## D  Proofs of realizable $\mathcal{H}$-consistency for comp-sum losses

**Theorem C.4.** *Assume that $\mathcal{H}$ is closed under scaling. Then, $\widetilde{\ell}_{\log}, \widetilde{\ell}_{\exp}, \widetilde{\ell}_{\mathrm{gce}}$ and $\widetilde{\ell}_{\mathrm{mae}}$ are realizable $\mathcal{H}$-consistent with respect to $\ell_{0-1}$.*

*Proof.* Since the distribution is realizable, there exists a hypothesis $h \in \mathcal{H}$ such that

$$\mathbb{P}_{(x,y)\sim\mathcal{D}}(h^*(x,y) > h^*(x, \mathsf{h}_2(x))) = 1.$$

Therefore, for the logistic loss, by using the Lebesgue dominated convergence theorem,

$$\mathcal{M}_{\widetilde{\ell}_{\log}}(\mathcal{H}) \leq \mathcal{E}^*_{\widetilde{\ell}_{\log}}(\mathcal{H}) \leq \lim_{\beta \to +\infty} \mathcal{E}_{\widetilde{\ell}_{\log}}(\beta h) = \lim_{\beta \to +\infty} \log\left[1 + \sum_{y' \neq y} e^{\beta(h^*(x,y') - h^*(x,y))}\right] = 0.$$

For the sum exponential loss, by using the Lebesgue dominated convergence theorem,

$$\mathcal{M}_{\widetilde{\ell}_{\exp}}(\mathcal{H}) \leq \mathcal{E}^*_{\widetilde{\ell}_{\exp}}(\mathcal{H}) \leq \lim_{\beta \to +\infty} \mathcal{E}_{\widetilde{\ell}_{\exp}}(\beta h) = \lim_{\beta \to +\infty} \sum_{y' \neq y} e^{\beta(h^*(x,y') - h^*(x,y))} = 0.$$

For the generalized cross entropy loss, by using the Lebesgue dominated convergence theorem,

$$\mathcal{M}_{\widetilde{\ell}_{\mathrm{gce}}}(\mathcal{H}) \leq \mathcal{E}^*_{\widetilde{\ell}_{\mathrm{gce}}}(\mathcal{H}) \leq \lim_{\beta \to +\infty} \mathcal{E}_{\widetilde{\ell}_{\mathrm{gce}}}(\beta h) = \lim_{\beta \to +\infty} \frac{1}{q}\left[1 - \left[\sum_{y' \in \mathcal{Y}} e^{\beta(h^*(x,y') - h^*(x,y))}\right]^{-q}\right] = 0.$$

For the mean absolute error loss, by using the Lebesgue dominated convergence theorem,

$$\mathcal{M}_{\widetilde{\ell}_{\mathrm{mae}}}(\mathcal{H}) \leq \mathcal{E}^*_{\widetilde{\ell}_{\mathrm{mae}}}(\mathcal{H}) \leq \lim_{\beta \to +\infty} \mathcal{E}_{\widetilde{\ell}_{\mathrm{mae}}}(\beta h) = \lim_{\beta \to +\infty} 1 - \left[\sum_{y' \in \mathcal{Y}} e^{\beta(h^*(x,y') - h^*(x,y))}\right]^{-1} = 0.$$

Therefore, by Theorem 4.5, the proof is completed. □

## E $\mathcal{H}$-Consistency bounds for constrained losses

Constrained losses are defined as a summation of a function $\Phi$ applied to the scores, subject to a constraint, as shown in [Lee et al., 2004, Awasthi et al., 2022b]. For any $h \in \mathcal{H}$ and $(x, y) \in \mathcal{X} \times \mathcal{Y}$, they are expressed as

$$\widetilde{\ell}_{\mathrm{cstnd}}(h, x, y) = \sum_{y' \neq y} \Phi(-h(x, y')),$$

with the constraint $\sum_{y \in \mathcal{Y}} h(x, y) = 0$, where $\Phi: \mathbb{R} \to \mathbb{R}_+$ is non-increasing. When $\Phi$ is chosen as the function $t \mapsto e^{-t}$, $t \mapsto \max\{0, 1 - t\}^2$, $t \mapsto \max\{0, 1 - t\}$ and $t \mapsto \min\{\max\{0, 1 - t/\rho\}, 1\}$, $\rho > 0$, $\widetilde{\ell}_{\mathrm{cstnd}}(h, x, y)$ are referred to as the constrained exponential loss $\widetilde{\ell}^{\mathrm{cstnd}}_{\exp}(h, x, y) = \sum_{y' \neq y} e^{h(x,y')}$, the constrained squared hinge loss $\widetilde{\ell}_{\mathrm{sq-hinge}}(h, x, y) = \sum_{y' \neq y} \max\{0, 1 + h(x, y')\}^2$, the constrained hinge loss $\widetilde{\ell}_{\mathrm{hinge}}(h, x, y) = \sum_{y' \neq y} \max\{0, 1 + h(x, y')\}$, and the constrained $\rho$-margin loss $\widetilde{\ell}_\rho(h, x, y) = \sum_{y' \neq y} \min\{\max\{0, 1 + h(x, y')/\rho\}, 1\}$, respectively [Awasthi et al., 2022b]. We now study these loss functions and show that they benefit from $\mathcal{H}$-consistency bounds with respect to the top-$k$ loss.

**Theorem E.1.** *Assume that $\mathcal{H}$ is symmetric and complete. Then, for any $1 \leq k \leq n$, the following $\mathcal{H}$-consistency bound holds for the constrained loss:*

$$\mathcal{E}_{\ell_k}(h) - \mathcal{E}^*_{\ell_k}(\mathcal{H}) + \mathcal{M}_{\ell_k}(\mathcal{H}) \leq k\gamma\Big(\mathcal{E}_{\widetilde{\ell}_{\mathrm{cstnd}}}(h) - \mathcal{E}^*_{\widetilde{\ell}_{\mathrm{cstnd}}}(\mathcal{H}) + \mathcal{M}_{\widetilde{\ell}_{\mathrm{cstnd}}}(\mathcal{H})\Big).$$

*In the special case where $\mathcal{A}_{\widetilde{\ell}_{\mathrm{cstnd}}}(\mathcal{H}) = 0$, for any $1 \leq k \leq n$, the following bound holds:*

$$\mathcal{E}_{\ell_k}(h) - \mathcal{E}^*_{\ell_k}(\mathcal{H}) \leq k\gamma\Big(\mathcal{E}_{\widetilde{\ell}_{\mathrm{cstnd}}}(h) - \mathcal{E}^*_{\widetilde{\ell}_{\mathrm{cstnd}}}(\mathcal{H})\Big),$$

*where $\gamma(t) = 2\sqrt{t}$ when $\widetilde{\ell}_{\mathrm{cstnd}}$ is either $\widetilde{\ell}^{\mathrm{cstnd}}_{\exp}$ or $\widetilde{\ell}_{\mathrm{sq-hinge}}$; $\gamma(t) = t$ when $\widetilde{\ell}_{\mathrm{cstnd}}$ is either $\widetilde{\ell}_{\mathrm{hinge}}$ or $\widetilde{\ell}_\rho$.*

The proof is included in Appendix F. The second part follows from the fact that when the hypothesis set $\mathcal{H}$ is sufficiently rich such that $\mathcal{A}_{\widetilde{\ell}_{\mathrm{cstnd}}}(\mathcal{H}) = 0$, we have $\mathcal{M}_{\widetilde{\ell}_{\mathrm{cstnd}}}(\mathcal{H}) = 0$. Therefore, the constrained loss is $\mathcal{H}$-consistent and Bayes-consistent with respect to $\ell_k$. If the surrogate estimation error $\mathcal{E}_{\widetilde{\ell}_{\mathrm{cstnd}}}(h) - \mathcal{E}^*_{\widetilde{\ell}_{\mathrm{cstnd}}}(\mathcal{H})$ is $\epsilon$, then, the target estimation error satisfies $\mathcal{E}_{\ell_k}(h) - \mathcal{E}^*_{\ell_k}(\mathcal{H}) \leq k\gamma(\epsilon)$. Note that the constrained exponential loss and the constrained squared hinge loss both admit a square root $\mathcal{H}$-consistency bound while the bounds for the constrained hinge loss and $\rho$-margin loss are both linear.

# F  Proofs of $\mathcal{H}$-consistency bounds for constrained losses

The conditional error for the constrained loss can be expressed as follows:

$$\mathcal{C}_{\widetilde{\ell}_{\mathrm{cstnd}}}(h,x) = \sum_{y=1}^{n} p(x,y)\widetilde{\ell}_{\mathrm{cstnd}}(h,x,y) = \sum_{y=1}^{n} p(x,y) \sum_{y'\neq y} \Phi(-h(x,y')) = \sum_{y\in\mathcal{Y}}(1 - p(x,y))\Phi(-h(x,y)).$$

**Theorem E.1.** *Assume that $\mathcal{H}$ is symmetric and complete. Then, for any $1 \le k \le n$, the following $\mathcal{H}$-consistency bound holds for the constrained loss:*

$$\mathcal{E}_{\ell_k}(h) - \mathcal{E}_{\ell_k}^*(\mathcal{H}) + \mathcal{M}_{\ell_k}(\mathcal{H}) \le k\gamma\Big(\mathcal{E}_{\widetilde{\ell}_{\mathrm{cstnd}}}(h) - \mathcal{E}_{\widetilde{\ell}_{\mathrm{cstnd}}}^*(\mathcal{H}) + \mathcal{M}_{\widetilde{\ell}_{\mathrm{cstnd}}}(\mathcal{H})\Big).$$

*In the special case where $\mathcal{A}_{\widetilde{\ell}_{\mathrm{cstnd}}}(\mathcal{H}) = 0$, for any $1 \le k \le n$, the following bound holds:*

$$\mathcal{E}_{\ell_k}(h) - \mathcal{E}_{\ell_k}^*(\mathcal{H}) \le k\gamma\Big(\mathcal{E}_{\widetilde{\ell}_{\mathrm{cstnd}}}(h) - \mathcal{E}_{\widetilde{\ell}_{\mathrm{cstnd}}}^*(\mathcal{H})\Big),$$

*where $\gamma(t) = 2\sqrt{t}$ when $\widetilde{\ell}_{\mathrm{cstnd}}$ is either $\widetilde{\ell}_{\exp}^{\mathrm{cstnd}}$ or $\widetilde{\ell}_{\mathrm{sq-hinge}}$; $\gamma(t) = t$ when $\widetilde{\ell}_{\mathrm{cstnd}}$ is either $\widetilde{\ell}_{\mathrm{hinge}}$ or $\widetilde{\ell}_{\rho}$.*

*Proof.* **Case I: $\widetilde{\ell}_{\mathrm{cstnd}} = \widetilde{\ell}_{\exp}^{\mathrm{cstnd}}$.** For the constrained exponential loss $\widetilde{\ell}_{\exp}^{\mathrm{cstnd}}$, the conditional regret can be written as

$$\begin{aligned}
\Delta\mathcal{C}_{\widetilde{\ell}_{\exp}^{\mathrm{cstnd}},\mathcal{H}}(h,x) &= \sum_{y=1}^{n} p(x,y)\widetilde{\ell}_{\exp}^{\mathrm{cstnd}}(h,x,y) - \inf_{h\in\mathcal{H}} \sum_{y=1}^{n} p(x,y)\widetilde{\ell}_{\exp}^{\mathrm{cstnd}}(h,x,y)\\
&\ge \sum_{y=1}^{n} p(x,y)\widetilde{\ell}_{\exp}^{\mathrm{cstnd}}(h,x,y) - \inf_{\mu\in\mathbb{R}} \sum_{y=1}^{n} p(x,y)\widetilde{\ell}_{\exp}^{\mathrm{cstnd}}(h_{\mu,i},x,y),
\end{aligned}$$

where for any $i \in [k]$,

$$h_{\mu,i}(x,y) = \begin{cases} h(x,y), & y \notin \{\mathsf{p}_i(x),\mathsf{h}_i(x)\}\\ h(x,\mathsf{p}_i(x)) + \mu & y = \mathsf{h}_i(x)\\ h(x,\mathsf{h}_i(x)) - \mu & y = \mathsf{p}_i(x). \end{cases}$$

Note that such a choice of $h_{\mu,i}$ leads to the following equality holds:

$$\sum_{y\notin\{\mathsf{h}_i(x),\mathsf{p}_i(x)\}} p(x,y)\widetilde{\ell}_{\exp}^{\mathrm{cstnd}}(h,x,y) = \sum_{y\notin\{\mathsf{h}_i(x),\mathsf{p}_i(x)\}} p(x,y)\widetilde{\ell}_{\exp}^{\mathrm{cstnd}}(h_{\mu,i},x,y).$$

Let $q(x,\mathsf{p}_i(x)) = 1 - p(x,\mathsf{p}_i(x))$ and $q(x,\mathsf{h}_i(x)) = 1 - p(x,\mathsf{h}_i(x))$. Therefore, for any $i \in [k]$, the conditional regret of constrained exponential loss can be lower bounded as

$$\begin{aligned}
&\Delta\mathcal{C}_{\widetilde{\ell}_{\exp}^{\mathrm{cstnd}},\mathcal{H}}(h,x)\\
&\ge \inf_{h\in\mathcal{H}}\sup_{\mu\in\mathbb{R}}\Big\{q(x,\mathsf{p}_i(x))\big(e^{h(x,\mathsf{p}_i(x))} - e^{h(x,\mathsf{h}_i(x))-\mu}\big) + q(x,\mathsf{h}_i(x))\big(e^{h(x,\mathsf{h}_i(x))} - e^{h(x,\mathsf{p}_i(x))+\mu}\big)\Big\}\\
&= \Big(\sqrt{q(x,\mathsf{p}_i(x))} - \sqrt{q(x,\mathsf{h}_i(x))}\Big)^2 \qquad\qquad \text{(differentiating with respect to $\mu$, $h$ to optimize)}\\
&= \left(\frac{q(x,\mathsf{h}_i(x)) - q(x,\mathsf{p}_i(x))}{\sqrt{q(x,\mathsf{p}_i(x))} + \sqrt{q(x,\mathsf{h}_i(x))}}\right)^2\\
&\ge \frac{1}{4}\big(q(x,\mathsf{h}_i(x)) - q(x,\mathsf{p}_i(x))\big)^2 \qquad\qquad (0 \le q(x,y) \le 1)\\
&= \frac{1}{4}\big(p(x,\mathsf{p}_i(x)) - p(x,\mathsf{h}_i(x))\big)^2.
\end{aligned}$$

Therefore, by Lemma 4.4, the conditional regret of the top-$k$ loss can be upper bounded as follows:

$$\Delta\mathcal{C}_{\ell_k,\mathcal{H}}(h,x) = \sum_{i=1}^{k}(p(x,\mathsf{p}_i(x)) - p(x,\mathsf{h}_i(x))) \le 2k\Big(\Delta\mathcal{C}_{\widetilde{\ell}_{\exp}^{\mathrm{cstnd}},\mathcal{H}}(h,x)\Big)^{\frac{1}{2}}.$$

By the concavity, taking expectations on both sides of the preceding equation, we obtain

$$\mathcal{E}_{\ell_k}(h) - \mathcal{E}^*_{\ell_k}(\mathcal{H}) + \mathcal{M}_{\ell_k}(\mathcal{H}) \le 2k\left(\mathcal{E}_{\widetilde{\ell}^{\mathrm{cstnd}}_{\exp}}(h) - \mathcal{E}^*_{\widetilde{\ell}^{\mathrm{cstnd}}_{\exp}}(\mathcal{H}) + \mathcal{M}_{\widetilde{\ell}^{\mathrm{cstnd}}_{\exp}}(\mathcal{H})\right)^{\frac{1}{2}}.$$

The second part follows from the fact that when $\mathcal{A}_{\widetilde{\ell}^{\mathrm{cstnd}}_{\exp}}(\mathcal{H}) = 0$, we have $\mathcal{M}_{\widetilde{\ell}^{\mathrm{cstnd}}_{\exp}}(\mathcal{H}) = 0$.

**Case II:** $\widetilde{\ell}_{\mathrm{cstnd}} = \widetilde{\ell}_{\mathrm{sq-hinge}}$**.** For the constrained squared hinge loss $\widetilde{\ell}_{\mathrm{sq-hinge}}$, the conditional regret can be written as

$$\Delta\mathcal{C}_{\widetilde{\ell}_{\mathrm{sq-hinge}},\mathcal{H}}(h,x) = \sum_{y=1}^{n} p(x,y)\widetilde{\ell}_{\mathrm{sq-hinge}}(h,x,y) - \inf_{h\in\mathcal{H}}\sum_{y=1}^{n} p(x,y)\widetilde{\ell}_{\mathrm{sq-hinge}}(h,x,y)$$

$$\ge \sum_{y=1}^{n} p(x,y)\widetilde{\ell}_{\mathrm{sq-hinge}}(h,x,y) - \inf_{\mu\in\mathbb{R}}\sum_{y=1}^{n} p(x,y)\widetilde{\ell}_{\mathrm{sq-hinge}}(h_{\mu,i},x,y),$$

where for any $i \in [k]$,

$$h_{\mu,i}(x,y) = \begin{cases} h(x,y), & y \notin \{\mathsf{p}_i(x),\mathsf{h}_i(x)\} \\ h(x,\mathsf{p}_i(x)) + \mu & y = \mathsf{h}_i(x) \\ h(x,\mathsf{h}_i(x)) - \mu & y = \mathsf{p}_i(x). \end{cases}$$

Note that such a choice of $h_{\mu,i}$ leads to the following equality holds:

$$\sum_{y\notin\{\mathsf{h}_i(x),\mathsf{p}_i(x)\}} p(x,y)\widetilde{\ell}_{\mathrm{sq-hinge}}(h,x,y) = \sum_{y\notin\{\mathsf{h}_i(x),\mathsf{p}_i(x)\}} p(x,y)\widetilde{\ell}_{\mathrm{sq-hinge}}(h_{\mu,i},x,y).$$

Let $q(x,\mathsf{p}_i(x)) = 1 - p(x,\mathsf{p}_i(x))$ and $q(x,\mathsf{h}_i(x)) = 1 - p(x,\mathsf{h}_i(x))$. Therefore, for any $i \in [k]$, the conditional regret of the constrained squared hinge loss can be lower bounded as

$$\Delta\mathcal{C}_{\widetilde{\ell}_{\mathrm{sq-hinge}},\mathcal{H}}(h,x)$$

$$\ge \inf_{h\in\mathcal{H}}\sup_{\mu\in\mathbb{R}}\left\{q(x,\mathsf{p}_i(x))\left(\max\{0,1+h(x,\mathsf{p}_i(x))\}^2 - \max\{0,1+h(x,\mathsf{h}_i(x))-\mu\}^2\right)\right.$$

$$\left. + q(x,\mathsf{h}_i(x))\left(\max\{0,1+h(x,\mathsf{h}_i(x))\}^2 - \max\{0,1+h(x,\mathsf{p}_i(x))+\mu\}^2\right)\right\}$$

$$\ge \frac{1}{4}(q(x,\mathsf{p}_i(x)) - q(x,\mathsf{h}_i(x)))^2 \qquad \text{(differentiating with respect to } \mu, h \text{ to optimize)}$$

$$= \frac{1}{4}(p(x,\mathsf{p}_i(x)) - p(x,\mathsf{h}_i(x)))^2$$

Therefore, by Lemma 4.4, the conditional regret of the top-$k$ loss can be upper bounded as follows:

$$\Delta\mathcal{C}_{\ell_k,\mathcal{H}}(h,x) = \sum_{i=1}^{k}(p(x,\mathsf{p}_i(x)) - p(x,\mathsf{h}_i(x))) \le 2k\left(\Delta\mathcal{C}_{\widetilde{\ell}_{\mathrm{sq-hinge}},\mathcal{H}}(h,x)\right)^{\frac{1}{2}}.$$

By the concavity, taking expectations on both sides of the preceding equation, we obtain

$$\mathcal{E}_{\ell_k}(h) - \mathcal{E}^*_{\ell_k}(\mathcal{H}) + \mathcal{M}_{\ell_k}(\mathcal{H}) \le 2k\left(\mathcal{E}_{\widetilde{\ell}_{\mathrm{sq-hinge}}}(h) - \mathcal{E}^*_{\widetilde{\ell}_{\mathrm{sq-hinge}}}(\mathcal{H}) + \mathcal{M}_{\widetilde{\ell}_{\mathrm{sq-hinge}}}(\mathcal{H})\right)^{\frac{1}{2}}.$$

The second part follows from the fact that when the hypothesis set $\mathcal{H}$ is sufficiently rich such that $\mathcal{A}_{\widetilde{\ell}_{\mathrm{sq-hinge}}}(\mathcal{H}) = 0$, we have $\mathcal{M}_{\widetilde{\ell}_{\mathrm{sq-hinge}}}(\mathcal{H}) = 0$.

**Case III:** $\widetilde{\ell}_{\mathrm{cstnd}} = \widetilde{\ell}_{\mathrm{hinge}}$**.** For the constrained hinge loss $\widetilde{\ell}_{\mathrm{hinge}}$, the conditional regret can be written as

$$\Delta\mathcal{C}_{\widetilde{\ell}_{\mathrm{hinge}},\mathcal{H}}(h,x) = \sum_{y=1}^{n} p(x,y)\widetilde{\ell}_{\mathrm{hinge}}(h,x,y) - \inf_{h\in\mathcal{H}}\sum_{y=1}^{n} p(x,y)\widetilde{\ell}_{\mathrm{hinge}}(h,x,y)$$

$$\ge \sum_{y=1}^{n} p(x,y)\widetilde{\ell}_{\mathrm{hinge}}(h,x,y) - \inf_{\mu\in\mathbb{R}}\sum_{y=1}^{n} p(x,y)\widetilde{\ell}_{\mathrm{hinge}}(h_{\mu,i},x,y),$$

where for any $i \in [k]$,

$$h_{\mu,i}(x,y) = \begin{cases} h(x,y), & y \notin \{\mathsf{p}_i(x), \mathsf{h}_i(x)\} \\ h(x,\mathsf{p}_i(x)) + \mu & y = \mathsf{h}_i(x) \\ h(x,\mathsf{h}_i(x)) - \mu & y = \mathsf{p}_i(x). \end{cases}$$

Note that such a choice of $h_{\mu,i}$ leads to the following equality holds:

$$\sum_{y \notin \{\mathsf{h}_i(x), \mathsf{p}_i(x)\}} p(x,y)\widetilde{\ell}_{\mathrm{hinge}}(h,x,y) = \sum_{y \notin \{\mathsf{h}_i(x), \mathsf{p}_i(x)\}} p(x,y)\widetilde{\ell}_{\mathrm{hinge}}(h_{\mu,i},x,y).$$

Let $q(x,\mathsf{p}_i(x)) = 1 - p(x,\mathsf{p}_i(x))$ and $q(x,\mathsf{h}_i(x)) = 1 - p(x,\mathsf{h}_i(x))$. Therefore, for any $i \in [k]$, the conditional regret of the constrained hinge loss can be lower-bounded as

$$\Delta\mathcal{C}_{\widetilde{\ell}_{\mathrm{hinge}},\mathcal{H}}(h,x) \geq \inf_{h \in \mathcal{H}} \sup_{\mu \in \mathbb{R}} \Big\{ q(x,\mathsf{p}_i(x))(\max\{0, 1 + h(x,\mathsf{p}_i(x))\} - \max\{0, 1 + h(x,\mathsf{h}_i(x)) - \mu\})$$

$$+ q(x,\mathsf{h}_i(x))(\max\{0, 1 + h(x,\mathsf{h}_i(x))\} - \max\{0, 1 + h(x,\mathsf{p}_i(x)) + \mu\}) \Big\}$$

$$\geq q(x,\mathsf{h}_i(x)) - q(x,\mathsf{p}_i(x)) \qquad \text{(differentiating with respect to } \mu, h \text{ to optimize)}$$

$$= p(x,\mathsf{p}_i(x)) - p(x,\mathsf{h}_i(x))$$

Therefore, by Lemma 4.4, the conditional regret of the top-$k$ loss can be upper bounded as follows:

$$\Delta\mathcal{C}_{\ell_k,\mathcal{H}}(h,x) = \sum_{i=1}^{k}(p(x,\mathsf{p}_i(x)) - p(x,\mathsf{h}_i(x))) \leq k\Delta\mathcal{C}_{\widetilde{\ell}_{\mathrm{hinge}},\mathcal{H}}(h,x).$$

By the concavity, taking expectations on both sides of the preceding equation, we obtain

$$\mathcal{E}_{\ell_k}(h) - \mathcal{E}_{\ell_k}^*(\mathcal{H}) + \mathcal{M}_{\ell_k}(\mathcal{H}) \leq k\Big(\mathcal{E}_{\widetilde{\ell}_{\mathrm{hinge}}}(h) - \mathcal{E}_{\widetilde{\ell}_{\mathrm{hinge}}}^*(\mathcal{H}) + \mathcal{M}_{\widetilde{\ell}_{\mathrm{hinge}}}(\mathcal{H})\Big).$$

The second part follows from the fact that when the hypothesis set $\mathcal{H}$ is sufficiently rich such that $\mathcal{A}_{\widetilde{\ell}_{\mathrm{hinge}}}(\mathcal{H}) = 0$, we have $\mathcal{M}_{\widetilde{\ell}_{\mathrm{hinge}}}(\mathcal{H}) = 0$.

**Case IV:** $\widetilde{\ell}_{\mathrm{cstnd}} = \widetilde{\ell}_\rho$. For the constrained $\rho$-margin loss $\widetilde{\ell}_\rho$, the conditional regret can be written as

$$\Delta\mathcal{C}_{\widetilde{\ell}_\rho,\mathcal{H}}(h,x) = \sum_{y=1}^{n} p(x,y)\widetilde{\ell}_\rho(h,x,y) - \inf_{h \in \mathcal{H}} \sum_{y=1}^{n} p(x,y)\widetilde{\ell}_\rho(h,x,y)$$

$$\geq \sum_{y=1}^{n} p(x,y)\widetilde{\ell}_\rho(h,x,y) - \inf_{\mu \in \mathbb{R}} \sum_{y=1}^{n} p(x,y)\widetilde{\ell}_\rho(h_{\mu,i},x,y),$$

where for any $i \in [k]$,

$$h_{\mu,i}(x,y) = \begin{cases} h(x,y), & y \notin \{\mathsf{p}_i(x), \mathsf{h}_i(x)\} \\ h(x,\mathsf{p}_i(x)) + \mu & y = \mathsf{h}_i(x) \\ h(x,\mathsf{h}_i(x)) - \mu & y = \mathsf{p}_i(x). \end{cases}$$

Note that such a choice of $h_{\mu,i}$ leads to the following equality holds:

$$\sum_{y \notin \{\mathsf{h}_i(x), \mathsf{p}_i(x)\}} p(x,y)\widetilde{\ell}_\rho(h,x,y) = \sum_{y \notin \{\mathsf{h}_i(x), \mathsf{p}_i(x)\}} p(x,y)\widetilde{\ell}_\rho(h_{\mu,i},x,y).$$

Let $q(x,\mathsf{p}_i(x)) = 1 - p(x,\mathsf{p}_i(x))$ and $q(x,\mathsf{h}_i(x)) = 1 - p(x,\mathsf{h}_i(x))$. Therefore, for any $i \in [k]$, the conditional regret of the constrained $\rho$-margin loss can be lower-bounded as

$$\Delta\mathcal{C}_{\widetilde{\ell}_\rho,\mathcal{H}}(h,x)$$

$$\geq \inf_{h \in \mathcal{H}} \sup_{\mu \in \mathbb{R}} \Big\{ q(x,\mathsf{p}_i(x))\Big(\min\Big\{\max\Big\{0, 1 + \frac{h(x,\mathsf{p}_i(x))}{\rho}\Big\}, 1\Big\} - \min\Big\{\max\Big\{0, 1 + \frac{h(x,\mathsf{h}_i(x)) - \mu}{\rho}\Big\}, 1\Big\}\Big)$$

$$+ q(x,\mathsf{h}_i(x))\Big(\min\Big\{\max\Big\{0, 1 + \frac{h(x,\mathsf{h}_i(x))}{\rho}\Big\}, 1\Big\} - \min\Big\{\max\Big\{0, 1 + \frac{h(x,\mathsf{p}_i(x)) + \mu}{\rho}\Big\}, 1\Big\}\Big) \Big\}$$

$$\geq q(x,\mathsf{h}_i(x)) - q(x,\mathsf{p}_i(x)) \qquad \text{(differentiating with respect to } \mu, h \text{ to optimize)}$$

$$= p(x,\mathsf{p}_i(x)) - p(x,\mathsf{h}_i(x))$$

Therefore, by Lemma 4.4, the conditional regret of the top-$k$ loss can be upper bounded as follows:

$$\Delta \mathcal{C}_{\ell_k, \mathcal{H}}(h, x) = \sum_{i=1}^{k} (p(x, \mathsf{p}_i(x)) - p(x, \mathsf{h}_i(x))) \le k \Delta \mathcal{C}_{\widetilde{\ell}_\rho, \mathcal{H}}(h, x).$$

By the concavity, taking expectations on both sides of the preceding equation, we obtain

$$\mathcal{E}_{\ell_k}(h) - \mathcal{E}^*_{\ell_k}(\mathcal{H}) + \mathcal{M}_{\ell_k}(\mathcal{H}) \le k \Big( \mathcal{E}_{\widetilde{\ell}_\rho}(h) - \mathcal{E}^*_{\widetilde{\ell}_\rho}(\mathcal{H}) + \mathcal{M}_{\widetilde{\ell}_\rho}(\mathcal{H}) \Big).$$

The second part follows from the fact that when the hypothesis set $\mathcal{H}$ is sufficiently rich such that $\mathcal{A}_{\widetilde{\ell}_\rho}(\mathcal{H}) = 0$, we have $\mathcal{M}_{\widetilde{\ell}_\rho}(\mathcal{H}) = 0$. $\qquad\square$

# G  Technical challenges and novelty in Section 4.2

The technical challenges and novelty of proofs in Section 4.2 lie in the following three aspects:

(1) Conditional regret of the top-$k$ loss: This involves a comprehensive analysis of the conditional regret associated with the top-$k$ loss, which is significantly more complex than that of the zero-one loss in a standard setting. The conditional regret of the top-$k$ loss incorporates both the top-$k$ conditional probabilities $\mathsf{p}_i(x)$, for $i = 1, \ldots, k$, and the top-$k$ scores $\mathsf{h}_i(x)$, for $i = 1, \ldots, k$, as characterized in Lemma 4.4.

(2) Relating to the conditional regret of the surrogate loss: To establish $\mathcal{H}$-consistency bounds, it is necessary to upper bound the conditional regret of the top-$k$ loss with that of the surrogate loss. This task is particularly challenging in the top-$k$ setting due to the intricate nature of the top-$k$ loss's conditional regret. A pivotal observation is that the conditional regret of the top-$k$ loss can be expressed as the sum of $k$ terms $(p(x, \mathsf{p}_i(x)) - p(x, \mathsf{h}_i(x)))$ for $i = 1, \ldots, k$. Each term $(p(x, \mathsf{p}_i(x)) - p(x, \mathsf{h}_i(x)))$ exhibits structural similarities to the conditional regret of the zero-one loss, $(p(x, \mathsf{p}_1(x)) - p(x, \mathsf{h}_1(x)))$. Consequently, we introduce a series of auxiliary hypotheses $h_{\mu,i}$, each dependent on $\mathsf{h}_i(x)$ and $\mathsf{p}_i(x)$ for $i \in [k]$. This approach transforms the challenge of upper bounding the conditional regret of the top-$k$ loss into $k$ subproblems, each focusing on upper bounding the term $(p(x, \mathsf{p}_i(x)) - p(x, \mathsf{h}_i(x)))$ with the conditional regret of the surrogate loss.

(3) Upper bounding each term $(p(x, \mathsf{p}_i(x)) - p(x, \mathsf{h}_i(x)))$: Following the approach in prior work [Mao et al., 2023f] for top-1 classification, we define $h_{\mu,i}(x, y)$ as:

$$h_{\mu,i}(x, y) = \begin{cases} h(x, y), & y \notin \{\mathsf{p}_i(x)), \mathsf{h}_i(x))\} \\ \log\!\big(e^{h(x, \mathsf{p}_i(x))} + \mu\big) & y = \mathsf{h}_i(x)) \\ \log\!\big(e^{h(x, \mathsf{h}_i(x)))} - \mu\big) & y = \mathsf{p}_i(x)). \end{cases}$$

for the proof of comp-sum losses (Theorem 4.5). The subsequent proof is considered straightforward.

However, for the proof of constrained losses (Theorem E.1), we adopt a different hypothesis formulation for $h_{\mu,i}(x, y)$, leveraging the constraint that the scores sum to zero and the specific structure of constrained losses. The hypothesis is defined as:

$$h_{\mu,i}(x, y) = \begin{cases} h(x, y), & y \notin \{\mathsf{p}_i(x)), \mathsf{h}_i(x))\} \\ h(x, \mathsf{p}_i(x)) + \mu & y = \mathsf{h}_i(x)) \\ h(x, \mathsf{h}_i(x))) - \mu & y = \mathsf{p}_i(x)). \end{cases}$$

The remainder of the proof then specifically addresses the peculiarities of constrained losses, which significantly diverges from the previous work.

In summary, aspects (1) and (2) are novel and represent significant advancements that have not been explored previously. For aspect (3), the proof for comp-sum loss closely follows the approach in [Mao et al., 2023f], which appears straightforward due to the innovative ideas presented in aspects (1) and (2). However, the proof for constrained losses significantly deviates from the previous work, particularly in terms of the new auxiliary hypothesis formulation and the specific constrained losses examined.

We would like to further emphasize that these results are significant and useful. They demonstrate that comp-sum losses, which include the cross-entropy loss commonly used in top-1 classification, and constrained losses, are $\mathcal{H}$-consistent in top-$k$ classification for any $k$. Notably, the cross-entropy loss is the only Bayes-consistent smooth surrogate loss for top-$k$ classification identified to date. Furthermore, the Bayes-consistency of loss functions within the constrained loss family is a novel exploration in the context of top-$k$ classification. These findings are pivotal as they highlight two broad families of smooth loss functions that are Bayes-consistent in top-$k$ classification. Additionally, they reveal that these families, including the cross-entropy loss, benefit from stronger, non-asymptotic and hypothesis set-specific guarantees—$\mathcal{H}$-consistency bounds—in top-$k$ classification.

# H Generalization bounds

Given a finite sample $S = ((x_1, y_1), \ldots, (x_m, y_m))$ drawn from $\mathcal{D}^m$, let $\widehat{h}_S$ be the minimizer of the empirical loss within $\mathcal{H}$ with respect to the top-$k$ surrogate loss $\widetilde{\ell}$: $\widehat{h}_S = \operatorname{argmin}_{h \in \mathcal{H}} \widehat{\mathcal{E}}_{\widetilde{\ell}, S}(h) = \operatorname{argmin}_{h \in \mathcal{H}} \frac{1}{m} \sum_{i=1}^{m} \widetilde{\ell}(h, x_i, y_i)$. Next, we will show that we can use $\mathcal{H}$-consistency bounds for $\widetilde{\ell}$ to derive generalization bounds for the top-$k$ loss by upper bounding the surrogate estimation error $\mathcal{E}_{\widetilde{\ell}}(\widehat{h}_S) - \mathcal{E}_{\widetilde{\ell}}^*(\mathcal{H})$ with the complexity (e.g. the Rademacher complexity) of the family of functions associated with $\widetilde{\ell}$ and $\mathcal{H}$: $\mathcal{H}_{\widetilde{\ell}} = \{(x, y) \mapsto \widetilde{\ell}(h, x, y) : h \in \mathcal{H}\}$.

Let $\mathfrak{R}_m^{\widetilde{\ell}}(\mathcal{H})$ be the Rademacher complexity of $\mathcal{H}_{\widetilde{\ell}}$ and $B_{\widetilde{\ell}}$ an upper bound of the surrogate loss $\widetilde{\ell}$. Then, we obtain the following generalization bounds for the top-$k$ loss.

**Theorem H.1** (**Generalization bound with comp-sum losses**). *Assume that $\mathcal{H}$ is symmetric and complete. Then, for any $1 \leq k \leq n$, the following top-$k$ generalization bound holds for $\widehat{h}_S$: for any $\delta > 0$, with probability at least $1 - \delta$ over the draw of an i.i.d sample $S$ of size $m$:*

$$\mathcal{E}_{\ell_k}(\widehat{h}_S) - \mathcal{E}_{\ell_k}^*(\mathcal{H}) + \mathcal{M}_{\ell_k}(\mathcal{H}) \leq k\psi^{-1}\left(4\mathfrak{R}_m^{\widetilde{\ell}}(\mathcal{H}) + 2B_{\widetilde{\ell}}\sqrt{\frac{\log\frac{2}{\delta}}{2m}} + \mathcal{M}_{\widetilde{\ell}}(\mathcal{H})\right).$$

*where $\psi(t) = \frac{1-t}{2}\log(1-t) + \frac{1+t}{2}\log(1+t)$, $t \in [0, 1]$ when $\widetilde{\ell}$ is $\widetilde{\ell}_{\log}$; $\psi(t) = 1 - \sqrt{1 - t^2}$, $t \in [0, 1]$ when $\widetilde{\ell}$ is $\widetilde{\ell}_{\exp}$; $\psi(t) = t/n$ when $\widetilde{\ell}$ is $\widetilde{\ell}_{\mathrm{mae}}$; and $\psi(t) = \frac{1}{qn^q}\left[\left[\frac{(1+t)^{\frac{1}{1-q}} + (1-t)^{\frac{1}{1-q}}}{2}\right]^{1-q} - 1\right]$, for all $q \in (0, 1)$, $t \in [0, 1]$ when $\widetilde{\ell}$ is $\widetilde{\ell}_{\mathrm{gce}}$.*

*Proof.* By using the standard Rademacher complexity bounds [Mohri et al., 2018], for any $\delta > 0$, with probability at least $1 - \delta$, the following holds for all $h \in \mathcal{H}$:

$$\left|\mathcal{E}_{\widetilde{\ell}}(h) - \widehat{\mathcal{E}}_{\widetilde{\ell}, S}(h)\right| \leq 2\mathfrak{R}_m^{\widetilde{\ell}}(\mathcal{H}) + B_{\widetilde{\ell}}\sqrt{\frac{\log(2/\delta)}{2m}}.$$

Fix $\epsilon > 0$. By the definition of the infimum, there exists $h^* \in \mathcal{H}$ such that $\mathcal{E}_{\widetilde{\ell}}(h^*) \leq \mathcal{E}_{\widetilde{\ell}}^*(\mathcal{H}) + \epsilon$. By definition of $\widehat{h}_S$, we have

$$
\begin{aligned}
&\mathcal{E}_{\widetilde{\ell}}(\widehat{h}_S) - \mathcal{E}_{\widetilde{\ell}}^*(\mathcal{H}) \\
&= \mathcal{E}_{\widetilde{\ell}}(\widehat{h}_S) - \widehat{\mathcal{E}}_{\widetilde{\ell}, S}(\widehat{h}_S) + \widehat{\mathcal{E}}_{\widetilde{\ell}, S}(\widehat{h}_S) - \mathcal{E}_{\widetilde{\ell}}^*(\mathcal{H}) \\
&\leq \mathcal{E}_{\widetilde{\ell}}(\widehat{h}_S) - \widehat{\mathcal{E}}_{\widetilde{\ell}, S}(\widehat{h}_S) + \widehat{\mathcal{E}}_{\widetilde{\ell}, S}(h^*) - \mathcal{E}_{\widetilde{\ell}}^*(\mathcal{H}) \\
&\leq \mathcal{E}_{\widetilde{\ell}}(\widehat{h}_S) - \widehat{\mathcal{E}}_{\widetilde{\ell}, S}(\widehat{h}_S) + \widehat{\mathcal{E}}_{\widetilde{\ell}, S}(h^*) - \mathcal{E}_{\widetilde{\ell}}^*(h^*) + \epsilon \\
&\leq 2\left[2\mathfrak{R}_m^{\widetilde{\ell}}(\mathcal{H}) + B_{\widetilde{\ell}}\sqrt{\frac{\log(2/\delta)}{2m}}\right] + \epsilon.
\end{aligned}
$$

Since the inequality holds for all $\epsilon > 0$, it implies:

$$\mathcal{E}_{\widetilde{\ell}}(\widehat{h}_S) - \mathcal{E}_{\widetilde{\ell}}^*(\mathcal{H}) \leq 4\mathfrak{R}_m^{\widetilde{\ell}}(\mathcal{H}) + 2B_{\widetilde{\ell}}\sqrt{\frac{\log(2/\delta)}{2m}}.$$

Plugging in this inequality in the bounds of Theorem 4.5 completes the proof. $\qquad\square$

**Theorem H.2** (**Generalization bound with constrained losses**). *Assume that $\mathcal{H}$ is symmetric and complete. Then, for any $1 \leq k \leq n$, the following top-$k$ generalization bound holds for $\widehat{h}_S$: for any $\delta > 0$, with probability at least $1 - \delta$ over the draw of an i.i.d sample $S$ of size $m$:*

$$\mathcal{E}_{\ell_k}(\widehat{h}_S) - \mathcal{E}_{\ell_k}^*(\mathcal{H}) + \mathcal{M}_{\ell_k}(\mathcal{H}) \leq k\gamma\left(4\mathfrak{R}_m^{\widetilde{\ell}}(\mathcal{H}) + 2B_{\widetilde{\ell}}\sqrt{\frac{\log\frac{2}{\delta}}{2m}} + \mathcal{M}_{\widetilde{\ell}}(\mathcal{H})\right).$$

*where $\gamma(t) = 2\sqrt{t}$ when $\widetilde{\ell}$ is either $\widetilde{\ell}_{\exp}^{\mathrm{cstnd}}$ or $\widetilde{\ell}_{\mathrm{sq-hinge}}$; $\gamma(t) = t$ when $\widetilde{\ell}$ is either $\widetilde{\ell}_{\mathrm{hinge}}$ or $\widetilde{\ell}_\rho$.*

*Proof.* By using the standard Rademacher complexity bounds [Mohri et al., 2018], for any $\delta > 0$, with probability at least $1 - \delta$, the following holds for all $h \in \mathcal{H}$:

$$\left|\mathcal{E}_{\widetilde{\ell}}(h) - \widehat{\mathcal{E}}_{\widetilde{\ell}, S}(h)\right| \leq 2\mathfrak{R}_m^{\widetilde{\ell}}(\mathcal{H}) + B_{\widetilde{\ell}}\sqrt{\frac{\log(2/\delta)}{2m}}.$$

Fix $\epsilon > 0$. By the definition of the infimum, there exists $h^* \in \mathcal{H}$ such that $\mathcal{E}_{\widetilde{\ell}}(h^*) \leq \mathcal{E}_{\widetilde{\ell}}^*(\mathcal{H}) + \epsilon$. By definition of $\widehat{h}_S$, we have

$$
\begin{aligned}
&\mathcal{E}_{\widetilde{\ell}}(\widehat{h}_S) - \mathcal{E}_{\widetilde{\ell}}^*(\mathcal{H}) \\
&= \mathcal{E}_{\widetilde{\ell}}(\widehat{h}_S) - \widehat{\mathcal{E}}_{\widetilde{\ell},S}(\widehat{h}_S) + \widehat{\mathcal{E}}_{\widetilde{\ell},S}(\widehat{h}_S) - \mathcal{E}_{\widetilde{\ell}}^*(\mathcal{H}) \\
&\leq \mathcal{E}_{\widetilde{\ell}}(\widehat{h}_S) - \widehat{\mathcal{E}}_{\widetilde{\ell},S}(\widehat{h}_S) + \widehat{\mathcal{E}}_{\widetilde{\ell},S}(h^*) - \mathcal{E}_{\widetilde{\ell}}^*(\mathcal{H}) \\
&\leq \mathcal{E}_{\widetilde{\ell}}(\widehat{h}_S) - \widehat{\mathcal{E}}_{\widetilde{\ell},S}(\widehat{h}_S) + \widehat{\mathcal{E}}_{\widetilde{\ell},S}(h^*) - \mathcal{E}_{\widetilde{\ell}}^*(h^*) + \epsilon \\
&\leq 2\left[ 2\mathfrak{R}_m^{\widetilde{\ell}}(\mathcal{H}) + B_{\widetilde{\ell}} \sqrt{\frac{\log(2/\delta)}{2m}} \right] + \epsilon.
\end{aligned}
$$

Since the inequality holds for all $\epsilon > 0$, it implies:

$$
\mathcal{E}_{\widetilde{\ell}}(\widehat{h}_S) - \mathcal{E}_{\widetilde{\ell}}^*(\mathcal{H}) \leq 4\mathfrak{R}_m^{\widetilde{\ell}}(\mathcal{H}) + 2B_{\widetilde{\ell}} \sqrt{\frac{\log(2/\delta)}{2m}}.
$$

Plugging in this inequality in the bounds of Theorem E.1 completes the proof. $\qquad\square$

To the best of our knowledge, Theorems H.1 and H.2 provide the first finite-sample guarantees for the estimation error of the minimizer of comp-sum losses and constrained losses, with respect to the top-$k$ loss, for any $1 \leq k \leq n$. The proofs use our $\mathcal{H}$-consistency bounds with respect to the top-$k$ loss, as well as standard Rademacher complexity guarantees.

# I  Proofs of $\mathcal{H}$-consistency bounds for cost-sensitive losses

We first characterize the best-in class conditional error and the conditional regret of the target cardinality aware loss function (2), which will be used in the analysis of $\mathcal{H}$-consistency bounds.

**Lemma I.1.** *Assume that $\mathcal{R}$ is symmetric and complete. Then, for any $r \in \mathcal{K}$ and $x \in \mathcal{X}$, the best-in class conditional error and the conditional regret of the target cardinality aware loss function can be expressed as follows:*

$$\mathcal{C}_\ell^*(\mathcal{R}, x) = \min_{k \in \mathcal{K}} \sum_{y \in \mathcal{Y}} p(x,y) c(x,k,y)$$

$$\Delta \mathcal{C}_{\ell,\mathcal{R}}(r,x) = \sum_{y \in \mathcal{Y}} p(x,y) c(x,\mathsf{r}(x),y) - \min_{k \in \mathcal{K}} \sum_{y \in \mathcal{Y}} p(x,y) c(x,k,y).$$

*Proof.* By definition, for any $r \in \mathcal{R}$ and $x \in \mathcal{X}$, the conditional error of the target cardinality aware loss function can be written as

$$\mathcal{C}_\ell(r,x) = \sum_{y \in \mathcal{Y}} p(x,y) c(x,\mathsf{r}(x),y).$$

Since $\mathcal{R}$ is symmetric and complete, we have

$$\mathcal{C}_\ell^*(\mathcal{R}, x) = \inf_{r \in \mathcal{R}} \sum_{y \in \mathcal{Y}} p(x,y) c(x,\mathsf{r}(x),y) = \min_{k \in \mathcal{K}} \sum_{y \in \mathcal{Y}} p(x,y) c(x,k,y).$$

Furthermore, the calibration gap can be expressed as

$$\Delta \mathcal{C}_{\ell,\mathcal{R}}(r,x) = \mathcal{C}_\ell(r,x) - \mathcal{C}_\ell^*(\mathcal{R}, x) = \sum_{y \in \mathcal{Y}} p(x,y) c(x,\mathsf{r}(x),y) - \min_{k \in \mathcal{K}} \sum_{y \in \mathcal{Y}} p(x,y) c(x,k,y),$$

which completes the proof. $\qquad\square$

## I.1  Proof of Theorem 4.6

For convenience, we let $\overline{c}(x,k,y) = 1 - c(x,k,y)$, $\overline{q}(x,k) = \sum_{y \in \mathcal{Y}} p(x,y) \overline{c}(x,k,y) \in [0,1]$ and $\mathcal{S}(x,k) = \frac{e^{r(x,k)}}{\sum_{k' \in \mathcal{K}} e^{r(x,k')}}$. We also let $k_{\min}(x) = \operatorname{argmin}_{k \in \mathcal{K}}(1 - \overline{q}(x,k)) = \operatorname{argmin}_{k \in \mathcal{K}} \sum_{y \in \mathcal{Y}} p(x,y) c(x,k,y)$.

**Theorem 4.6.** *Assume that $\mathcal{R}$ is symmetric and complete. Then, the following bound holds for the cost-sensitive comp-sum loss: for all $r \in \mathcal{R}$ and for any distribution,*

$$\mathcal{E}_\ell(r) - \mathcal{E}_\ell^*(\mathcal{R}) + \mathcal{M}_\ell(\mathcal{R}) \le \gamma\Big( \mathcal{E}_{\widetilde{\ell}_{\text{c-comp}}}(r) - \mathcal{E}_{\widetilde{\ell}_{\text{c-comp}}}^*(\mathcal{R}) + \mathcal{M}_{\widetilde{\ell}_{\text{c-comp}}}(\mathcal{R}) \Big);$$

*When $\mathcal{R} = \mathcal{R}_{\text{all}}$, the following holds: $\mathcal{E}_\ell(r) - \mathcal{E}_\ell^*(\mathcal{R}_{\text{all}}) \le \gamma\Big( \mathcal{E}_{\widetilde{\ell}_{\text{c-comp}}}(r) - \mathcal{E}_{\widetilde{\ell}_{\text{c-comp}}}^*(\mathcal{R}_{\text{all}}) \Big)$, where $\gamma(t) = 2\sqrt{t}$ when $\widetilde{\ell}_{\text{c-comp}}$ is either $\widetilde{\ell}_{\text{c-log}}$ or $\widetilde{\ell}_{\text{c-exp}}$; $\gamma(t) = 2\sqrt{|\mathcal{K}|^q t}$ when $\widetilde{\ell}_{\text{c-comp}}$ is $\widetilde{\ell}_{\text{c-gce}}$; and $\gamma(t) = |\mathcal{K}| t$ when $\widetilde{\ell}_{\text{c-comp}}$ is $\widetilde{\ell}_{\text{c-mae}}$.*

*Proof.* **Case I:** $\widetilde{\ell}_{\text{c-comp}} = \widetilde{\ell}_{\text{c-log}}$. For the cost-sensitive logistic loss $\widetilde{\ell}_{\text{c-log}}$, the conditional error can be written as

$$\mathcal{C}_{\widetilde{\ell}_{\text{c-log}}}(r,x) = - \sum_{y \in \mathcal{Y}} p(x,y) \sum_{k \in \mathcal{K}} \overline{c}(x,k,y) \log\left( \frac{e^{r(x,k)}}{\sum_{k' \in \mathcal{K}} e^{r(x,k')}} \right) = - \sum_{k \in \mathcal{K}} \log(\mathcal{S}(x,k)) \overline{q}(x,k).$$

The conditional regret can be written as

$$\Delta \mathcal{C}_{\widetilde{\ell}_{\text{c-log}},\mathcal{R}}(r,x) = - \sum_{k \in \mathcal{K}} \log(\mathcal{S}(x,k)) \overline{q}(x,k) - \inf_{r \in \mathcal{R}} \left( - \sum_{k \in \mathcal{K}} \log(\mathcal{S}(x,k)) \overline{q}(x,k) \right)$$

$$\ge - \sum_{k \in \mathcal{K}} \log(\mathcal{S}(x,k)) \overline{q}(x,k) - \inf_{\mu \in [-\mathcal{S}(x,k_{\min}(x)), \mathcal{S}(x,\mathsf{r}(x))]} \left( - \sum_{k \in \mathcal{K}} \log(\mathcal{S}_\mu(x,k)) \overline{q}(x,k) \right),$$

where for any $x \in \mathcal{X}$ and $k \in \mathcal{K}$, $\mathcal{S}_\mu(x,k) = \begin{cases} \mathcal{S}(x,y), & y \notin \{k_{\min}(x), \mathsf{r}(x)\} \\ \mathcal{S}(x, k_{\min}(x)) + \mu & y = \mathsf{r}(x) \\ \mathcal{S}(x, \mathsf{r}(x)) - \mu & y = k_{\min}(x). \end{cases}$ Note that such a choice of $\mathcal{S}_\mu$ leads to the following equality holds:

$$\sum_{k \notin \{\mathsf{r}(x), k_{\min}(x)\}} \log(\mathcal{S}(x,k)) \overline{q}(x,k) = \sum_{k \notin \{\mathsf{r}(x), k_{\min}(x)\}} \log(\mathcal{S}_\mu(x,k)) \overline{q}(x,k).$$

Therefore, the conditional regret of cost-sensitive logistic loss can be lower bounded as

$$\Delta \mathcal{C}_{\widetilde{\ell}_{\mathrm{c-log}}, \mathcal{H}}(h,x)$$

$$\geq \sup_{\mu \in [-\mathcal{S}(x, k_{\min}(x)), \mathcal{S}(x, \mathsf{r}(x))]} \Big\{ \overline{q}(x, k_{\min}(x)) [-\log(\mathcal{S}(x, k_{\min}(x))) + \log(\mathcal{S}(x, \mathsf{r}(x)) - \mu)]$$

$$+ \overline{q}(x, \mathsf{r}(x)) [-\log(\mathcal{S}(x, \mathsf{r}(x))) + \log(\mathcal{S}(x, k_{\min}(x)) + \mu)] \Big\}.$$

By the concavity of the function, differentiate with respect to $\mu$, we obtain that the supremum is achieved by $\mu^* = \frac{\overline{q}(x, \mathsf{r}(x)) \mathcal{S}(x, \mathsf{r}(x)) - \overline{q}(x, k_{\min}(x)) \mathcal{S}(x, k_{\min}(x))}{\overline{q}(x, k_{\min}(x)) + \overline{q}(x, \mathsf{r}(x))}$. Plug in $\mu^*$, we obtain

$$\Delta \mathcal{C}_{\widetilde{\ell}_{\mathrm{c-log}}, \mathcal{H}}(h,x)$$

$$\geq \overline{q}(x, k_{\min}(x)) \log \frac{(\mathcal{S}(x, \mathsf{r}(x)) + \mathcal{S}(x, k_{\min}(x))) \overline{q}(x, k_{\min}(x))}{\mathcal{S}(x, k_{\min}(x))(\overline{q}(x, k_{\min}(x)) + \overline{q}(x, \mathsf{r}(x)))}$$

$$+ \overline{q}(x, \mathsf{r}(x)) \log \frac{(\mathcal{S}(x, \mathsf{r}(x)) + \mathcal{S}(x, k_{\min}(x))) \overline{q}(x, \mathsf{r}(x))}{\mathcal{S}(x, \mathsf{r}(x))(\overline{q}(x, k_{\min}(x)) + \overline{q}(x, \mathsf{r}(x)))}$$

$$\geq \overline{q}(x, k_{\min}(x)) \log \frac{2\overline{q}(x, k_{\min}(x))}{\overline{q}(x, k_{\min}(x)) + \overline{q}(x, \mathsf{r}(x))} + \overline{q}(x, \mathsf{r}(x)) \log \frac{2\overline{q}(x, \mathsf{r}(x))}{\overline{q}(x, k_{\min}(x)) + \overline{q}(x, \mathsf{r}(x))}$$

$$\text{(minimum is achieved when } \mathcal{S}(x, \mathsf{r}(x)) = \mathcal{S}(x, k_{\min}(x)))$$

$$\geq \frac{(\overline{q}(x, \mathsf{r}(x)) - \overline{q}(x, k_{\min}(x)))^2}{2(\overline{q}(x, \mathsf{r}(x)) + \overline{q}(x, k_{\min}(x)))}$$

$$(a \log \frac{2a}{a+b} + b \log \frac{2b}{a+b} \geq \frac{(a-b)^2}{2(a+b)}, \forall a,b \in [0,1] \text{ [Mohri et al., 2018, Proposition E.7])}$$

$$\geq \frac{(\overline{q}(x, \mathsf{r}(x)) - \overline{q}(x, k_{\min}(x)))^2}{4}. \qquad (0 \leq \overline{q}(x, \mathsf{r}(x)) + \overline{q}(x, k_{\min}(x)) \leq 2)$$

Therefore, by Lemma I.1, the conditional regret of the target cardinality aware loss function can be upper bounded as follows:

$$\Delta \mathcal{C}_{\ell, \mathcal{H}}(r,x) = \overline{q}(x, k_{\min}(x)) - \overline{q}(x, \mathsf{r}(x)) \leq 2 \Big( \Delta \mathcal{C}_{\widetilde{\ell}_{\mathrm{c-log}}, \mathcal{R}}(r,x) \Big)^{\frac{1}{2}}.$$

By the concavity, taking expectations on both sides of the preceding equation, we obtain

$$\mathcal{E}_\ell(r) - \mathcal{E}_\ell^*(\mathcal{R}) + \mathcal{M}_\ell(\mathcal{R}) \leq 2 \Big( \mathcal{E}_{\widetilde{\ell}_{\mathrm{c-log}}}(r) - \mathcal{E}_{\widetilde{\ell}_{\mathrm{c-log}}}^*(\mathcal{R}) + \mathcal{M}_{\widetilde{\ell}_{\mathrm{c-log}}}(\mathcal{R}) \Big)^{\frac{1}{2}}.$$

The second part follows from the fact that $\mathcal{M}_{\widetilde{\ell}_{\mathrm{c-log}}}(\mathcal{R}_{\mathrm{all}}) = 0$.

**Case II: $\widetilde{\ell}_{\mathrm{c-comp}} = \widetilde{\ell}_{\mathrm{c-exp}}$.** For the cost-sensitive sum exponential loss $\widetilde{\ell}_{\mathrm{c-exp}}$, the conditional error can be written as

$$\mathcal{C}_{\widetilde{\ell}_{\mathrm{c-exp}}}(r,x) = \sum_{y \in \mathcal{Y}} p(x,y) \sum_{k \in \mathcal{K}} \overline{c}(x,k,y) \sum_{k' \neq k'} e^{r(x,k') - r(x,k)} = \sum_{k \in \mathcal{K}} \left( \frac{1}{\mathcal{S}(x,k)} - 1 \right) \overline{q}(x,k).$$

The conditional regret can be written as

$$\Delta \mathcal{C}_{\widetilde{\ell}_{\mathrm{c-exp}}, \mathcal{R}}(r,x) = \sum_{k \in \mathcal{K}} \left( \frac{1}{\mathcal{S}(x,k)} - 1 \right) \overline{q}(x,k) - \inf_{r \in \mathcal{R}} \left( \sum_{k \in \mathcal{K}} \left( \frac{1}{\mathcal{S}(x,k)} - 1 \right) \overline{q}(x,k) \right)$$

$$\geq \sum_{k \in \mathcal{K}} \left( \frac{1}{\mathcal{S}(x,k)} - 1 \right) \overline{q}(x,k) - \inf_{\mu \in [-\mathcal{S}(x, k_{\min}(x)), \mathcal{S}(x, \mathsf{r}(x))]} \left( \sum_{k \in \mathcal{K}} \left( \frac{1}{\mathcal{S}_\mu(x,k)} - 1 \right) \overline{q}(x,k) \right),$$

where for any $x \in \mathcal{X}$ and $k \in \mathcal{K}$, $\mathcal{S}_\mu(x, k) = \begin{cases} \mathcal{S}(x, y), & y \notin \{k_{\min}(x), \mathsf{r}(x)\} \\ \mathcal{S}(x, k_{\min}(x)) + \mu & y = \mathsf{r}(x) \\ \mathcal{S}(x, \mathsf{r}(x)) - \mu & y = k_{\min}(x). \end{cases}$ Note that

such a choice of $\mathcal{S}_\mu$ leads to the following equality holds:

$$\sum_{k \notin \{\mathsf{r}(x), k_{\min}(x)\}} \left( \frac{1}{\mathcal{S}(x, k)} - 1 \right) \overline{q}(x, k) = \sum_{k \notin \{\mathsf{r}(x), k_{\min}(x)\}} \left( \frac{1}{\mathcal{S}_\mu(x, k)} - 1 \right) \overline{q}(x, k).$$

Therefore, the conditional regret of cost-sensitive sum exponential loss can be lower bounded as

$$\Delta \mathcal{C}_{\widetilde{\ell}_{\text{c-exp}}, \mathcal{H}}(h, x) \geq \sup_{\mu \in [-\mathcal{S}(x, k_{\min}(x)), \mathcal{S}(x, \mathsf{r}(x))]} \left\{ \overline{q}(x, k_{\min}(x)) \left[ \frac{1}{\mathcal{S}(x, k_{\min}(x))} - \frac{1}{\mathcal{S}(x, \mathsf{r}(x)) - \mu} \right] \right.$$
$$\left. + \overline{q}(x, \mathsf{r}(x)) \left[ \frac{1}{\mathcal{S}(x, \mathsf{r}(x))} - \frac{1}{\mathcal{S}(x, k_{\min}(x)) + \mu} \right] \right\}.$$

By the concavity of the function, differentiate with respect to $\mu$, we obtain that the supremum is achieved by $\mu^* = \frac{\sqrt{\overline{q}(x, \mathsf{r}(x))} \mathcal{S}(x, \mathsf{r}(x)) - \sqrt{\overline{q}(x, k_{\min}(x))} \mathcal{S}(x, k_{\min}(x))}{\sqrt{\overline{q}(x, k_{\min}(x))} + \sqrt{\overline{q}(x, \mathsf{r}(x))}}$. Plug in $\mu^*$, we obtain

$$\Delta \mathcal{C}_{\widetilde{\ell}_{\text{c-exp}}, \mathcal{H}}(h, x)$$

$$\geq \frac{\overline{q}(x, k_{\min}(x))}{\mathcal{S}(x, k_{\min}(x))} + \frac{\overline{q}(x, \mathsf{r}(x)))}{\mathcal{S}(x, \mathsf{r}(x)))} - \frac{\left( \sqrt{\overline{q}(x, k_{\min}(x))} + \sqrt{\overline{q}(x, \mathsf{r}(x)))} \right)^2}{\mathcal{S}(x, k_{\min}(x)) + \mathcal{S}(x, \mathsf{r}(x)))}$$

$$\geq \left( \sqrt{\overline{q}(x, k_{\min}(x))} - \sqrt{\overline{q}(x, \mathsf{r}(x)))} \right)^2$$
$$\text{(minimum is achieved when } \mathcal{S}(x, \mathsf{r}(x)) = \mathcal{S}(x, k_{\min}(x)) = \tfrac{1}{2})$$

$$\geq \frac{(\overline{q}(x, \mathsf{r}(x))) - \overline{q}(x, k_{\min}(x)))^2}{\left( \sqrt{\overline{q}(x, \mathsf{r}(x)))} + \sqrt{\overline{q}(x, k_{\min}(x))} \right)^2}$$

$$\geq \frac{(\overline{q}(x, \mathsf{r}(x))) - \overline{q}(x, k_{\min}(x)))^2}{4}. \qquad (\sqrt{a} + \sqrt{b} \leq 2, \forall a, b \in [0, 1], a + b \leq 2)$$

Therefore, by Lemma I.1, the conditional regret of the target cardinality aware loss function can be upper bounded as follows:

$$\Delta \mathcal{C}_{\ell, \mathcal{H}}(r, x) = \overline{q}(x, k_{\min}(x)) - \overline{q}(x, \mathsf{r}(x)) \leq 2 \left( \Delta \mathcal{C}_{\widetilde{\ell}_{\text{c-exp}}, \mathcal{R}}(r, x) \right)^{\frac{1}{2}}.$$

By the concavity, taking expectations on both sides of the preceding equation, we obtain

$$\mathcal{E}_\ell(r) - \mathcal{E}_\ell^*(\mathcal{R}) + \mathcal{M}_\ell(\mathcal{R}) \leq 2 \left( \mathcal{E}_{\widetilde{\ell}_{\text{c-exp}}}(r) - \mathcal{E}_{\widetilde{\ell}_{\text{c-exp}}}^*(\mathcal{R}) + \mathcal{M}_{\widetilde{\ell}_{\text{c-exp}}}(\mathcal{R}) \right)^{\frac{1}{2}}.$$

The second part follows from the fact that $\mathcal{M}_{\widetilde{\ell}_{\text{c-exp}}}(\mathcal{R}_{\text{all}}) = 0$.

**Case III:** $\widetilde{\ell}_{\text{c-comp}} = \widetilde{\ell}_{\text{c-gce}}$. For the cost-sensitive generalized cross-entropy loss $\widetilde{\ell}_{\text{c-gce}}$, the conditional error can be written as

$$\mathcal{C}_{\widetilde{\ell}_{\text{c-gce}}}(r, x) = \sum_{y \in \mathcal{Y}} p(x, y) \sum_{k \in \mathcal{K}} \overline{c}(x, k, y) \frac{1}{q} \left( 1 - \left( \frac{e^{r(x, k)}}{\sum_{k' \in \mathcal{K}} e^{r(x, k')}} \right)^q \right) = \frac{1}{q} \sum_{k \in \mathcal{K}} (1 - \mathcal{S}(x, k)^q) \overline{q}(x, k).$$

The conditional regret can be written as

$$\Delta \mathcal{C}_{\widetilde{\ell}_{\text{c-gce}}, \mathcal{R}}(r, x) = \frac{1}{q} \sum_{k \in \mathcal{K}} (1 - \mathcal{S}(x, k)^q) \overline{q}(x, k) - \inf_{r \in \mathcal{R}} \left( \frac{1}{q} \sum_{k \in \mathcal{K}} (1 - \mathcal{S}(x, k)^q) \overline{q}(x, k) \right)$$

$$\geq \frac{1}{q} \sum_{k \in \mathcal{K}} (1 - \mathcal{S}(x, k)^q) \overline{q}(x, k) - \inf_{\mu \in [-\mathcal{S}(x, k_{\min}(x)), \mathcal{S}(x, \mathsf{r}(x))]} \left( \frac{1}{q} \sum_{k \in \mathcal{K}} (1 - \mathcal{S}_\mu(x, k)^q) \overline{q}(x, k) \right),$$

where for any $x \in \mathcal{X}$ and $k \in \mathcal{K}$, $\mathcal{S}_\mu(x, k) = \begin{cases} \mathcal{S}(x, y), & y \notin \{k_{\min}(x), \mathsf{r}(x)\} \\ \mathcal{S}(x, k_{\min}(x)) + \mu & y = \mathsf{r}(x) \\ \mathcal{S}(x, \mathsf{r}(x)) - \mu & y = k_{\min}(x). \end{cases}$ Note that

such a choice of $\mathcal{S}_\mu$ leads to the following equality holds:

$$\sum_{k \notin \{\mathsf{r}(x), k_{\min}(x)\}} \frac{1}{q} \sum_{k \in \mathcal{K}} (1 - \mathcal{S}(x, k)^q) \overline{q}(x, k) = \sum_{k \notin \{\mathsf{r}(x), k_{\min}(x)\}} \frac{1}{q} \sum_{k \in \mathcal{K}} (1 - \mathcal{S}_\mu(x, k)^q) \overline{q}(x, k).$$

Therefore, the conditional regret of cost-sensitive generalized cross-entropy loss can be lower bounded as

$$\Delta\mathcal{C}_{\widetilde{\ell}_{\text{c-gce}}, \mathcal{H}}(h, x) = \frac{1}{q} \sup_{\mu \in [-\mathcal{S}(x, k_{\min}(x)), \mathcal{S}(x, \mathsf{r}(x))]} \left\{ \overline{q}(x, k_{\min}(x)) \left[ -\mathcal{S}(x, k_{\min}(x))^q + (\mathcal{S}(x, \mathsf{r}(x)) - \mu)^q \right] \right.$$
$$\left. + \overline{q}(x, \mathsf{r}(x)) \left[ -\mathcal{S}(x, \mathsf{r}(x))^q + (\mathcal{S}(x, k_{\min}(x)) + \mu)^q \right] \right\}.$$

By the concavity of the function, differentiate with respect to $\mu$, we obtain that the supremum is achieved by $\mu^* = \frac{\overline{q}(x, \mathsf{r}(x))^{\frac{1}{1-q}} \mathcal{S}(x, \mathsf{r}(x)) - \overline{q}(x, k_{\min}(x))^{\frac{1}{1-q}} \mathcal{S}(x, k_{\min}(x))}{\overline{q}(x, k_{\min}(x))^{\frac{1}{1-q}} + \overline{q}(x, \mathsf{r}(x))^{\frac{1}{1-q}}}$. Plug in $\mu^*$, we obtain

$$\Delta\mathcal{C}_{\widetilde{\ell}_{\text{c-gce}}, \mathcal{H}}(h, x)$$
$$\geq \frac{1}{q} (\mathcal{S}(x, \mathsf{r}(x)) + \mathcal{S}(x, k_{\min}(x)))^q \left( \overline{q}(x, k_{\min}(x))^{\frac{1}{1-q}} + \overline{q}(x, \mathsf{r}(x))^{\frac{1}{1-q}} \right)^{1-q}$$
$$- \frac{1}{q} \overline{q}(x, k_{\min}(x)) \mathcal{S}(x, k_{\min}(x))^q - \frac{1}{q} \overline{q}(x, \mathsf{r}(x)) \mathcal{S}(x, \mathsf{r}(x))^q$$
$$\geq \frac{1}{q|\mathcal{K}|^q} \left[ 2^q \left( \overline{q}(x, k_{\min}(x))^{\frac{1}{1-q}} + \overline{q}(x, \mathsf{r}(x))^{\frac{1}{1-q}} \right)^{1-q} - \overline{q}(x, k_{\min}(x)) - \overline{q}(x, \mathsf{r}(x)) \right]$$
$$\text{(minimum is achieved when } \mathcal{S}(x, \mathsf{r}(x)) = \mathcal{S}(x, k_{\min}(x)) = \frac{1}{|\mathcal{K}|})$$
$$\geq \frac{(\overline{q}(x, \mathsf{r}(x)) - \overline{q}(x, k_{\min}(x)))^2}{4|\mathcal{K}|^q}.$$
$$\left( \left( \frac{a^{\frac{1}{1-q}} + b^{\frac{1}{1-q}}}{2} \right)^{1-q} - \frac{a+b}{2} \geq \frac{q}{4}(a - b)^2, \forall a, b \in [0, 1], 0 \leq a + b \leq 1 \right)$$

Therefore, by Lemma I.1, the conditional regret of the target cardinality aware loss function can be upper bounded as follows:

$$\Delta\mathcal{C}_{\ell, \mathcal{H}}(r, x) = \overline{q}(x, k_{\min}(x)) - \overline{q}(x, \mathsf{r}(x)) \leq 2|\mathcal{K}|^{\frac{q}{2}} \left( \Delta\mathcal{C}_{\widetilde{\ell}_{\text{c-gce}}, \mathcal{R}}(r, x) \right)^{\frac{1}{2}}.$$

By the concavity, taking expectations on both sides of the preceding equation, we obtain

$$\mathcal{E}_\ell(r) - \mathcal{E}_\ell^*(\mathcal{R}) + \mathcal{M}_\ell(\mathcal{R}) \leq 2|\mathcal{K}|^{\frac{q}{2}} \left( \mathcal{E}_{\widetilde{\ell}_{\text{c-gce}}}(r) - \mathcal{E}_{\widetilde{\ell}_{\text{c-gce}}}^*(\mathcal{R}) + \mathcal{M}_{\widetilde{\ell}_{\text{c-gce}}}(\mathcal{R}) \right)^{\frac{1}{2}}.$$

The second part follows from the fact that $\mathcal{M}_{\widetilde{\ell}_{\text{c-gce}}}(\mathcal{R}_{\text{all}}) = 0$.

**Case IV:** $\widetilde{\ell}_{\text{c-comp}} = \widetilde{\ell}_{\text{c-mae}}$. For the cost-sensitive mean absolute error loss $\widetilde{\ell}_{\text{c-mae}}$, the conditional error can be written as

$$\mathcal{C}_{\widetilde{\ell}_{\text{c-mae}}}(r, x) = \sum_{y \in \mathcal{Y}} p(x, y) \sum_{k \in \mathcal{K}} \overline{c}(x, k, y) \left( 1 - \left( \frac{e^{r(x, k)}}{\sum_{k' \in \mathcal{K}} e^{r(x, k')}} \right) \right) = \sum_{k \in \mathcal{K}} (1 - \mathcal{S}(x, k)) \overline{q}(x, k).$$

The conditional regret can be written as

$$\Delta\mathcal{C}_{\widetilde{\ell}_{\text{c-mae}}, \mathcal{R}}(r, x) = \sum_{k \in \mathcal{K}} (1 - \mathcal{S}(x, k)) \overline{q}(x, k) - \inf_{r \in \mathcal{R}} \left( \sum_{k \in \mathcal{K}} (1 - \mathcal{S}(x, k)) \overline{q}(x, k) \right)$$
$$\geq \sum_{k \in \mathcal{K}} (1 - \mathcal{S}(x, k)) \overline{q}(x, k) - \inf_{\mu \in [-\mathcal{S}(x, k_{\min}(x)), \mathcal{S}(x, \mathsf{r}(x))]} \left( \sum_{k \in \mathcal{K}} (1 - \mathcal{S}_\mu(x, k)) \overline{q}(x, k) \right),$$

where for any $x \in \mathcal{X}$ and $k \in \mathcal{K}$, $\mathcal{S}_\mu(x,k) = \begin{cases} \mathcal{S}(x,y), & y \notin \{k_{\min}(x), \mathsf{r}(x)\} \\ \mathcal{S}(x, k_{\min}(x)) + \mu & y = \mathsf{r}(x) \\ \mathcal{S}(x, \mathsf{r}(x)) - \mu & y = k_{\min}(x). \end{cases}$  Note that such a choice of $\mathcal{S}_\mu$ leads to the following equality holds:

$$\sum_{k \in \mathcal{K}} (1 - \mathcal{S}(x,k))\overline{q}(x,k) = \sum_{k \in \mathcal{K}} (1 - \mathcal{S}_\mu(x,k))\overline{q}(x,k).$$

Therefore, the conditional regret of cost-sensitive mean absolute error can be lower bounded as

$$\Delta\mathcal{C}_{\widetilde{\ell}_{\text{c-mae}}, \mathcal{H}}(h,x) \geq \sup_{\mu \in [-\mathcal{S}(x,k_{\min}(x)), \mathcal{S}(x,\mathsf{r}(x))]} \Big\{ \overline{q}(x, k_{\min}(x))[-\mathcal{S}(x, k_{\min}(x)) + \mathcal{S}(x, \mathsf{r}(x)) - \mu]$$

$$+ \overline{q}(x, \mathsf{r}(x))[-\mathcal{S}(x, \mathsf{r}(x)) + \mathcal{S}(x, k_{\min}(x)) + \mu] \Big\}.$$

By the concavity of the function, differentiate with respect to $\mu$, we obtain that the supremum is achieved by $\mu^* = -\mathcal{S}(x, k_{\min}(x))$. Plug in $\mu^*$, we obtain

$$\Delta\mathcal{C}_{\widetilde{\ell}_{\text{c-mae}}, \mathcal{H}}(h,x)$$
$$\geq \overline{q}(x, k_{\min}(x))\mathcal{S}(x, \mathsf{r}(x)) - \overline{q}(x, \mathsf{r}(x))\mathcal{S}(x, \mathsf{r}(x))$$
$$\geq \frac{1}{|\mathcal{K}|}(\overline{q}(x, k_{\min}(x)) - \overline{q}(x, \mathsf{r}(x))). \qquad \text{(minimum is achieved when } \mathcal{S}(x, \mathsf{r}(x)) = \tfrac{1}{|\mathcal{K}|})$$

Therefore, by Lemma I.1, the conditional regret of the target cardinality aware loss function can be upper bounded as follows:

$$\Delta\mathcal{C}_{\ell, \mathcal{H}}(r,x) = \overline{q}(x, k_{\min}(x)) - \overline{q}(x, \mathsf{r}(x)) \leq |\mathcal{K}|\big(\Delta\mathcal{C}_{\widetilde{\ell}_{\text{c-mae}}, \mathcal{R}}(r,x)\big).$$

By the concavity, taking expectations on both sides of the preceding equation, we obtain

$$\mathcal{E}_\ell(r) - \mathcal{E}_\ell^*(\mathcal{R}) + \mathcal{M}_\ell(\mathcal{R}) \leq |\mathcal{K}|\Big(\mathcal{E}_{\widetilde{\ell}_{\text{c-mae}}}(r) - \mathcal{E}_{\widetilde{\ell}_{\text{c-mae}}}^*(\mathcal{R}) + \mathcal{M}_{\widetilde{\ell}_{\text{c-mae}}}(\mathcal{R})\Big).$$

The second part follows from the fact that $\mathcal{M}_{\widetilde{\ell}_{\text{c-mae}}}(\mathcal{R}_{\text{all}}) = 0$. $\qquad \square$

## I.2  Proof of Theorem 4.7

The conditional error for the cost-sensitive constrained loss can be expressed as follows:

$$\begin{aligned} \mathcal{C}_{\widetilde{\ell}_{\text{c-cstnd}}}(r,x) &= \sum_{y \in \mathcal{Y}} p(x,y)\widetilde{\ell}_{\text{c-cstnd}}(r,x,y) \\ &= \sum_{y \in \mathcal{Y}} p(x,y) \sum_{k \in \mathcal{K}} c(x,k,y)\Phi(-r(x,k)) \\ &= \sum_{k \in \mathcal{K}} \widetilde{q}(x,k)\Phi(-r(x,k)), \end{aligned}$$

where $\widetilde{q}(x,k) = \sum_{y \in \mathcal{Y}} p(x,y)c(x,k,y) \in [0,1]$. Let $k_{\min}(x) = \operatorname{argmin}_{k \in \mathcal{K}} \widetilde{q}(x,k)$. We denote by $\Phi_{\exp}: t \mapsto e^{-t}$ the exponential loss function, $\Phi_{\text{sq-hinge}}: t \mapsto \max\{0, 1-t\}^2$ the squared hinge loss function, $\Phi_{\text{hinge}}: t \mapsto \max\{0, 1-t\}$ the hinge loss function, and $\Phi_\rho: t \mapsto \min\{\max\{0, 1-t/\rho\}, 1\}$, $\rho > 0$ the $\rho$-margin loss function.

**Theorem 4.7.** *Assume that $\mathcal{R}$ is symmetric and complete. Then, the following bound holds for the cost-sensitive constrained loss: for all $r \in \mathcal{R}$ and for any distribution,*

$$\mathcal{E}_\ell(r) - \mathcal{E}_\ell^*(\mathcal{R}) + \mathcal{M}_\ell(\mathcal{R}) \leq \gamma\Big(\mathcal{E}_{\widetilde{\ell}_{\text{c-cstnd}}}(r) - \mathcal{E}_{\widetilde{\ell}_{\text{c-cstnd}}}^*(\mathcal{R}) + \mathcal{M}_{\widetilde{\ell}_{\text{c-cstnd}}}(\mathcal{R})\Big);$$

*When $\mathcal{R} = \mathcal{R}_{\text{all}}$, the following holds: $\mathcal{E}_\ell(r) - \mathcal{E}_\ell^*(\mathcal{R}_{\text{all}}) \leq \gamma\Big(\mathcal{E}_{\widetilde{\ell}_{\text{c-cstnd}}}(r) - \mathcal{E}_{\widetilde{\ell}_{\text{c-cstnd}}}^*(\mathcal{R}_{\text{all}})\Big)$, where $\gamma(t) = 2\sqrt{t}$ when $\widetilde{\ell}_{\text{c-cstnd}}$ is $\widetilde{\ell}_{\text{c-exp}}^{\text{cstnd}}$ or $\widetilde{\ell}_{\text{c-sq-hinge}}$; $\gamma(t) = t$ when $\widetilde{\ell}_{\text{c-cstnd}}$ is $\widetilde{\ell}_{\text{c-hinge}}$ or $\widetilde{\ell}_{\text{c-}\rho}$.*

*Proof.* **Case I:** $\ell = \widetilde{\ell}_{\text{c-exp}}^{\text{cstnd}}$. For the cost-sensitive constrained exponential loss $\widetilde{\ell}_{\text{c-exp}}^{\text{cstnd}}$, the conditional regret can be written as

$$\Delta \mathcal{C}_{\widetilde{\ell}_{\text{c-exp}}^{\text{cstnd}}, \mathcal{R}}(r, x) = \sum_{k \in \mathcal{K}} \widetilde{q}(x, k) \Phi_{\exp}(-r(x, k)) - \inf_{r \in \mathcal{R}} \sum_{k \in \mathcal{K}} \widetilde{q}(x, k) \Phi_{\exp}(-r(x, k))$$

$$\geq \sum_{k \in \mathcal{K}} \widetilde{q}(x, k) \Phi_{\exp}(-r(x, k)) - \inf_{\mu \in \mathbb{R}} \sum_{k \in \mathcal{K}} \widetilde{q}(x, k) \Phi_{\exp}(-r_\mu(x, k)),$$

where for any $k \in \mathcal{K}$, $r_\mu(x, k) = \begin{cases} r(x, y), & y \notin \{k_{\min}(x), \mathsf{r}(x)\} \\ r(x, k_{\min}(x)) + \mu & y = \mathsf{r}(x) \\ r(x, \mathsf{r}(x)) - \mu & y = k_{\min}(x). \end{cases}$ Note that such a choice

of $r_\mu$ leads to the following equality holds:

$$\sum_{k \notin \{\mathsf{r}(x), k_{\min}(x)\}} \widetilde{q}(x, k) \Phi_{\exp}(-r(x, k)) = \sum_{k \notin \{\mathsf{r}(x), k_{\min}(x)\}} \sum_{k \in \mathcal{K}} \widetilde{q}(x, k) \Phi_{\exp}(-r_\mu(x, k)).$$

Therefore, the conditional regret of cost-sensitive constrained exponential loss can be lower bounded as

$$\Delta \mathcal{C}_{\widetilde{\ell}_{\text{c-exp}}^{\text{cstnd}}, \mathcal{R}}(r, x)$$

$$\geq \inf_{r \in \mathcal{R}} \sup_{\mu \in \mathbb{R}} \left\{ \widetilde{q}(x, k_{\min}(x)) \left( e^{r(x, k_{\min}(x))} - e^{r(x, \mathsf{r}(x)) - \mu} \right) + \widetilde{q}(x, \mathsf{r}(x)) \left( e^{r(x, \mathsf{r}(x))} - e^{r(x, k_{\min}(x)) + \mu} \right) \right\}$$

$$= \left( \sqrt{\widetilde{q}(x, k_{\min}(x))} - \sqrt{\widetilde{q}(x, \mathsf{r}(x))} \right)^2 \qquad \text{(differentiating with respect to } \mu, r \text{ to optimize)}$$

$$= \left( \frac{\widetilde{q}(x, \mathsf{r}(x)) - \widetilde{q}(x, k_{\min}(x))}{\sqrt{\widetilde{q}(x, k_{\min}(x))} + \sqrt{\widetilde{q}(x, \mathsf{r}(x))}} \right)^2$$

$$\geq \frac{1}{4} \left( \widetilde{q}(x, \mathsf{r}(x)) - \widetilde{q}(x, k_{\min}(x)) \right)^2. \qquad (0 \leq \widetilde{q}(x, k) \leq 1)$$

Therefore, by Lemma I.1, the conditional regret of the target cardinality aware loss function can be upper bounded as follows:

$$\Delta \mathcal{C}_{\ell, \mathcal{H}}(r, x) = \widetilde{q}(x, \mathsf{r}(x)) - \widetilde{q}(x, k_{\min}(x)) \leq 2 \left( \Delta \mathcal{C}_{\widetilde{\ell}_{\text{c-exp}}^{\text{cstnd}}, \mathcal{R}}(r, x) \right)^{\frac{1}{2}}.$$

By the concavity, taking expectations on both sides of the preceding equation, we obtain

$$\mathcal{E}_\ell(r) - \mathcal{E}_\ell^*(\mathcal{R}) + \mathcal{M}_\ell(\mathcal{R}) \leq 2 \left( \mathcal{E}_{\widetilde{\ell}_{\text{c-exp}}^{\text{cstnd}}}(r) - \mathcal{E}_{\widetilde{\ell}_{\text{c-exp}}^{\text{cstnd}}}^*(\mathcal{R}) + \mathcal{M}_{\widetilde{\ell}_{\text{c-exp}}^{\text{cstnd}}}(\mathcal{R}) \right)^{\frac{1}{2}}.$$

The second part follows from the fact that $\mathcal{M}_{\widetilde{\ell}_{\text{c-exp}}^{\text{cstnd}}}(\mathcal{R}_{\text{all}}) = 0$.

**Case II:** $\ell = \widetilde{\ell}_{c-\text{sq-hinge}}$. For the cost-sensitive constrained squared hinge loss $\widetilde{\ell}_{c-\text{sq-hinge}}$, the conditional regret can be written as

$$\Delta \mathcal{C}_{\widetilde{\ell}_{c-\text{sq-hinge}}, \mathcal{R}}(r, x) = \sum_{k \in \mathcal{K}} \widetilde{q}(x, k) \Phi_{\text{sq-hinge}}(-r(x, k)) - \inf_{r \in \mathcal{R}} \sum_{k \in \mathcal{K}} \widetilde{q}(x, k) \Phi_{\text{sq-hinge}}(-r(x, k))$$

$$\geq \sum_{k \in \mathcal{K}} \widetilde{q}(x, k) \Phi_{\text{sq-hinge}}(-r(x, k)) - \inf_{\mu \in \mathbb{R}} \sum_{k \in \mathcal{K}} \widetilde{q}(x, k) \Phi_{\text{sq-hinge}}(-r_\mu(x, k)),$$

where for any $k \in \mathcal{K}$,

$$r_\mu(x, k) = \begin{cases} r(x, y), & y \notin \{k_{\min}(x), \mathsf{r}(x)\} \\ r(x, k_{\min}(x)) + \mu & y = \mathsf{r}(x) \\ r(x, \mathsf{r}(x)) - \mu & y = k_{\min}(x). \end{cases}$$

Note that such a choice of $r_\mu$ leads to the following equality holds:

$$\sum_{k \notin \{\mathsf{r}(x), k_{\min}(x)\}} \widetilde{q}(x, k) \Phi_{\text{sq-hinge}}(-r(x, k)) = \sum_{k \notin \{\mathsf{r}(x), k_{\min}(x)\}} \sum_{k \in \mathcal{K}} \widetilde{q}(x, k) \Phi_{\text{sq-hinge}}(-r_\mu(x, k)).$$

Therefore, the conditional regret of cost-sensitive constrained squared hinge loss can be lower bounded as

$$\Delta\mathcal{C}_{\widetilde{\ell}_{c-\mathrm{sq-hinge}},\mathcal{R}}(r,x)$$

$$\geq \inf_{r\in\mathcal{R}}\sup_{\mu\in\mathbb{R}}\left\{\widetilde{q}(x,k_{\min}(x))\Big(\max\{0,1+r(x,k_{\min}(x))\}^2 - \max\{0,1+r(x,\mathsf{r}(x))-\mu\}^2\Big)\right.$$

$$\left. + \widetilde{q}(x,\mathsf{r}(x))\Big(\max\{0,1+r(x,\mathsf{r}(x))\}^2 - \max\{0,1+r(x,k_{\min}(x))+\mu\}^2\Big)\right\}$$

$$\geq \frac{1}{4}\big(\widetilde{q}(x,k_{\min}(x))-\widetilde{q}(x,\mathsf{r}(x))\big)^2. \qquad \text{(differentiating with respect to } \mu, r \text{ to optimize)}$$

Therefore, by Lemma I.1, the conditional regret of the target cardinality aware loss function can be upper bounded as follows:

$$\Delta\mathcal{C}_{\ell,\mathcal{H}}(r,x) = \widetilde{q}(x,\mathsf{r}(x)) - \widetilde{q}(x,k_{\min}(x)) \leq 2\Big(\Delta\mathcal{C}_{\widetilde{\ell}_{c-\mathrm{sq-hinge}},\mathcal{R}}(r,x)\Big)^{\frac{1}{2}}.$$

By the concavity, taking expectations on both sides of the preceding equation, we obtain

$$\mathcal{E}_\ell(r) - \mathcal{E}_\ell^*(\mathcal{R}) + \mathcal{M}_\ell(\mathcal{R}) \leq 2\Big(\mathcal{E}_{\widetilde{\ell}_{c-\mathrm{sq-hinge}}}(r) - \mathcal{E}_{\widetilde{\ell}_{c-\mathrm{sq-hinge}}}^*(\mathcal{R}) + \mathcal{M}_{\widetilde{\ell}_{c-\mathrm{sq-hinge}}}(\mathcal{R})\Big)^{\frac{1}{2}}.$$

The second part follows from the fact that $\mathcal{M}_{\widetilde{\ell}_{c-\mathrm{sq-hinge}}}(\mathcal{R}_{\mathrm{all}}) = 0$.

**Case III:** $\ell = \widetilde{\ell}_{c-\mathrm{hinge}}$. For the cost-sensitive constrained hinge loss $\widetilde{\ell}_{c-\mathrm{hinge}}$, the conditional regret can be written as

$$\Delta\mathcal{C}_{\widetilde{\ell}_{c-\mathrm{hinge}},\mathcal{R}}(r,x) = \sum_{k\in\mathcal{K}}\widetilde{q}(x,k)\Phi_{\mathrm{hinge}}(-r(x,k)) - \inf_{r\in\mathcal{R}}\sum_{k\in\mathcal{K}}\widetilde{q}(x,k)\Phi_{\mathrm{hinge}}(-r(x,k))$$

$$\geq \sum_{k\in\mathcal{K}}\widetilde{q}(x,k)\Phi_{\mathrm{hinge}}(-r(x,k)) - \inf_{\mu\in\mathbb{R}}\sum_{k\in\mathcal{K}}\widetilde{q}(x,k)\Phi_{\mathrm{hinge}}(-r_\mu(x,k)),$$

where for any $k\in\mathcal{K}$,

$$r_\mu(x,k) = \begin{cases} r(x,y), & y\notin\{k_{\min}(x),\mathsf{r}(x)\} \\ r(x,k_{\min}(x))+\mu & y=\mathsf{r}(x) \\ r(x,\mathsf{r}(x))-\mu & y=k_{\min}(x). \end{cases}$$

Note that such a choice of $r_\mu$ leads to the following equality holds:

$$\sum_{k\notin\{\mathsf{r}(x),k_{\min}(x)\}}\widetilde{q}(x,k)\Phi_{\mathrm{hinge}}(-r(x,k)) = \sum_{k\notin\{\mathsf{r}(x),k_{\min}(x)\}}\sum_{k\in\mathcal{K}}\widetilde{q}(x,k)\Phi_{\mathrm{hinge}}(-r_\mu(x,k)).$$

Therefore, the conditional regret of cost-sensitive constrained hinge loss can be lower bounded as

$$\Delta\mathcal{C}_{\widetilde{\ell}_{c-\mathrm{hinge}},\mathcal{R}}(r,x)$$

$$\geq \inf_{r\in\mathcal{R}}\sup_{\mu\in\mathbb{R}}\left\{q(x,k_{\min}(x))(\max\{0,1+r(x,k_{\min}(x))\} - \max\{0,1+r(x,\mathsf{r}(x))-\mu\})\right.$$

$$\left. + q(x,\mathsf{r}(x))(\max\{0,1+r(x,\mathsf{r}(x))\} - \max\{0,1+r(x,k_{\min}(x))+\mu\})\right\}$$

$$\geq q(x,\mathsf{r}(x)) - q(x,k_{\min}(x)). \qquad \text{(differentiating with respect to } \mu, r \text{ to optimize)}$$

Therefore, by Lemma I.1, the conditional regret of the target cardinality aware loss function can be upper bounded as follows:

$$\Delta\mathcal{C}_{\ell,\mathcal{H}}(r,x) = \widetilde{q}(x,\mathsf{r}(x)) - \widetilde{q}(x,k_{\min}(x)) \leq \Delta\mathcal{C}_{\widetilde{\ell}_{c-\mathrm{hinge}},\mathcal{R}}(r,x).$$

By the concavity, taking expectations on both sides of the preceding equation, we obtain

$$\mathcal{E}_\ell(r) - \mathcal{E}_\ell^*(\mathcal{R}) + \mathcal{M}_\ell(\mathcal{R}) \leq \mathcal{E}_{\widetilde{\ell}_{c-\mathrm{hinge}}}(r) - \mathcal{E}_{\widetilde{\ell}_{c-\mathrm{hinge}}}^*(\mathcal{R}) + \mathcal{M}_{\widetilde{\ell}_{c-\mathrm{hinge}}}(\mathcal{R}).$$

The second part follows from the fact that $\mathcal{M}_{\widetilde{\ell}_{c-\mathrm{hinge}}}(\mathcal{R}_{\mathrm{all}}) = 0$.

**Case IV:** $\ell = \widetilde{\ell}_{c-\rho}$. For the cost-sensitive constrained $\rho$-margin loss $\widetilde{\ell}_{c-\rho}$, the conditional regret can be written as

$$\Delta\mathcal{C}_{\widetilde{\ell}_{c-\rho},\mathcal{R}}(r,x) = \sum_{k\in\mathcal{K}}\widetilde{q}(x,k)\Phi_\rho(-r(x,k)) - \inf_{r\in\mathcal{R}}\sum_{k\in\mathcal{K}}\widetilde{q}(x,k)\Phi_\rho(-r(x,k))$$

$$\geq \sum_{k\in\mathcal{K}}\widetilde{q}(x,k)\Phi_\rho(-r(x,k)) - \inf_{\mu\in\mathbb{R}}\sum_{k\in\mathcal{K}}\widetilde{q}(x,k)\Phi_\rho(-r_\mu(x,k)),$$

where for any $k \in \mathcal{K}$,

$$r_\mu(x,k) = \begin{cases} r(x,y), & y \notin \{k_{\min}(x),\mathsf{r}(x)\} \\ r(x,k_{\min}(x)) + \mu & y = \mathsf{r}(x) \\ r(x,\mathsf{r}(x)) - \mu & y = k_{\min}(x). \end{cases}$$

Note that such a choice of $r_\mu$ leads to the following equality holds:

$$\sum_{k\notin\{\mathsf{r}(x),k_{\min}(x)\}}\widetilde{q}(x,k)\Phi_\rho(-r(x,k)) = \sum_{k\notin\{\mathsf{r}(x),k_{\min}(x)\}}\sum_{k\in\mathcal{K}}\widetilde{q}(x,k)\Phi_\rho(-r_\mu(x,k)).$$

Therefore, the conditional regret of cost-sensitive constrained $\rho$-margin loss can be lower bounded as

$$\Delta\mathcal{C}_{\widetilde{\ell}_{c-\rho},\mathcal{R}}(r,x)$$

$$\geq \inf_{r\in\mathcal{R}}\sup_{\mu\in\mathbb{R}}\left\{\widetilde{q}(x,k_{\min}(x))\left(\min\left\{\max\left\{0,1+\frac{r(x,k_{\min}(x))}{\rho}\right\},1\right\} - \min\left\{\max\left\{0,1+\frac{r(x,\mathsf{r}(x))-\mu}{\rho}\right\},1\right\}\right)\right.$$

$$\left.+ \widetilde{q}(x,\mathsf{r}(x))\left(\min\left\{\max\left\{0,1+\frac{r(x,\mathsf{r}(x))}{\rho}\right\},1\right\} - \min\left\{\max\left\{0,1+\frac{r(x,k_{\min}(x))+\mu}{\rho}\right\},1\right\}\right)\right\}$$

$$\geq \widetilde{q}(x,\mathsf{r}(x)) - \widetilde{q}(x,k_{\min}(x)). \qquad \text{(differentiating with respect to } \mu, r \text{ to optimize)}$$

Therefore, by Lemma I.1, the conditional regret of the target cardinality aware loss function can be upper bounded as follows:

$$\Delta\mathcal{C}_{\ell,\mathcal{H}}(r,x) = \widetilde{q}(x,\mathsf{r}(x)) - \widetilde{q}(x,k_{\min}(x)) \leq \Delta\mathcal{C}_{\widetilde{\ell}_{c-\rho},\mathcal{R}}(r,x).$$

By the concavity, taking expectations on both sides of the preceding equation, we obtain

$$\mathcal{E}_\ell(r) - \mathcal{E}_\ell^*(\mathcal{R}) + \mathcal{M}_\ell(\mathcal{R}) \leq \mathcal{E}_{\widetilde{\ell}_{c-\rho}}(r) - \mathcal{E}_{\widetilde{\ell}_{c-\rho}}^*(\mathcal{R}) + \mathcal{M}_{\widetilde{\ell}_{c-\rho}}(\mathcal{R}).$$

The second part follows from the fact that $\mathcal{M}_{\widetilde{\ell}_{c-\rho}}(\mathcal{R}_{\text{all}}) = 0$. $\qquad\square$

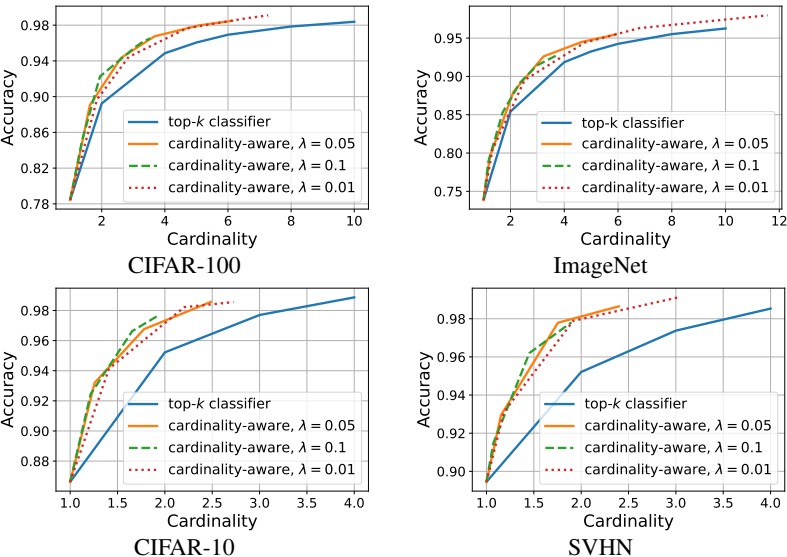

Figure 5: Accuracy versus cardinality on various datasets for $\text{cost}(|\mathbf{g}_k(x)|) = \log k$. Each curve of the cardinality-aware algorithm is for a fixed value of $\lambda$ and the points on the curve are obtained by varying the number of experts.

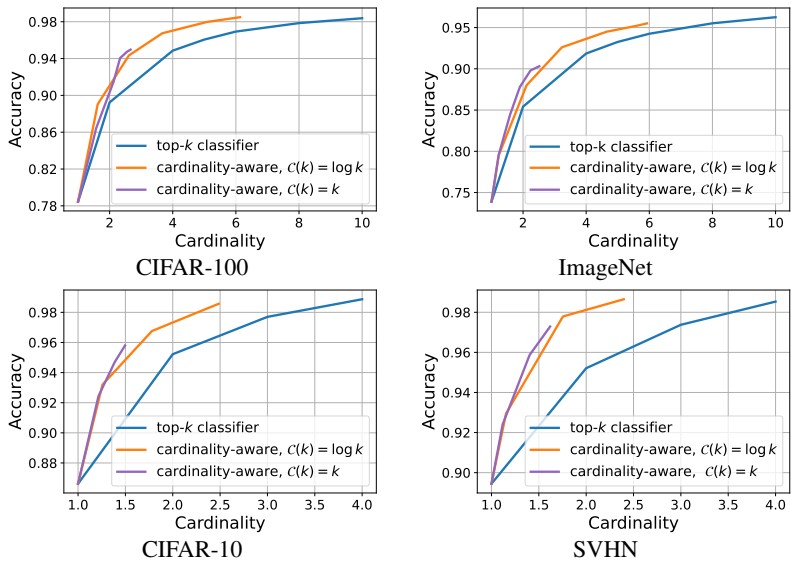

Figure 6: Accuracy versus cardinality on various datasets for $\text{cost}(|\mathbf{g}_k(x)|) = \log k$ and $\text{cost}(|\mathbf{g}_k(x)|) = k$, with $\lambda = 0.05$. The points on each curve of the cardinality-aware algorithm are obtained by varying the number of experts.

## J  Additional experimental results: top-$k$ classifiers

Here, we report additional experimental results with different choices of set $\mathcal{K}$ and $\text{cost}(|\mathbf{g}_k(x)|)$ on benchmark datasets CIFAR-10, CIFAR-100 [Krizhevsky, 2009], SVHN [Netzer et al., 2011], and ImageNet [Deng et al., 2009] and show that our cardinality-aware algorithm consistently outperforms top-$k$ classifiers across all configurations.

In Figure 5 and Figure 6, we began with a set $\mathcal{K} = \{1\}$ for the loss function and then progressively expanded it by adding choices of larger cardinality, each of which doubles the largest value currently in $\mathcal{K}$. The largest set $\mathcal{K}$ for the CIFAR-100 and ImageNet datasets is $\{1, 2, 4, 8, 16, 32, 64\}$, whereas for the CIFAR-10 and SVHN datasets, it is $\{1, 2, 4, 8\}$. As the set $\mathcal{K}$ expands, there is an increase in

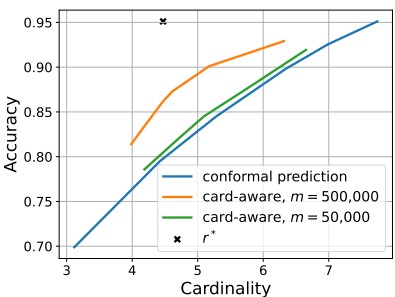

Figure 7: Accuracy versus cardinality on an artificial dataset for different training sample sizes $m$.

both the average cardinality and the accuracy. Figure 5 shows that the accuracy versus cardinality curve of the cardinality-aware algorithm is above that of top-$k$ classifiers for various values of $\lambda$. Figure 6 presents the comparison of $\text{cost}(|g_k(x)|) = k$ and $\text{cost}(|g_k(x)|) = \log k$ for $\lambda = 0.05$. These results demonstrate that different $\lambda$ and different $\text{cost}(|g_k(x)|)$ basically lead to the same curve, which verifies the effectiveness and benefit of our algorithm.

## K   Additional experimental results: threshold-based classifiers

We first characterize the Bayes predictor $r^*$ in this setting. We say that the scenario is deterministic if for all $x \in \mathcal{X}$, there exists some true label $y \in \mathcal{Y}$ such that $p(x, y) = 1$; otherwise, we say that the scenario is stochastic. To simplify the discussion, we will assume that $|g_k(x)|$ is an increasing function of $k$, for any $x$.

**Lemma K.1.** *Consider the deterministic scenario. Assume that $\lambda\text{cost}(|g_k(x)|) \leq 1$ for all $k$ and $x \in \mathcal{X}$. Then, the Bayes predictor $r^*$ for the cardinality-aware loss function $\ell$ satisfies:* $r^*(x) = \text{argmin}_{k:y\in g_k(x)} k$, *that is the smallest $k$ such that the true label $y$ is in $g_k(x)$.*

*Proof.* By the assumption, for $k < r^*(x)$, we can write $c(x, r^*(x), y) = \lambda\text{cost}(|g_{r^*(x)}(x)|) \leq 1 \leq 1_{y\notin g_k(x)} + \lambda\text{cost}(|g_k(x)|) = c(x, k, y)$. Furthermore, since $|g_k(x)|$ is an increasing function of $k$, we have $c(x, r^*(x), y) = \lambda\text{cost}(|g_{r^*(x)}(x)|) \leq \lambda\text{cost}(|g'_k(x)|) = c(x, k', y)$ for $k' > r^*(x)$. $\square$

**Lemma K.2.** *Consider the stochastic scenario. The Bayes predictor $r^*$ for the cardinality-aware loss function $\ell$ satisfies:*

$$r^*(x) = \underset{k\in\mathcal{K}}{\text{argmin}}\left(\lambda\text{cost}(|g_k(x)|) - \sum_{y\in g_k(x)} p(x, y)\right).$$

*Proof.* The conditional error can be written as follows:

$$
\begin{aligned}
\mathcal{C}_\ell(r, x, y) &= \sum_{y\in\mathcal{Y}} p(x, y)c(x, r(x), y) \\
&= \sum_{y\in\mathcal{Y}} p(x, y)\left(1_{y\notin g_{r(x)}(x)} + \lambda\text{cost}(|g_{r(x)}(x)|)\right) \\
&= \sum_{y\in\mathcal{Y}} p(x, y)1_{y\notin g_{r(x)}(x)} + \lambda\text{cost}(|g_{r(x)}(x)|) \\
&= 1 - \sum_{y\in g_{r(x)}(x)} p(x, y) + \lambda\text{cost}(|g_{r(x)}(x)|).
\end{aligned}
$$

Thus, the Bayes predictor can be characterized as

$$r^*(x) = \underset{k\in\mathcal{K}}{\text{argmin}}\left(\lambda\text{cost}(|g_k(x)|) - \sum_{y\in g_k(x)} p(x, y)\right).$$

$\square$

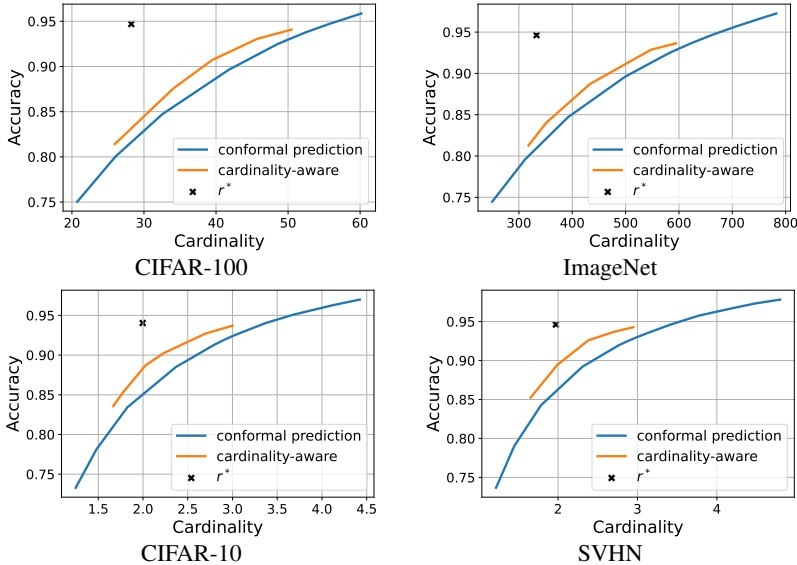

Figure 8: Accuracy versus cardinality on CIFAR-100, ImageNet, CIFAR-10, and SVHN datasets.

It is clear that Lemma K.2 implies Lemma K.1 when there exists some true $y \in \mathcal{Y}$ such that $p(x, y) = 1$ and $\lambda \mathsf{cost}(|\mathsf{g}_k(x)|) \leq 1$.

We first consider an artificial dataset containing 10 classes. Each class is modeled by a Gaussian distribution in a 100-dimensional space. As in Section 5, we plot the accuracy versus cardinality curve of the cardinality-aware algorithm by varying $\lambda$, where the set predictors used are threshold-based classifiers, and compare with that of conformal prediction. In Figure 7, we also indicate the point corresponding to $r^*$. The problem is close to being realizable, as we can train a predictor that performs almost as well as $r^*$ on the test set. Thus, the minimizability gaps vanish, and our $\mathcal{H}$-consistency bounds (Theorems 4.6 and 4.7) then suggest that with sufficient training data, we can get close to the optimal solution and therefore outperform conformal prediction. For some tasks, however, the problem is hard, and it appears that a very large training sample would be needed. Figure 7 demonstrates that on the artificial dataset, with training sample size $m = 50{,}000$, the performance of our cardinality-aware algorithm is only slightly better than that of conformal prediction. If we increase the training sample size to $m = 500{,}000$, then the curve of our algorithm becomes much closer to the optimal point and significantly outperforms conformal prediction.

Additionally, for a weaker scoring function, a smaller training sample suffices in many cases, and our cardinality-aware algorithm can outperform conformal prediction on real datasets as shown in Figure 8.

# L   Future work

While our framework of cardinality-aware set prediction is very general—applicable to any collection of set predictors (Section 2)—and leads to novel cardinality-aware algorithms (Section 3), benefits from theoretical guarantees with sufficient training data (Section 4), and demonstrates effectiveness and empirical advantages in top-$k$ classification (Section 5), the learning problem can be challenging for certain tasks, often requiring a very large training sample (as shown in Appendix K). This underscores the need for a more detailed investigation to enhance our algorithms in these scenarios.

