# OpenReview forum: "Cardinality-Aware Set Prediction and Top-$k$ Classification"
_NeurIPS.cc/2024/Conference — NeurIPS 2024 poster_

### Official Review · Reviewer_a6WU · 2024-07-09

**Soundness:** 3
**Presentation:** 4
**Contribution:** 4
**Rating:** 8
**Confidence:** 2

**Summary:**

This paper proposes a method to handle cardinality-aware top-k classification. It designs an optimization problem where the cardinality of  set selectors is also considered. The authors then generalize it to two types of surrogate losses. Given certain assumptions on the hypothesis set H, the authors theoretically justify their method.

**I must say I am no expert on this domain, so the AC must take my comments wisely.**

**Strengths:**

This paper tackles a very important problem. And achieves, according to my view, a very nice result. The authors manage to transfer the cardinality-aware term into surrogate terms, and theoretically demonstrate their sub-optimal properties. This paper is also very well written.

**Weaknesses:**

No error bar, very limited evaluation, very strong assumption on the hypothesis set, which is usually not easy to achieve in practice.

**Questions:**

In some cases, like those generative modeling questions, the cardinality of predictors may be evaluated by measure, meaning the true answer assembles a region of infinite points. Can your method apply to those scenarios where the cardinality is evaluated by measures?

**Limitations:**

As said in weakness.

---

> ### Author Rebuttal · Authors · 2024-08-06
>
> Thank you for your appreciation of our work. We will take your suggestions into account when preparing the final version. Below please find responses to specific questions.
>
> **Weaknesses: No error bar, very limited evaluation, very strong assumption on the hypothesis set, which is usually not easy to achieve in practice.**
>
> **Response:** We will include error bars in the final version, which show minimal variability and do not impact the demonstrated superiority of our cardinality-aware algorithms. We will also follow Reviewer 5ki4's suggestions to provide a more comprehensive experimental analysis of various aspects of our cardinality-aware algorithms in the final version. In the global response, we have included two additional experimental results related to this analysis. The assumptions of symmetry and completeness in Section 4.3 can be met by common hypothesis sets, including classes of linear models and multilayer feedforward neural networks typically used in practice. Note that these assumptions are made on the hypothesis set $\mathcal{R}$ of the cardinality selector, which we can choose ourselves. We impose no specific assumptions about the hypothesis set of base classifiers $h$ used in our cardinality-aware algorithms, which can be any practical hypothesis set used for various tasks.
>
> **Questions: In some cases, like those generative modeling questions, the cardinality of predictors may be evaluated by measure, meaning the true answer assembles a region of infinite points. Can your method apply to those scenarios where the cardinality is evaluated by measures?**
>
> **Response:** That's a great question. Our method can indeed be applied to those scenarios as well, where the corresponding measure—rather than standard cardinality—is taken into account in the cost function. Our analysis is general and makes no assumptions about costs, which provides flexibility in choosing costs adapted to different cases, including the one the reviewer pointed out. We will elaborate on this in the final version.

---

> > ### Comment · Reviewer_a6WU · 2024-08-09
> > **re**
> >
> > Good work. Please give me a short summary of your new analysis regarding the question before the due date. I have raised the score to 8.

---

> > > ### Author Response · Authors · 2024-08-11
> > >
> > > We thank the reviewer for their support of our work.
> > >
> > > Here's a short summary. In our additional experimental analysis attached in the global response, we found further evidence that our cardinality-aware algorithms effectively adjust the size of the prediction sets. Figure 1 illustrates how higher costs result in smaller prediction sets, while Figure 2 demonstrates that visually challenging images are associated with larger prediction sets.
> > >
> > > In the final version, we will expand upon these findings with significantly more empirical evidence, demonstrating a clear correlation between image difficulty and prediction set cardinality. This aligns with the core objective of our algorithm and theoretical framework. We will also include error bars to highlight the statistical significance of our results.
> > >
> > > Please feel free to reach out if the reviewer has any further questions.

---

### Official Review · Reviewer_5ki4 · 2024-07-15

**Soundness:** 3
**Presentation:** 2
**Contribution:** 2
**Rating:** 4
**Confidence:** 2

**Summary:**

This paper introduces a new loss function for top-k set prediction where k may vary as a function of the input, which the authors call cardinality-aware top-k classification. However, this loss is intractable to optimize in all but trivial cases, so the authors introduce surrogate loss functions for learning these cardinality-aware set predictors, and provide a theoretical analysis of these surrogate losses. The paper also explores the proposed algorithms empirically, on benchmark computer vision datasets.

**Strengths:**

- The contribution seems novel -- the authors identify a new type of prediction problem in which top-k sets of input-varying size can be predicted.
- The introduction of two new surrogate loss functions for optimizing the proposed (but intractable) cardinality-aware top-k loss seems to be a good contribution.
- Empirically, the authors show on benchmark datasets that by giving up control over specific cardinalities, their top-k predictors can outperform existing top-k predictors that prescribe specific cardinalities a priori.

**Weaknesses:**

- It isn't immediately clear why the cost-constrained surrogate loss is tractable to optimize, while the original (non-surrogate) loss is intractable. Please clarify this in the text.
- (clarity of the abstract) The abstract should elaborate on H-consistency and the types of losses being introduced, or discuss these at a higher level. Without knowing what these are prior to reading the abstract, the significance of these is unclear.
- The usage of the term "accuracy" in Figure 1 is strange, as the actual "accuracy" definitions vary slightly between the two methods being compared. Additionally, the expected cardinality is used, which makes the comparison strange. Furthermore, this comparison on four benchmark computer vision datasets makes up the bulk of the experimental results. The experimental results section could be additionally strengthened by analyzing other aspects of the predictors generated by the authors' framework, such as the distribution of cardinalities, whether or not harder examples indeed correspond to higher cardinality, the top-k accuracy scores of worst-case cardinalities and best-case cardinalities, and so on.
- The value of Figure 2 is unclear. What is the significance of making the cost function linear?

**Questions:**

- Out of curiosity, have the authors considered simpler techniques for cardinality-varying top-k prediction, such as setting a total probability threshold on prediction probabilities, and returns the top-k that maximizes the probability, subject to the threshold? At least empirically, it seems like simpler techniques such as this one should be included as a baseline.
- Is it in fact the case that for "harder" examples, the top-k cardinality of the predictions are larger in this framework? Have the authors explored this empirically? Examples of this would strengthen the argument for this new type of prediction.

**Limitations:**

The authors discuss limitations.

---

> ### Author Rebuttal · Authors · 2024-08-06
>
> Thank you for your thoughtful feedback and suggestions. We will take them all into account when preparing the final version. Below please find responses to specific questions.
>
> **Weaknesses:**
>
> **1. It isn't immediately clear why the cost-constrained surrogate loss is tractable to optimize, while the original (non-surrogate) loss is intractable. Please clarify this in the text.**
>
> **Response:** The original loss comprises indicator functions, which are discrete and non-continuous, making them difficult to optimize. In contrast, the cost-sensitive surrogate loss is continuous, smooth, and differentiable, which makes it tractable for optimization. This is analogous to how the zero-one loss is intractable to optimize in standard classification, whereas the standard constrained loss is tractable. We will clarify this point in the text.
>
> **2. (clarity of the abstract) The abstract should elaborate on H-consistency and the types of losses being introduced, or discuss these at a higher level. Without knowing what these are prior to reading the abstract, the significance of these is unclear.**
>
> **Response:** Thank you for the suggestion. We will certainly provide additional explanations about the new cost-sensitive loss functions proposed and their H-consistency guarantees. This will highlight the novelty of our surrogate losses and underscore the strong theoretical foundation of our algorithms.
>
> **3. The usage of the term "accuracy" ... and so on.**
>
> **Response:** We use the same definition of accuracy for both methods: for top-$k$ classifiers and cardinality-aware algorithms, accuracy measures the fraction of samples where the prediction sets include the true label. However, the definition of the prediction set differs between the methods. In top-$k$ classifiers, the prediction set consists of the labels associated with the top $k$ scores. In contrast, for cardinality-aware algorithms, the prediction set is determined by the set predictor $\mathsf g_k$ chosen by the selector $r$.
>
> The average cardinality appears to be a natural measure of the size of the prediction sets. But, we are open to alternative metrics suggested by reviewers and would be happy to include results based on such alternatives.
>
> We appreciate the feedback on enhancing the experimental section. We will provide a more detailed analysis of our cardinality-aware algorithms in the final version. Our algorithm dynamically adjusts prediction set cardinality based on input complexity—choosing larger sets for more challenging inputs to ensure high accuracy and smaller sets for simpler inputs to keep the cardinality low. This dynamic adjustment has been validated through our experiments.
>
> In the global response, we have included two additional experimental results related to this analysis:
>
> Figure 1 illustrates the cardinality distribution for top-$k$ experts $\mathcal{K} =  \\{1, 2, 4, 8\\}$ for the CIFAR-10 and CIFAR-100 datasets, analyzed under two different $\lambda$ values. For a given dataset, increasing $\lambda$ results in fewer samples with the largest cardinality of $8$, and more samples with smaller cardinalities. This is because increasing $\lambda$ amplifies the impact of cardinality in the cost functions. When comparing across different datasets, the distribution differs for the same $\lambda$ due to the varying complexity levels of the classification tasks.
>
> Figure 2 illustrates the comparison of hard and easy images as judged by humans in original quality on the CIFAR-10 dataset for top-$k$ experts $\mathcal{K} =  \\{1, 2, 4, 8\\}$. *Hard images* are predicted correctly by our algorithms with a cardinality of $8$ but predicted incorrectly when using a cardinality of $4$ instead. *Easy images* are correctly predicted by our algorithms with a cardinality of $1$.
>
> Additionally, we wish to emphasize that further experiments with top-$k$ classifiers (see Figures 3 and 4 in Appendix J) and threshold-based classifiers (see Figures 5 and 6 in Appendix K) illustrate the robustness and superiority of our cardinality-aware algorithms.
>
> **4. The value of Figure 2 is unclear. What is the significance of making the cost function linear?**
>
> **Response:** The value of Figure 2 lies in demonstrating that the choice of the cost function—whether linear or logarithmic—has a negligible impact on our algorithm's performance. This highlights its robustness in this regard, as our framework is very general and allows for the choice of any cost function.
>
> **Questions:**
>
> **1. Out of curiosity, have the authors ... included as a baseline.**
>
> **Response:** We have included additional experiments with threshold-based classifiers (Figures 5 and 6) in Appendix K, which demonstrate the superiority of our cardinality-aware algorithms in these scenarios. These include a comparison to conformal prediction, which provides cardinality-varying prediction sets using a threshold on prediction probabilities. We are happy to provide more comparisons as requested by the reviewers.
>
> **2. Is it in fact the case that for "harder" examples ... this new type of prediction.**
>
> **Response:** Yes, our algorithm dynamically adjusts the cardinality of its prediction sets based on input instances. It selects larger sets for more difficult inputs to ensure high accuracy and opts for smaller sets for simpler inputs to maintain low cardinality. This property of our algorithm has been substantiated through our experimental analysis. In the global response, we present one example (Figure 2) from the analysis, with additional examples to be included in the final version. Figure 2 illustrates the comparison of hard and easy images as judged by humans in original quality on the CIFAR-10 dataset for top-$k$ experts $\mathcal{K} =  \\{1, 2, 4, 8\\}$. *Hard images* are predicted correctly by our algorithms with a cardinality of $8$ but predicted incorrectly when using a cardinality of $4$ instead. *Easy images* are correctly predicted by our algorithms with a cardinality of $1$.

---

### Official Review · Reviewer_TDsY · 2024-07-21

**Soundness:** 3
**Presentation:** 3
**Contribution:** 3
**Rating:** 7
**Confidence:** 3

**Summary:**

The paper introduced a new cardinality-aware set prediction algorithm with cost-sensitive comp-sum and constrained surrogate losses. Additionally, the paper established theoretical guarantees for top-k classification with fixed cardinality k using the H-consistency bounds. Finally, experiments on linear classifiers show the superiority of their algorithms.

**Strengths:**

1. The paper is presented clearly and is easy to follow.
2. The paper considers the set prediction, which is a novel setting in recent years.
3. The paper proposes new algorithms with rigorous guarantees and good performance on various datasets.

**Weaknesses:**

1. Since the theoretical results include the neural network class as a special case, the experiments should additionally consider training whole neural networks on the mentioned datasets (at least CIFAR), which is more important in practice.

**Questions:**

1. Can we estimate the $M$ in H-consistency bounds?
2. Authors claim the framework includes the standard conformal prediction, so do some relationships exist between the H-consistency bounds and coverage bounds?

**Limitations:**

The authors have adequately discussed the limitations.

---

> ### Author Rebuttal · Authors · 2024-08-06
>
> Thank you for your encouraging review. We will take your suggestions into account when preparing the final version. Please find responses to your specific questions below.
>
> **Weaknesses:**
>
> **1. Since the theoretical results include the neural network class as a special case, the experiments should additionally consider training whole neural networks on the mentioned datasets (at least CIFAR), which is more important in practice.**
>
> **Response:** The main focus of our cardinality-aware algorithm is in training the cardinality selector $r$, which we do in fact implement as a neural network. We used a two-hidden-layer feedforward neural network, which aligns with the assumptions of our theoretical guarantees for cardinality-aware surrogate losses. The base classifier $h$ is derived from well-trained neural networks (ResNet for CIFAR-10, CIFAR-100, and SVHN datasets; CLIP for ImageNet) with an additional linear model on top. This setup mirrors common practices in the field such as linear probing, resembling the fine-tuning of existing neural network models.
>
> Furthermore, we are happy to report additional experimental results in which the base classifier $h$ is trained on entire neural networks without fine-tuning in the final version. However, we do not anticipate significant differences in results, as our algorithm focuses on training the cardinality selector $r$. The base classifier $h$ in our approach is pre-trained and remains fixed, providing specified costs and inputs to the algorithm.
>
> We would also like to clarify that our theoretical guarantees in Section 4.3 make no assumptions about the base classifier $h$, which appears in the cost functions. We also assume only symmetry and completeness for the cardinality selector $r$.
>
> **Questions:**
>
> **1. Can we estimate the $M$ in H-consistency bounds?**
>
> **Response:** This is an excellent question. Currently, no general method exists for accurately estimating $M$ from finite samples because the problem seems to require estimating both the best-in-class loss and the average pointwise infimum. However, useful upper bounds can be derived in specific cases. For instance, Theorems 4.1 and 4.2 in Mao et al.'s paper [1] provide bounds for the family of comp-sum losses.
>
> The challenge of deriving more precise estimates based on samples, along with leveraging known information about the hypothesis set and distribution, remains an interesting avenue for future research.
>
> Additionally, minimizability gaps are upper bounded by the approximation error, which is often small or even zero in practical applications of neural networks. In cases where the approximation error is zero, minimizability gaps also vanish.
>
> [1] Mao et al., "Cross-Entropy Loss Functions: Theoretical Analysis and Applications," ICML 2023.
>
> **2. Authors claim the framework includes the standard conformal prediction, so do some relationships exist between the H-consistency bounds and coverage bounds?**
>
> **Response:** Yes, our framework covers threshold-based classifiers, including those used in conformal prediction. In conformal prediction, the threshold is determined using an $\alpha$-quantile based on a scoring function derived from a validation set. However, while standard conformal prediction provides coverage bounds that guarantee a desired $(1 - \alpha)$ accuracy, it does not provide any guarantee regarding the size of the prediction set.
>
> In contrast, our approach offers guarantees that account for both accuracy and the size of the prediction set. Specifically, for our cardinality-aware algorithms, $H$-consistency bounds ensure that the cardinality selector $r$ is optimized to minimize the cardinality-aware loss function. This means that each input instance $x$ is assigned the most appropriate prediction set to achieve high accuracy while also keeping the average cardinality low.
>
> Therefore, $H$-consistency bounds provide a stronger guarantee than standard coverage bounds, as they address both accuracy and cardinality, rather than focusing solely on accuracy.

---

> > ### Comment · Reviewer_TDsY · 2024-08-08
> >
> > Thank you for providing the rebuttal. The authors adequately addressed my concerns and promised to revise the paper according to the response (e.g., adding new experimental results). So, my rating has been increased from 6 to 7.

---

> > > ### Author Response · Authors · 2024-08-08
> > >
> > > We are glad that we have addressed the reviewer's concerns. We thank the reviewer once again for their valuable suggestions and insightful comments. Please let us know if there is any other question we can address.

---

### Author Rebuttal · Authors · 2024-08-06

We thank all reviewers for their insightful comments. We share additional experimental results in the attached PDF, as suggested by Reviewer 5ki4.

Figure 1 illustrates the cardinality distribution for top-$k$ experts $\mathcal{K} =  \\{1, 2, 4, 8\\}$ for the CIFAR-10 and CIFAR-100 datasets, analyzed under two different $\lambda$ values. For a given dataset, increasing $\lambda$ results in fewer samples with the largest cardinality of $8$, and more samples with smaller cardinalities. This is because increasing $\lambda$ amplifies the impact of cardinality in the cost functions. When comparing across different datasets, the distribution differs for the same $\lambda$ due to the varying complexity levels of the classification tasks.

Figure 2 illustrates the comparison of hard and easy images as judged by humans in original quality on the CIFAR-10 dataset for top-$k$ experts $\mathcal{K} =  \\{1, 2, 4, 8\\}$. *Hard images* are predicted correctly by our algorithms with a cardinality of $8$ but predicted incorrectly when using a cardinality of $4$ instead. *Easy images* are correctly predicted by our algorithms with a cardinality of $1$.

---

### Decision · Program_Chairs · 2024-09-25

**Decision:**

Accept (poster)

**Comment:**

The paper introduces a novel cardinality-aware set prediction algorithm that incorporates cost-sensitive comp-sum and constrained surrogate losses. It also provides theoretical guarantees for top-k classification with a fixed cardinality k using H-consistency bounds. Experimental results with linear classifiers demonstrate the superior performance of the proposed algorithms.

Although the paper has a mixed rating, reviewer 5ki4 (with a rating of 4) didn't participate in the discussion nor respond to the rebuttal. I have read the comments, which are mostly related to the presentation, and believe that the rebuttal has clarified the issues.

Therefore, I recommend acceptance.